# Variance-Reduced Off-Policy TDC Learning: Non-Asymptotic Convergence Analysis

**Shaocong Ma**
Department of ECE
University of Utah
Salt Lake City, UT 84112
s.ma@utah.edu

**Yi Zhou**
Department of ECE
University of Utah
Salt Lake City, UT 84112
yi.zhou@utah.edu

**Shaofeng Zou**
Department of EE
University at Buffalo
Buffalo, NY 14260
szou3@buffalo.edu

## Abstract

Variance reduction techniques have been successfully applied to temporal-difference (TD) learning and help to improve the sample complexity in policy evaluation. However, the existing work applied variance reduction to either the less popular one time-scale TD algorithm or the two time-scale GTD algorithm but with a finite number of i.i.d. samples, and both algorithms apply to only the on-policy setting. In this work, we develop a variance reduction scheme for the two time-scale TDC algorithm in the off-policy setting and analyze its non-asymptotic convergence rate over both i.i.d. and Markovian samples. In the i.i.d. setting, our algorithm achieves a sample complexity $\mathcal{O}(\epsilon^{-\frac{3}{5}} \log \epsilon^{-1})$ that is lower than the state-of-the-art result $\mathcal{O}(\epsilon^{-1} \log \epsilon^{-1})$. In the Markovian setting, our algorithm achieves the state-of-the-art sample complexity $\mathcal{O}(\epsilon^{-1} \log \epsilon^{-1})$ that is near-optimal. Experiments demonstrate that the proposed variance-reduced TDC achieves a smaller asymptotic convergence error than both the conventional TDC and the variance-reduced TD.

## 1 Introduction

In reinforcement learning applications, we often need to evaluate the value function of a target policy by sampling the Markov decision process (MDP) generated by either the target policy (on-policy) or a certain behavior policy (off-policy) [3, 9, 20, 24, 22, 27]. In the on-policy setting, temporal-difference (TD) learning [23, 24] and Q-learning [9] algorithms have been developed with convergence guarantee. However, in the more popular off-policy setting, these conventional policy evaluation algorithms have been shown to possibly diverge under linear function approximation [2]. To address this issue, [18, 25, 26] developed a family of gradient-based TD (GTD) algorithms that have convergence guarantee in the off-policy setting. In particular, the TD with gradient correction (TDC) algorithm has been shown to have superior performance and is widely used in practice.

Although TD-type algorithms achieve a great success in policy evaluation, their convergence suffer from a large variance caused by the stochastic samples obtained from a dynamic environment. A conventional approach that addresses this issue is to use a diminishing stepsize [4, 21], but it significantly slows down the convergence in practice. Recently, several work proposed to apply the variance reduction technique developed in the stochastic optimization literature to reduce the variance of TD learning. Specifically, [11] considered a convex-concave TD learning problem with a finite number of i.i.d. samples, and they applied the SVRG [12] and SAGA [10] variance reduction techniques to develop variance-reduced primal-dual batch gradient algorithms for solving the problem. In [19], two variants of SVRG-based GTD2 algorithms for i.i.d. samples were proposed to save the computation cost. While these work developed variance-reduced TD-type algorithms

for i.i.d. samples, practical reinforcement learning applications often involve *non-i.i.d.* samples that are generated by an underlying MDP. In [14], the authors proposed a variance-reduced TD (VRTD) algorithm for Markovian samples in the on-policy setting. However, their analysis of the algorithm has a technical error and has been corrected in the recent work [28]. To summarize, the existing developments of variance-reduced TD-type algorithms consider only the on-policy setting, and only the one time-scale VRTD algorithm applies to Markovian samples. Therefore, it is very much desired to develop a two time-scale variance-reduced algorithm in the off-policy setting for Markovian samples, which constitutes to **the goal of this paper:** we develop two variance-reduced TDC algorithms in the off-policy setting for i.i.d. samples and Markovian samples, respectively, and analyze their non-asymptotic convergence rates. We summarize our contributions as follows.

## 1.1  Our Contributions

We develop two variance-reduced TDC algorithms (named VRTDC) respectively for i.i.d. samples and Markovian samples in the off-policy setting and analyze their non-asymptotic convergence rates as well as sample complexities. To the best of our knowledge, our work provides the first study on variance reduction for two time-scale TD learning over Markovian samples.

For i.i.d. samples with constant step sizes $\alpha, \beta$, we show that VRTDC converges at a linear rate to a neighborhood of the optimal solution with an asymptotic convergence error $\mathcal{O}(\beta M^{-1} + \beta^4)$, where $M$ denotes the batch size of the outer-loops. Consequently, to achieve an $\epsilon$-accurate solution, the required sample complexity for VRTDC is $\mathcal{O}(\epsilon^{-\frac{3}{5}} \log \epsilon^{-1})$, which is lower than that $\mathcal{O}(\epsilon^{-1} \log \epsilon^{-1})$ of VRTD for i.i.d. samples [28]. Also, the tracking error of VRTDC diminishes linearly with an asymptotic error $\mathcal{O}(\beta M^{-1} + \beta^3)$ and has a corresponding sample complexity $\mathcal{O}(\epsilon^{-\frac{1}{2}} \log \epsilon^{-1})$. For Markovian samples with constant step sizes $\alpha, \beta$, we show that VRTDC converges at a linear rate to a neighborhood of the optimal solution with an asymptotic convergence error $\mathcal{O}(M^{-1} + \beta^2)$, and the required sample complexity to achieve an $\epsilon$-accurate solution is $\mathcal{O}(\epsilon^{-1} \log \epsilon^{-1})$, which matches the best existing result of VRTD [28] and TDC [13] and nearly matches the theoretical lower bound $\mathcal{O}(\epsilon^{-1})$ [13]. Also, the tracking error of VRTDC diminishes linearly with an asymptotic error $\mathcal{O}(M^{-1} + \beta^3)$ and has a corresponding sample complexity $\mathcal{O}(\epsilon^{-1} \log \epsilon^{-1})$. Furthermore, our experiments on the Garnet problem and frozen lake game demonstrate that VRTDC achieves a smaller asymptotic convergence error than both the conventional TDC and the variance-reduced TD.

Our analysis of VRTDC requires substantial developments of bounding techniques. Specifically, we develop much refined bounds for the tracking error and the convergence error via a recursive refinement strategy: we first develop a preliminary bound for the tracking error $\|\tilde{z}^{(m)}\|^2$ and then use it to develop a preliminary bound for the convergence error $\|\theta^{(m)} - \theta^*\|^2$. Then, by leveraging the relation between tracking error and convergence error induced by the two time-scale updates of VRTDC, we further utilize the preliminary bound for the convergence error to obtain a refined bound for the tracking error. Finally, we apply the refined bound for the tracking error to develop a refined bound for the convergence error by leveraging the two time-scale updates. These refined bounds are the key to establish the reported sample complexities of VRTDC.

## 1.2  Related Work

**Off-policy two time-scale TDC and SA:**  Two time-scale policy evaluation algorithms such as TDC and GTD2 were first introduced in [26], where the asymptotic convergence of both algorithms with i.i.d. samples were established. Their non-asymptotic convergence rates were established in [8] as special cases of a two time-scale linear stochastic approximation (SA) algorithm. Recently, the non-asymptotic convergence analysis of TDC and two-time scale linear SA over Makovian samples were established in [29] and [13], respectively.

**TD learning with variance reduction:**  In the existing literature, two settings of variance reduction for TD learning have been considered. The first setting considers evaluating the value function based on a fixed number of samples. In this setting, it is preferred to use the batch TD algorithm [17]. [11] rewrote the original MSPBE minimization problem into a convex-concave saddle-point optimization problem and applied SVRG and SAGA to the primal-dual batch gradient algorithm. The second setting considers online policy evaluation problem, where the trajectory follows from an MDP. In [14], the variance-reduced TD algorithm was introduced for solving the MSPBE minimization problem. [28] pointed out a technical error in the analysis of [14] and provided a correct non-asymptotic analysis for the variance-reduced TD algorithm over Markovian samples.

## 2 Preliminaries: Off-Policy TDC with Linear Function Approximation

In this section, we provide an overview of TDC learning with linear function approximation in the off-policy setting and define the notations that are used throughout the paper.

### 2.1 Off-Policy Value Function Evaluation

In reinforcement learning, an agent interacts with the environment via a Markov decision process (MDP) that is denoted as $(\mathcal{S}, \mathcal{A}, \mathbf{P}, r, \gamma)$. Specifically, $\mathcal{S}$ denotes a state space and $\mathcal{A}$ denotes an action space, both of which are assumed to have finite number of elements. Then, a given policy $\pi$ maps a state $s \in \mathcal{S}$ to a certain action in $\mathcal{A}$ following a conditional distribution $\pi(\cdot|s)$. The associated Markov chain is denoted as $p(s'|s) := \sum_{a \in \mathcal{A}} \mathbf{P}(s'|s, a)\pi(a|s)$ and is assumed to be ergodic, and the induced stationary distribution is denoted as $\mu_\pi(s') := \sum_s p(s'|s)\mu_\pi(s)$.

In the off-policy setting, the action of the agent is determined by a behavior policy $\pi_b$, which controls the behavior of the agent as follows: Suppose the agent is in a certain state $s_t$ at time-step $t$ and takes an action $a_t$ based on the policy $\pi_b(\cdot|s_t)$. Then, the agent transfers to a new state $s_{t+1}$ in the next time-step according to the transition kernel $\mathbf{P}(s_{t+1}|s_t, a_t)$ and receives a reward $r_t = r(s_t, a_t, s_{t+1})$. The goal of off-policy value function evaluation is to use samples of the MDP to estimate the following value function $V^\pi(s)$ of the target policy $\pi$.

$$V^\pi(s) = \mathbb{E}\Big[\sum_{t=0}^{\infty} \gamma^t r(s_t, a_t, s_{t+1})|s_0 = s, \pi\Big],$$

where $\gamma \in (0, 1)$ is a discount factor. Define the Bellman operator $T^\pi$ for any function $\xi(s)$ as $T^\pi\xi(s) := r^\pi(s) + \gamma\mathbb{E}_{s'|s}\xi(s')$, where $r^\pi(s) = \mathbb{E}_{a,s'|s}r(s, a, s')$ is the expected reward of the Markov chain induced by the policy $\pi$. It is known that the value function $V^\pi(s)$ is the unique fixed point of the Bellman operator $T^\pi$, i.e., $V^\pi(s) = T^\pi V^\pi(s)$.

### 2.2 TDC Learning with Linear Function Approximation

In practice, the state space $\mathcal{S}$ usually contains a large number of states that induces much computation overhead in policy evaluation. To address this issue, a popular approach is to approximate the value function via linear function approximation. Specifically, given a set of feature functions $\phi_i : \mathcal{S} \to \mathbb{R}$ for $i = 1, 2, \ldots, d$ and define $\phi = [\phi_1, ..., \phi_d]^\top$, the value function of a given state $s$ can be approximated via the linear model $\widehat{V}_\theta(s) := \phi(s)^\top\theta$, where $\theta = [\theta_1, \theta_2, ..., \theta_d]^\top$ denotes all the model parameters. Suppose the state space includes states $s_1, ..., s_n$, we denote the total value function as $\widehat{V}_\theta := [\widehat{V}_\theta(s_1), ..., \widehat{V}_\theta(s_n)]^\top = \Phi\theta$, where $\Phi := [\phi(s_1), ..., \phi(s_n)]^\top$. In TDC learning, the goal is to evaluate the value function under linear function approximation via minimizing the following mean square projected Bellman error (MSPBE).

$$\text{MSPBE}(\theta) := \mathbb{E}_{\mu_b}\|\widehat{V}_\theta - \Pi_{R_\theta}T^\pi\widehat{V}_\theta\|^2,$$

where $\Pi_{R_\theta}$ is the projection operator to the Euclidean ball with radius $R_\theta$.

In the off-policy TDC learning, we sample a trajectory of the MDP induced by the behavior policy $\pi_b$ and obtain samples $\{s_0, a_0, r_0, s_1, \ldots, s_t, a_t, r_t, s_{t+1}, ...\}$. For the $t$-th sample $x_t = (s_t, a_t, r_t, s_{t+1})$, we define the following parameters

$$A_t := \rho(s_t, a_t)\phi(s_t)(\gamma\phi(s_{t+1}) - \phi(s_t))^\top, \quad b_t := r_t\rho(s_t, a_t)\phi(s_t), \tag{1}$$
$$B_t := -\gamma\rho(s_t, a_t)\phi(s_{t+1})\phi(s_t)^\top, \quad C_t := -\phi(s_t)\phi(s_t)^\top,$$

where $\rho(s, a) := \frac{\pi(a|s)}{\pi_b(a|s)}$ is the importance sampling ratio. Then, with learning rates $\alpha, \beta > 0$ and initialization parameters $\theta_0, w_0$, the two time-scale off-policy TDC algorithm takes the following recursive updates for $t = 0, 1, 2, ...$

$$\text{(Off-Policy TDC):} \begin{cases} \theta_{t+1} = \theta_t + \alpha(A_t\theta_t + b_t + B_tw_t), \\ w_{t+1} = w_t + \beta(A_t\theta_t + b_t + C_tw_t). \end{cases} \tag{2}$$

Also, for an arbitrary sample $(s, a, r, s')$, we define the following expectation terms for convenience of the analysis: $A := \mathbb{E}_{\mu_{\pi_b}}[\rho(s, a)\phi(s)(\gamma\phi(s') - \phi(s))^\top]$, $B := -\gamma\mathbb{E}_{\mu_{\pi_b}}[\rho(s, a)\phi(s')\phi(s)^\top]$, $C := -\mathbb{E}_{\mu_{\pi_b}}[\phi(s)\phi(s)^\top]$ and $b := \mathbb{E}_{\mu_{\pi_b}}[r\rho(s, a)\phi(s)]$.

With these notations, we introduce the following standard assumptions for our analysis [28].

**Assumption 2.1** (Problem solvability). *The matrix $A$ and $C$ are non-singular.*

**Assumption 2.2** (Boundedness). *For all states $s, s' \in \mathcal{S}$ and all actions $a \in \mathcal{A}$,*

1. *The feature function is uniformly bounded as $\|\phi(s)\| \leq 1$;*
2. *The reward is uniformly bounded as $r(s, a, s') \leq r_{\max}$;*
3. *The importance sampling ratio is uniformly bounded as $\rho(s, a) \leq \rho_{\max}$.*

**Assumption 2.3** (Geometric ergodicity). *There exists $\kappa > 0$ and $\rho \in (0, 1)$ such that for all $t \geq 0$,*

$$\sup_{s \in \mathcal{S}} d_{TV}\left(\mathbb{P}(s_t \in \cdot | s_0 = s), \mu_{\pi_b}\right) \leq \kappa \rho^t,$$

*where $d_{TV}(P, Q)$ denotes the total-variation distance between the probability measures $P$ and $Q$.*

Under Assumption 2.1, the optimal parameter $\theta^*$ can be written as $\theta^* = -A^{-1}b$.

## 3 Variance-Reduced TDC for I.I.D. Samples

In this section, we propose a variance reduction scheme for the off-policy TDC over i.i.d. samples and analyze its non-asymptotic convergence rate.

---

**Algorithm 1:** Variance-Reduced TDC for I.I.D. Samples

---

**Input:** learning rates $\alpha, \beta$, batch size $M$, initial parameters $\tilde{\theta}^{(0)}, \tilde{w}^{(0)}$.
**for** $m = 1, 2, \ldots$ **do**

    Initialize $\theta_0^{(m)} = \tilde{\theta}^{(m-1)}, w_0^{(m)} = \tilde{w}^{(m-1)}$.
    Query a set $B_m$ of $M$ independent samples from $\mu_{\pi_b}$ and compute

$$\widetilde{G}^{(m)} = \frac{1}{M}\sum_{x \in B_m}\left(A_x\tilde{\theta}^{(m-1)} + b_x + B_x\tilde{w}^{(m-1)}\right), \quad \widetilde{H}^{(m)} = \frac{1}{M}\sum_{x \in B_m}\left(A_x\tilde{\theta}^{(m-1)} + b_x + C_x\tilde{w}^{(m-1)}\right).$$

    **for** $t = 0, 1, \ldots, M - 1$ **do**

        Query a new sample $x_t^{(m)}$ from $\mu_{\pi_b}$ and compute

$$\theta_{t+1}^{(m)} = \Pi_{R_\theta}\left[\theta_t^{(m)} + \alpha\left(G_t^{(m)}(\theta_t^{(m)}, w_t^{(m)}) - G_t^{(m)}(\tilde{\theta}^{(m-1)}, \tilde{w}^{(m-1)}) + \widetilde{G}^{(m)}\right)\right],$$

$$w_{t+1}^{(m)} = \Pi_{R_w}\left[w_t^{(m)} + \beta\left(H_t^{(m)}(\theta_t^{(m)}, w_t^{(m)}) - H_t^{(m)}(\tilde{\theta}^{(m-1)}, \tilde{w}^{(m-1)}) + \widetilde{H}^{(m)}\right)\right],$$

        where for any $\theta, w$,
        $G_t^{(m)}(\theta, w) = A_t^{(m)}\theta + b_t^{(m)} + B_t^{(m)}w, \quad H_t^{(m)}(\theta, w) = A_t^{(m)}\theta + b_t^{(m)} + C_t^{(m)}w.$
    **end**

    Set $\tilde{\theta}^{(m)} = \frac{1}{M}\sum_{t=0}^{M-1}\theta_t^{(m-1)}, \quad \tilde{w}^{(m)} = \frac{1}{M}\sum_{t=0}^{M-1}w_t^{(m-1)}$.
**end**
**Output:** $\tilde{\theta}^{(m)}$.

---

### 3.1 Algorithm Design

In the i.i.d. setting, we assume that one can query independent samples $x = (s, a, r, s')$ from the stationary distribution $\mu_{\pi_b}$ induced by the behavior policy $\pi_b$. In particular, we define $A_x, B_x, C_x, b_x$ based on the sample $x$ and define $A_t^{(m)}, B_t^{(m)}, C_t^{(m)}, b_t^{(m)}$ based on the sample $x_t^{(m)}$ in a similar way as how we define $A_t, B_t, C_t, b_t$ based on the sample $x_t$ in eq. (1).

We then propose the variance-reduced TDC algorithm for i.i.d. samples in Algorithm 1. To elaborate, the algorithm runs for $m$ outer-loops, each of which consists of $M$ inner-loops. Specifically, in the $m$-th outer-loop, we first initialize the parameters $\theta_0^{(m)}, w_0^{(m)}$ with $\tilde{\theta}^{(m-1)}, \tilde{w}^{(m-1)}$, respectively, which are the output of the previous outer-loop. Then, we query $M$ independent samples from $\mu_{\pi_b}$ and compute a pair of batch pseudo-gradients $\widetilde{G}^{(m)}, \widetilde{H}^{(m)}$ to be used in the inner-loops. In the $t$-th inner-loop, we query a new independent sample $x_t^{(m)}$ and compute the corresponding stochastic pseudo-gradients $G_t^{(m)}(\theta_t^{(m)}, w_t^{(m)}), G_t^{(m)}(\tilde{\theta}^{(m-1)}, \tilde{w}^{(m-1)})$. Then, we update the parameters $\theta_{t+1}^{(m)}$

and $w_{t+1}^{(m)}$ using the batch pseudo-gradient and stochastic pseudo-gradients via the SVRG variance reduction scheme. At the end of each outer-loop, we set the parameters $\tilde{\theta}^{(m)}, \tilde{w}^{(m)}$ to be the average of the parameters $\{\theta_t^{(m-1)}, w_t^{(m-1)}\}_{t=0}^{M-1}$ obtained in the inner-loops, respectively. We note that the updates of $\theta_{t+1}^{(m)}$ and $w_{t+1}^{(m)}$ involve two projection operators, which are widely adopted in the literature, e.g., [4, 6–8, 15, 16, 29, 30]. Throughout this paper, we assume the radius $R_\theta, R_w$ of the projected Euclidean balls satisfy that $R_\theta \geq \max\{\|A\|\|b\|, \|\theta^*\|\}$, $R_w \geq 2\|C^{-1}\|\|A\|R_\theta$.

Compare to the conventional variance-reduced TD for i.i.d. samples that applies variance reduction to only the one time-scale update of $\theta_t^{(m)}$ [14, 28], our VRTDC for i.i.d. samples applies variance reduction to both $\theta_t^{(m)}$ and $w_t^{(m)}$ that are in two different time-scales. As we show in the following subsection, such a two time-scale variance reduction scheme leads to an improved sample complexity of VRTDC over that of VRTD [28].

## 3.2 Non-Asymptotic Convergence Analysis

The following theorem presents the convergence result of VRTDC for i.i.d. samples. Due to space limitation, we omit other constant factors in the bound. The exact bound can be found in Appendix B.

**Theorem 3.1.** *Let Assumptions 2.1, 2.2 and 2.3 hold. Connsider the VRTDC for i.i.d. samples in Algorithm 1. If the learning rates $\alpha$, $\beta$ and the batch size $M$ satisfy the conditions specified in eqs.* (3) *to* (9) *(see the appendix) and $\beta = \mathcal{O}(\alpha^{\frac{2}{3}})$, then, the output of the algorithm satisfies*

$$\mathbb{E}\|\tilde{\theta}^{(m)} - \theta^*\|^2 \leq \mathcal{O}(mD^m + \beta^4 + \beta M^{-1}),$$

*where $D = \frac{12}{\lambda_{\widehat{A}}}\left\{\frac{1}{\alpha M} + \alpha \cdot 5(1+\gamma)^2 \rho_{\max}^2\left(1 + \frac{\gamma\rho_{\max}}{\min|\lambda(C)|}\right)^2 + \frac{\alpha^2}{\beta^2}\cdot C_1 + \beta\cdot C_2\right\} \in (0, 1)$ (see Appendix A for the definitions of $\lambda_{\widehat{A}}, C_1, C_2$). In particular, choose $\alpha = \mathcal{O}(\epsilon^{\frac{3}{5}}), \beta = \mathcal{O}(\epsilon^{\frac{2}{5}}), m = \mathcal{O}(\log \epsilon^{-1})$ and $M = \mathcal{O}(\epsilon^{-\frac{3}{5}})$, then the total sample complexity to achieve $\mathbb{E}\|\tilde{\theta}^{(m)} - \theta^*\|^2 \leq \epsilon$ is in the order of $\mathcal{O}(\epsilon^{-\frac{3}{5}}\log \epsilon^{-1})$.*

Theorem 3.1 shows that in the i.i.d. setting, VRTDC with constant stepsizes converges to a neighborhood of the optimal solution $\theta^*$ at a linear convergence rate. In particular, the asymptotic convergence error is in the order of $\mathcal{O}(\beta^4 + \beta M^{-1})$, which can be driven arbitrarily close to zero by choosing a sufficiently small stepsize $\beta$. Also, the required sample complexity for VRTDC to achieve an $\epsilon$-accurate solution is $\mathcal{O}(\epsilon^{-\frac{3}{5}}\log \epsilon^{-1})$, which is lower than the sample complexity $\mathcal{O}(\epsilon^{-1}\log \epsilon^{-1})$ (by a factor of $\epsilon^{-\frac{2}{5}}$) required by the conventional VRTD for i.i.d. samples [28].

Our analysis of VRTDC in the i.i.d. setting requires substantial developments of new bounding techniques. To elaborate, note that in the analysis of the one time-scale VRTD [28], they only need to deal with the parameter $\tilde{\theta}^{(m)}$ and bound its variance error using constant-level bounds. As a comparison, in the analysis of VRTDC we need to develop much refined variance reduction bounds for both $\tilde{\theta}^{(m)}$ and $\tilde{w}^{(m)}$ that are correlated with each other. Specifically, we first develop a preliminary bound for the tracking error $\|\tilde{z}^{(m)}\|^2 = \|\tilde{w}^{(m)} + C^{-1}(b + A(\tilde{\theta}^{(m)}))\|^2$ that is in the order of $\mathcal{O}(D^m + \beta)$ (Lemma D.2), which is further used to develop a preliminary bound for the convergence error $\|\theta^{(m)} - \theta^*\|^2 \leq \mathcal{O}(D^m + M^{-1})$ (Lemma D.4). Then, by leveraging the relation between tracking error and convergence error induced by the two time-scale updates of VRTDC (Lemma D.3), we further apply the preliminary bound of $\|\theta^{(m)} - \theta^*\|^2$ to obtain a refined bound for the tracking error $\|z^{(m)}\|^2 \leq \mathcal{O}(D^m + \beta^3 + \beta M^{-1})$ (Lemma D.5). Finally, the refined bound of $\|z^{(m)}\|^2$ is applied to derive a refined bound for the convergence error $\|\theta^{(m)} - \theta^*\|^2 \leq \mathcal{O}(D^m + \beta^4 + \beta M^{-1})$ by leveraging the two time-scale updates (Lemma D.1). These refined bounds are the key to establish the improved sample complexity of VRTDC for i.i.d. samples over the state-of-the-art result.

We also obtain the following convergence rate of the tracking error of VRTDC in the i.i.d. setting.

**Corollary 3.2.** *Under the same settings and parameter choices as those of Theorem 3.1, the tracking error of VRTDC for i.i.d. samples satisfies*

$$\mathbb{E}\|\tilde{z}^{(m)}\|^2 \leq \mathcal{O}(D^m + \beta^3 + \beta M^{-1}).$$

*Moreover, the total sample complexity to achieve $\mathbb{E}\|\tilde{z}^{(m)}\|^2 \leq \epsilon$ is in the order of $\mathcal{O}(\epsilon^{-\frac{1}{2}}\log \epsilon^{-1})$.*

# 4 Variance-Reduced TDC for Markovian Samples

In this section, we propose a variance-reduced TDC algorithm for Markovian samples and characterize its non-asymptotic convergence rate.

---

**Algorithm 2:** TDC with Variance Reduction for Markovian Samples

---

**Input:** learning rates $\alpha, \beta$, batch size $M$, initial parameters $\tilde{\theta}^{(0)}, \tilde{w}^{(0)}$,
  Markovian samples of MDP $\{x_1, x_2, x_3, ...\}$.

**for** $m = 1, 2, \ldots$ **do**

  Initialize $\theta_0^{(m)} = \tilde{\theta}^{(m-1)}$, $w_0^{(m)} = \tilde{w}^{(m-1)}$.
  Query the set of samples $B_m = \{x_{(m-1)M}, ..., x_{mM-1}\}$ and compute

$$\widetilde{G}^{(m)} = \frac{1}{M}\sum_{x \in B_m}\big(A_x\tilde{\theta}^{(m-1)} + b_x + B_x\tilde{w}^{(m-1)}\big), \quad \widetilde{H}^{(m)} = \frac{1}{M}\sum_{x \in B_m}\big(A_x\tilde{\theta}^{(m-1)} + b_x + C_x\tilde{w}^{(m-1)}\big).$$

  **for** $t = 0, 1, \ldots, M-1$ **do**

    Query a sample $x_t^{(m)}$ from $B_m$ uniformly at random and update

$$\theta_{t+1}^{(m)} = \Pi_{R_\theta}\Big[\theta_t^{(m)} + \alpha\big(G_t^{(m)}(\theta_t^{(m)}, w_t^{(m)}) - G_t^{(m)}(\tilde{\theta}^{(m-1)}, \tilde{w}^{(m-1)}) + \widetilde{G}^{(m)}\big)\Big],$$

$$w_{t+1}^{(m)} = \Pi_{R_w}\Big[w_t^{(m)} + \beta\big(H_t^{(m)}(\theta_t^{(m)}, w_t^{(m)}) - H_t^{(m)}(\tilde{\theta}^{(m-1)}, \tilde{w}^{(m-1)}) + \widetilde{H}^{(m)}\big)\Big],$$

    where for any $\theta, w$, we define

$$G_t^{(m)}(\theta, w) = A_t^{(m)}\theta + b_t^{(m)} + B_t^{(m)}w, \quad H_t^{(m)}(\theta, w) = A_t^{(m)}\theta + b_t^{(m)} + C_t^{(m)}w.$$

  **end**

  Set $\tilde{\theta}^{(m)} = \frac{1}{M}\sum_{t=0}^{M-1}\theta_t^{(m-1)}$, $\tilde{w}^{(m)} = \frac{1}{M}\sum_{t=0}^{M-1}w_t^{(m-1)}$.

**end**

**Output:** $\tilde{\theta}^{(m)}$.

---

## 4.1 Algorithm Design

In the Markovian setting, we generate a single trajectory of the MDP and obtain a series of Markovian samples $\{x_1, x_2, x_3, ...\}$, where the $t$-th sample is $x_t = (s_t, a_t, r_t, s_{t+1})$. The detailed steps of variance-reduced TDC for Markovian samples are presented in Algorithm 2. To elaborate, in the Markovian case, we divide the samples of the Markovian trajectory into $m$ batches. In the $m$-th outer-loop, we query the $m$-th batch of samples to compute the batch pseudo-gradients $\widetilde{G}^{(m)}, \widetilde{H}^{(m)}$. Then, in each of the corresponding inner-loops we query a sample from the same $m$-th batch of samples and perform variance-reduced updates on the parameters $\theta_t^{(m)}, w_t^{(m)}$. As a comparison, in our design of VRTDC for i.i.d. samples, the samples used in both the outer-loops and the inner-loops are independently drawn from the stationary distribution.

## 4.2 Non-Asymptotic Convergence Analysis

In the following theorem, we present the convergence result of VRTDC for Markovian samples. Due to space limitation, we omit the constants in the bound. The exact bound can be found in Appendix F.

**Theorem 4.1.** *Let Assumptions 2.1, 2.2 and 2.3 hold and consider the VRTDC in Algorithm 2 for Markovian samples. Choose learning rates $\alpha$, $\beta$ and the batch size $M$ that satisfy the conditions specified in eqs. (35) to (40) (see the appendix) and $\beta = \mathcal{O}(\alpha^{\frac{2}{3}})$. Then, the output of the algorithm satisfies*

$$\mathbb{E}\|\tilde{\theta}^{(m)} - \theta^*\|^2 \leq \mathcal{O}(D^m + M^{-1} + \beta^2),$$

*where $D = \frac{16}{\lambda_{\widehat{A}}}\Big[\frac{1}{\alpha M} + \alpha \cdot 5(1+\gamma)^2\rho_{\max}^2\big(1 + \frac{\gamma\rho_{\max}}{\min|\lambda(C)|}\big)^2 + \frac{\alpha^2}{\beta^2}\cdot C_1 + \beta \cdot C_2\Big] \in (0, 1)$ (see Appendix A for the definitions of $\lambda_{\widehat{A}}, C_1, C_2$). In particular, choose $\alpha = \mathcal{O}(\epsilon^{\frac{3}{4}}), \beta = \mathcal{O}(\epsilon^{\frac{1}{2}})$, $m = \mathcal{O}(\log \epsilon^{-1})$ and $M = \mathcal{O}(\epsilon^{-1})$, the total sample complexity to achieve $\mathbb{E}\|\tilde{\theta}^{(m)} - \theta^*\|^2 \leq \epsilon$ is in the order of $\mathcal{O}(\epsilon^{-1}\log \epsilon^{-1})$.*

The above theorem shows that in the Markovian setting, VRTDC with constant learning rates converges linearly to a neighborhood of $\theta^*$ with an asymptotic convergence error $\mathcal{O}(M^{-1} + \beta^2)$. As expected, such an asymptotic convergence error is larger than that of VRTDC for i.i.d. samples established in Theorem 3.1. We also note that the overall sample complexity of VRTDC for Markovian samples is in the order of $\mathcal{O}(\epsilon^{-1} \log \epsilon^{-1})$, which matches that of TDC for Markovian samples [13] and VRTD for Markovian samples [28]. Such a complexity result nearly matches the theoretical lower bound $\mathcal{O}(\epsilon^{-1})$ given in [13]. Moreover, as we show later in the experiments, our VRTDC always achieves a smaller asymptotic convergence error than that of TDC and VRTD in practice.

We note that in the Markovian case, one can also develop refined error bounds following the recursive refinement strategy used in the proof of Theorem 3.1. However, the proof of Theorem 4.1 only applies the preliminary bounds for the tracking error and the convergence error, which suffices to obtain the above desired near-optimal sample complexity result. This is because the error terms in Theorem 4.1 are dominated by $\frac{1}{M}$ and hence applying the refined bounds does not lead to a better sample complexity result.

We also obtain the following convergence rate of the tracking error of VRTDC in the Markovian setting, where the total sample complexity matches that of the TDC for Markovian samples [13, 29].

**Corollary 4.2.** *Under the same settings and parameter choices as those of Theorem 4.1, the tracking error of VRTDC for Markovian samples satisfies*

$$\mathbb{E}\|\tilde{z}^{(m)}\|^2 \leq \mathcal{O}(D^m + \beta^3 + M^{-1}).$$

*Moreover, the total sample complexity to achieve $\mathbb{E}\|\tilde{z}^{(m)}\|^2 \leq \epsilon$ is in the order of $\mathcal{O}(\epsilon^{-1} \log \epsilon^{-1})$.*

## 5 Experiments

In this section, we conduct two reinforcement learning experiments, Garnet problem and Frozen Lake game, to explore the performance of VRTDC in the off-policy setting with Markovian samples, and compare it with TD, TDC, VRTD and VRTDC in the Markovian setting.

### 5.1 Garnet Problem

We first consider the Garnet problem [1, 29] that is specified as $\mathcal{G}(n_{\mathcal{S}}, n_{\mathcal{A}}, b, d)$, where $n_{\mathcal{S}}$ and $n_{\mathcal{A}}$ denote the cardinality of the state and action spaces, respectively, $b$ is referred to as the branching factor–the number of states that have strictly positive probability to be visited after an action is taken, and $d$ denotes the dimension of the features. We set $n_{\mathcal{S}} = 500$, $n_{\mathcal{A}} = 20$, $b = 50, d = 15$ and generate the features $\Phi \in \mathbb{R}^{n_{\mathcal{S}} \times d}$ via the uniform distribution on $[0, 1]$. We then normalize its rows to have unit norm. Then, we randomly generate a state-action transition kernel $\mathbf{P} \in \mathbb{R}^{n_{\mathcal{S}} \times n_{\mathcal{A}} \times n_{\mathcal{S}}}$ via the uniform distribution on $[0, 1]$ (with proper normalization). We set the behavior policy as the uniform policy, i.e., $\pi_b(a|s) = n_{\mathcal{A}}^{-1}$ for any $s$ and $a$, and we generate the target policy $\pi(\cdot|s)$ via the uniform distribution on $[0, 1]$ with proper normalization for every state $s$. The discount factor is set to be $\gamma = 0.95$. As the transition kernel and the features are known, we compute $\theta^*$ and use $\|\theta - \theta^*\|$ to evaluate the performance of all the algorithms.

We set the learning rate $\alpha = 0.1$ for all the four algorithms, and set the other learning rate $\beta = 0.02$ for both VRTDC and TDC. For VRTDC and VRTD, we set the batch size $M = 3000$. In Figure 1 (Left), we plot the convergence error as a function of number of pseudo stochastic gradient computations for all these algorithms. Specifically, we use 100 Garnet trajectories with length 50k to obtain 100 convergence error curves for each algorithm. The upper and lower envelopes of the curves correspond to the $95\%$ and $5\%$ percentiles of the 100 curves, respectively. It can be seen that both VRTD and VRTDC outperform TD and TDC and asymptotically achieve smaller mean convergence errors with reduced numerical variances of the curves. This demonstrates the advantage of applying variance reduction to policy evaluation. Furthermore, comparing VRTDC with VRTD, one can observe that VRTDC achieves a smaller mean convergence error than that achieved by VRTD, and the numerical variance of the VRTDC curves is smaller than that of the VRTD curves. This demonstrates the effectiveness of applying variance reduction to two time-scale updates.

We further compare the asymptotic error of VRTDC and VRTD under different batch sizes $M$. We use the same learning rate setting as mentioned above and run 100k iterations for each of the 250 Garnet trajectories. For each trajectory, we use the mean of the convergence error of the last 10k iterations as an estimate of the asymptotic convergence error (the training curves are already

flattened). Figure 1 (Right) shows the box plot of the 250 samples of the asymptotic convergence error of VRTDC and VRTD under different batch sizes $M$. It can be seen that the two time-scale VRTDC achieves a smaller mean asymptotic convergence error with a smaller numerical variance than that achieved by the one time-scale VRTD.

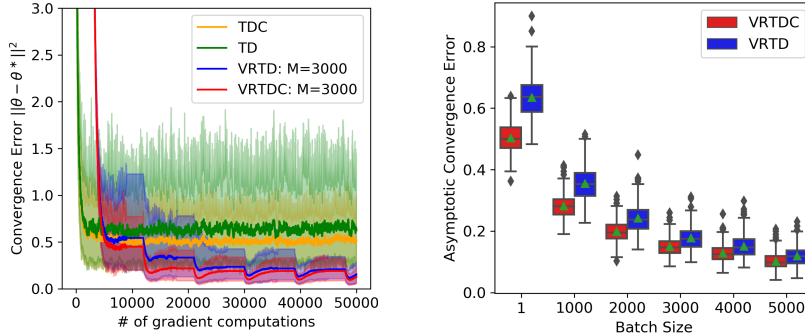

Figure 1: Comparison of TD, TDC, VRTD, VRTDC in solving the Garnet problem.

We further study the variance reduction effect of VRTDC. We plot the estimated variance of the stochastic updates of $\theta$ (see Figure 2 left) and $w$ (see Figure 2 right) for different algorithms. It can be seen that VRTDC significantly reduces the variance of TDC in both time-scales. For each step, we estimate the variance of each pseudo-gradient update using Monte-Carlo method with 500 additional samples under the same learning rates and batch size setting as mentioned previously.

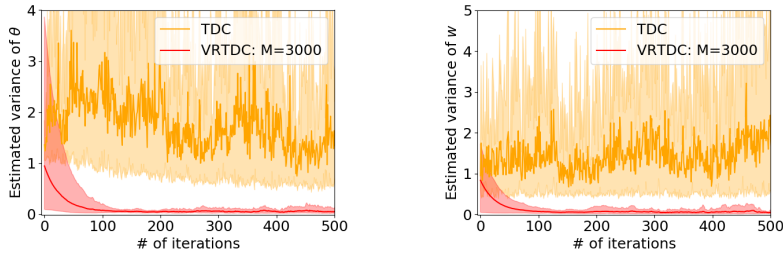

Figure 2: Comparison of TDC, VRTDC in solving the Garnet problem.

## 5.2 Frozen Lake Game

Our second experiment considers the frozen lake game in the OpenAI Gym [5]. We generate a Gaussian feature matrix with dimension 4 to linearly approximate the value function and we aim to evaluate a target policy based on a behavior policy, generated via the uniform distribution. We set the learning rates $\alpha = 0.1$ and $\beta = 0.01$ for all the algorithms and set the batch size $M = 3000$ for the variance-reduced algorithms. We run 50k iterations for each of the 100 trajectories. Figure 3 (Left) plots the convergence error as a function of number of gradient computations for the four algorithms using 5% and 95% percentiles of the 100 curves. One can see that our VRTDC asymptotically achieves the smallest mean convergence error with the smallest numerical variance of the curves. In particular, one can see that TDC achieves a comparable asymptotic error to that of VRTD, and VRTDC outperforms both of the algorithms. Figure 3 (Right) further compares the asymptotic convergence error of VRTDC with that of VRTD under different batch sizes $M$. Similar to the Garnet experiment, for each of the 100 trajectories we use the mean of the convergence errors of the last 10k iterations as an estimate of the asymptotic error, and the boxes include the samples between 25% percentile and 75% percentile. One can see from the figure that VRTDC achieves smaller asymptotic errors with smaller numerical variances than VRTD under all choices of batch size.

Similar to Figure 2, we also estimate the variance of each pseudo-gradient update using 500 Monte Carlo samples for the Frozen Lake problem. We can find that VRTDC also has lower variance for the Frozen Lake problem.

## 6 Conclusion

In this paper, we proposed two variance-reduced off-policy TDC algorithms for policy evaluation with i.i.d. samples and Markovian samples, respectively. We developed new analysis techniques

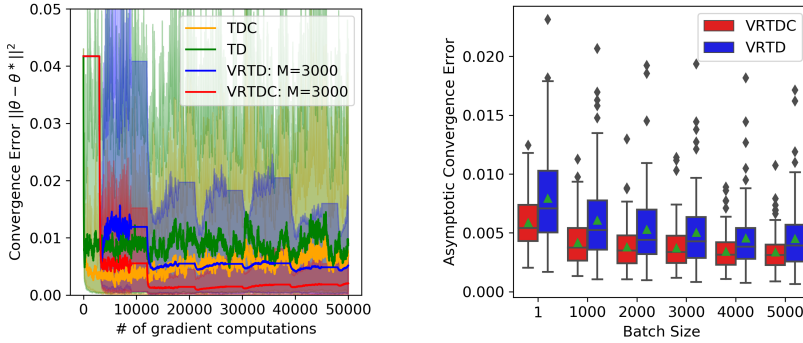

Figure 3: Comparison of TD, TDC, VRTD, VRTDC in solving the frozen lake problem.

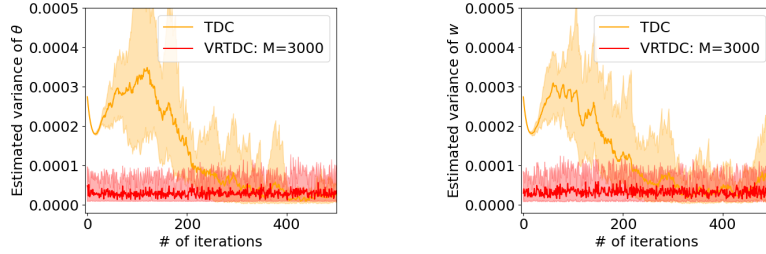

Figure 4: Comparison of TDC, VRTDC in solving the frozen lake problem.

and showed that VRTDC for i.i.d. samples achieves an improved sample complexity over the best existing result, and VRTDC for Markovian samples achieves the best existing sample complexity. We expect that the developed VRTDC algorithm can help reduce the stochastic variance in reinforcement learning applications and improve the solution accuracy.

## Acknowledgement

The work of S. Zou was supported by the National Science Foundation under Grant CCF-2007783.

## Broader Impact

This work exploits techniques in multidisciplinary areas including reinforcement learning, stochastic optimization and statistics, and contributes new technical developments to analyze TD learning algorithm under stochastic variance reduction. The proposed two time-scale VRTDC significantly improves the solution quality of TD learning and reduces the variance and uncertainty in training reinforcement learning policies. Therefore it has the potential to be applied to reinforcement learning applications such as autonomous driving, decision making and control to reduce the risk caused by uncertainty of the policy.

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
