[Supplementary Material · supplementary.pdf]

# Appendix

## Table of Contents

## A　Filtration, Additional Notations and List of Constants

### Filtration for I.I.D. samples

The definition of filtration is similar as that in VRTD (Appendix D, [27]). Recall that in Algorithm 1, $B_m$ consists of $M$ independent samples that are sampled from $\mu_{\pi_b}$, and $x_t^{(m)}$ is another independent sample sampled in the $t$-th iteration of the $m$-th epoch. Let $\sigma(A \cup B)$ be the smallest $\sigma$-field that includes both $A$ and $B$. Then we define the filtration for I.I.D. samples as follow

$$F_{1,0} = \sigma(\tilde{\theta}^{(0)}, \tilde{w}^{(0)}), F_{1,1} = \sigma(F_{1,0} \cup \sigma(B_1) \cup \sigma(x_0^{(1)})), \dots, F_{1,M} = \sigma(F_{1,M-1} \cup \sigma(x_{M-1}^{(1)}))$$

$$F_{2,0} = \sigma\big(F_{1,M} \cup \sigma(\tilde{\theta}^{(1)}, \tilde{w}^{(1)})\big), F_{2,1} = \sigma(F_{2,0} \cup \sigma(B_2) \cup \sigma(x_0^{(2)})), \dots, F_{2,M} = \sigma(F_{2,M-1} \cup \sigma(x_{M-1}^{(2)}))$$

$$\vdots$$

$$F_{m,0} = \sigma\big(F_{m-1,M} \cup \sigma(\tilde{\theta}^{(m-1)}, \tilde{w}^{(m-1)})\big), F_{m,1} = \sigma(F_{m,0} \cup \sigma(B_m) \cup \sigma(x_0^{(m)})), \dots,$$

$$F_{m,M} = \sigma(F_{m,M-1} \cup \sigma(x_{M-1}^{(m)})).$$

Moreover, we define $\mathbb{E}_{t,m}$ as the conditional expectation with respect to the $\sigma$-field $F_{t,m}$.

### Filtration for Markovian samples

The definition of filtration is similar as that in VRTD (Appendix E, [27]). We first recall that $B_m$ denotes the set of Markovian samples used in the $m$-th epoch, and we also abuse the notation here by letting $x_t^{(m)}$ be the sample picked in the $t$-th iteration of the $m$-th epoch. Then, we define the filtration for Markovian samples as follows

$$F_{1,0} = \sigma(B_0 \cup \sigma(\tilde{\theta}^{(0)}, \tilde{w}^{(0)})), F_{1,1} = \sigma(F_{1,0} \cup \sigma(x_0^{(1)})), \dots, F_{1,M} = \sigma(F_{1,M-1} \cup \sigma(x_{M-1}^{(1)}))$$

$$F_{2,0} = \sigma\big(B_1 \cup F_{1,M} \cup \sigma(\tilde{\theta}^{(1)}, \tilde{w}^{(1)})\big), F_{2,1} = \sigma(F_{2,0} \cup \sigma(x_0^{(2)})), \dots, F_{2,M} = \sigma(F_{2,M-1} \cup \sigma(x_{M-1}^{(2)}))$$

$$\vdots$$

$$F_{m,0} = \sigma\big(B_{m-1} \cup F_{m-1,M} \cup \sigma(\tilde{\theta}^{(m-1)}, \tilde{w}^{(m-1)})\big), F_{m,1} = \sigma(F_{m,0} \cup \sigma(x_0^{(m)})), \dots,$$

$$F_{m,M} = \sigma(F_{m,M-1} \cup \sigma(x_{M-1}^{(m)})).$$

407 Moreover, we define $\mathbb{E}_{t,m}$ as the conditional expectation with respect to the $\sigma$-field $F_{t,m}$.

### Additional Notations

409 Recall the one-step TDC update at $\theta_t^{(m)}$:

$$A_t^{(m)}\theta + b_t^{(m)} + B_t^{(m)}w = A_t^{(m)}\theta + b_t^{(m)} + B_t^{(m)}z + B_t^{(m)}\left(-C^{-1}(b + A\theta)\right)$$
$$= \left(A_t^{(m)} - B_t^{(m)}C^{-1}A\right)\theta + b_t^{(m)} - B_t^{(m)}C^{-1}b + B_t^{(m)}z.$$

410 Define $\widehat{A}_t^{(m)} := A_t^{(m)} - B_t^{(m)}C^{-1}A$ and $\hat{b}_t^{(m)} := b_t^{(m)} - B_t^{(m)}C^{-1}b$. Then, we further define

$$G_t^{(m)}(\theta, z) := \widehat{A}_t^{(m)}\theta + \hat{b}_t^{(m)} + B_t^{(m)}z.$$

411 Moreover, we define

$$\lambda_{\widehat{A}} := -\lambda_{\max}(\widehat{A} + \widehat{A}^\top) = -\lambda_{\max}(2A^\top C^{-1}A) < 0.$$

412 Similarly, recall the one-step TDC update at $w_t^{(m)}$:

$$A_t^{(m)}\theta + b_t^{(m)} + C_t^{(m)}w = A_t^{(m)}\theta + b_t^{(m)} + C_t^{(m)}z + C_t^{(m)}\left(-C^{-1}(b + A\theta)\right)$$
$$= \left(A_t^{(m)} - C_t^{(m)}C^{-1}A\right)\theta + b_t^{(m)} - C_t^{(m)}C^{-1}b + C_t^{(m)}z.$$

413 Define $\bar{A}_t^{(m)} := A_t^{(m)} - C_t^{(m)}C^{-1}A$ and $\bar{b}_t^{(m)} := b_t^{(m)} - C_t^{(m)}C^{-1}b$. Then, we further define

$$H_t^{(m)}(\theta, z) := \bar{A}_t^{(m)}\theta + \bar{b}_t^{(m)} + C_t^{(m)}z.$$

Moreover, we define

$$\lambda_C := -\lambda_{\max}(C + C^\top) = -\lambda_{\max}(2C) < 0.$$

### List of Constants

415 We summerize all the constants that are used in the proof as follows.

### Constants for both i.i.d. and Markovian setting:

417
- $G_{\text{VR}} := 3\left[(1 + \gamma)R_\theta + r_{\max}\right]\rho_{\max}\left(1 + \frac{\gamma\rho_{\max}}{\min|\lambda(C)|}\right).$

418
- $H_{\text{VR}} := 3\left[(1 + \gamma)R_\theta + r_{\max}\right]\rho_{\max}\left(1 + \frac{1}{\min|\lambda(C)|}\right).$

### Constants for i.i.d. setting:

420
- $K_1 = \left[(1 + \gamma)R_\theta + r_{\max}\right]^2 \rho_{\max}^2\left(1 + \frac{\gamma\rho_{\max}}{\min|\lambda(C)|}\right)^2,$

421
- $K_2 = \left[(1 + \gamma)R_\theta + r_{\max}\right]^2\left(1 + \frac{1}{\min|\lambda_C|}\right)^2,$

422
- $C_1 = \frac{2\rho_{\max}^2\gamma^2}{\lambda_{\widehat{A}}} \cdot \frac{3}{\lambda_C} \cdot 10(1 + \gamma)^2\rho_{\max}^2 \cdot \left(1 + \frac{\gamma\rho_{\max}}{\min|\lambda(C)|}\right)^2\left(1 + \frac{2}{\lambda_C}\right) \cdot \left(\rho_{\max}\frac{1+\gamma}{\min|\lambda(C)|}\right)^2,$

423
- $C_2 = \frac{2\rho_{\max}^2\gamma^2}{\lambda_{\widehat{A}}} \cdot \frac{3}{\lambda_C} \cdot 10(1 + \gamma)^2\rho_{\max}^2 \cdot \left(1 + \frac{1}{\min|\lambda(C)|}\right)^2.$

424
- $C_3 = 10(1 + \gamma)^2\rho_{\max}^2 \cdot \left(1 + \frac{\gamma\rho_{\max}}{\min|\lambda(C)|}\right)^2\left(1 + \frac{2}{\lambda_C}\right) \cdot \left(\rho_{\max}\frac{1+\gamma}{\min|\lambda(C)|}\right)^2,$

425
- $C_4 = 10(1 + \gamma)^2\rho_{\max}^2 \cdot \left(1 + \frac{1}{\min|\lambda(C)|}\right)^2,$

426
- $D = \frac{12}{\lambda_{\widehat{A}}}\left\{\frac{1}{\alpha M} + \alpha \cdot 5(1 + \gamma)^2\rho_{\max}^2\left(1 + \frac{\gamma\rho_{\max}}{\min|\lambda(C)|}\right)^2 + \frac{\alpha^2}{\beta^2} \cdot C_1 + \beta \cdot C_2\right\},$

427
- $E = \frac{1}{M\beta} \cdot \frac{2}{\lambda_C},$

428
- $F = \frac{4}{\lambda_C}\left[\frac{1}{\beta M} + \beta \cdot 10 + \frac{\alpha^2}{\beta^2} \cdot 10\gamma^2\rho_{\max}^2\left(1 + \frac{2}{\lambda_C}\right) \cdot \left(\rho_{\max}\frac{1+\gamma}{\min|\lambda(C)|}\right)^2 + \left(\frac{\alpha^3}{\beta^2} \cdot C_3 + \alpha\beta \cdot\right.\right.$

429
$$\left.\left. C_4\right)\frac{30}{\lambda_{\widehat{A}}}\gamma^2\rho_{\max}^2\right].$$

**Constants for Markovian setting:**

- $K_1 := [(1+\gamma)R_\theta + r_{\max}]^2 \rho_{\max}^2 \left(1 + \frac{\gamma\rho_{\max}}{\min|\lambda(C)|}\right)^2 \cdot \left(1 + \kappa\frac{2\rho}{1-\rho}\right),$

- $K_2 := \frac{2}{\lambda_{\widehat{A}}} \left[R_\theta^2(1+\gamma)^2 + r_{\max}^2\right] \cdot 4\rho_{\max}^2 \left(1 + \frac{\gamma\rho_{\max}}{\min|\lambda(C)|}\right)^2 \left[1 + \kappa\frac{\rho}{1-\rho}\right],$

- $K_3 := \left(\frac{32}{\lambda_C}\left[R_\theta^2(1+\gamma)^2 + r_{\max}^2\right]\cdot\rho_{\max}^2 + \frac{16}{\lambda_C}\frac{\rho_{\max}(1+\gamma)R_\theta + \rho_{\max}r_{\max}}{\min|\lambda(C)|}\right)\left[1 + \kappa\frac{\rho}{1-\rho}\right],$

- $K_4 := \frac{12}{\lambda_C}R_w^2\left[1 + \kappa\frac{\rho}{1-\rho}\right],$

- $K_5 := [(1+\gamma)R_\theta + r_{\max}]^2 \rho_{\max}^2 \left(1 + \frac{1}{\min|\lambda(C)|}\right)^2 \cdot \left(1 + \kappa\frac{2\rho}{1-\rho}\right),$

- $C_1 = \left(1 + \frac{\gamma\rho_{\max}}{\min|\lambda(C)|}\right)^2\left(1 + \frac{2}{\lambda_C}\right) \cdot \left(\rho_{\max}\frac{1+\gamma}{\min|\lambda(C)|}\right)^2 \cdot \frac{96}{\lambda_{\widehat{A}}\lambda_C}\gamma^2\rho_{\max}^2 \cdot 10(1+\gamma)^2\rho_{\max}^2,$

- $C_2 = \left(1 + \frac{1}{\min|\lambda(C)|}\right)^2 \cdot \frac{96}{\lambda_{\widehat{A}}\lambda_C}\gamma^2\rho_{\max}^2 \cdot 10(1+\gamma)^2\rho_{\max}^2,$

- $D = \frac{16}{\lambda_{\widehat{A}}}\left[\frac{1}{\alpha M} + \alpha \cdot 5(1+\gamma)^2\rho_{\max}^2\left(1 + \frac{\gamma\rho_{\max}}{\min|\lambda(C)|}\right)^2 + \frac{\alpha^2}{\beta^2}\cdot C_1 + \beta \cdot C_2\right],$

- $E = \frac{1}{M\beta}\cdot\frac{12}{\lambda_C},$

- $F = \frac{24}{\lambda_C}\cdot\left[\frac{1}{\beta M} + 10\beta + 10\gamma^2\rho_{\max}^2\left(1 + \frac{1}{\lambda_C}\right)\cdot\left(\rho_{\max}\frac{1+\gamma}{\min|\lambda(C)|}\right)^2\frac{\alpha^2}{\beta^2} + \alpha\cdot120(1+\gamma)^2\rho_{\max}^2\frac{1}{\lambda_{\widehat{A}}}\cdot\right.$

  $\left.\left[\left(1 + \frac{\gamma\rho_{\max}}{\min|\lambda(C)|}\right)^2\left(1 + \frac{2}{\lambda_C}\right)\cdot\left(\rho_{\max}\frac{1+\gamma}{\min|\lambda(C)|}\right)^2\frac{\alpha^2}{\beta^2} + + \left(1 + \frac{1}{\min|\lambda(C)|}\right)^2\beta\right]\cdot 5\gamma^2\rho_{\max}^2\right].$

# B  Proof of Theorem 3.1

Throughout the proof, we assume the learning rates $\alpha, \beta$ and the batch size $M$ satisfy the following conditions.

$$\alpha \leq \min\left\{\frac{1}{5\lambda_{\widehat{A}}}, \frac{\lambda_{\widehat{A}}}{60}\Big/\left[(1+\gamma)^2\rho_{\max}^2\left(1 + \frac{\gamma\rho_{\max}}{\min|\lambda(C)|}\right)^2\right]\right\}, \tag{3}$$

$$\frac{\alpha^2}{\beta^2}\cdot C_3 + \beta \cdot C_4 \leq \min\left\{\frac{1-D}{144}\frac{\lambda_{\widehat{A}}^2\lambda_C}{\rho_{\max}^2\gamma^2}, 5(1-D), C_4\right\}, \tag{4}$$

$$M\beta > \frac{4}{\lambda_C}, \tag{5}$$

$$\frac{\lambda_C}{6}\beta - 10\beta^2 - 10\gamma^2\rho_{\max}^2\left(\alpha^2 + \frac{2\alpha^2}{\lambda_C}\frac{1}{\beta}\right)\cdot\left(\rho_{\max}\frac{1+\gamma}{\min|\lambda(C)|}\right)^2 \geq 0, \tag{6}$$

$$\frac{1}{\alpha M} + \alpha \cdot 5(1+\gamma)^2\rho_{\max}^2\left(1 + \frac{\gamma\rho_{\max}}{\min|\lambda(C)|}\right)^2 \leq \frac{\lambda_{\widehat{A}}}{6}, \tag{7}$$

$$\frac{\alpha}{\beta^2 M}\cdot\frac{72\rho_{\max}^2\gamma^2}{\lambda_{\widehat{A}}^2}\frac{1}{\lambda_C} + \beta\cdot\frac{720\rho_{\max}^2\gamma^2}{\lambda_{\widehat{A}}^2}\frac{1}{\lambda_C}$$
$$+ \frac{\alpha^2}{\beta^2}\cdot\frac{720\rho_{\max}^2\gamma^4}{\lambda_{\widehat{A}}^2}\frac{1}{\lambda_C}\rho_{\max}^2\left(1 + \frac{2}{\lambda_C}\right)\cdot\left(\rho_{\max}\frac{1+\gamma}{\min|\lambda(C)|}\right)^2 + \alpha\cdot\frac{60}{\lambda_{\widehat{A}}}\gamma^2\rho_{\max}^2 \leq 1, \tag{8}$$

$$\max\{D, E, F\} < 1, \tag{9}$$

where $C_3$ and $C_4$ are specified in eq.(24) and eq.(25), respectively, and $D, E, F$ are specified in eq.(18), eq.(19), and eq.(23), respectively. We note that under the above conditions, all the supporting lemmas for proving the theorem are satisfied. Also, we note that for a sufficiently small target accuracy $\epsilon$, our choices of learning rates and batch size $\alpha = \mathcal{O}(\epsilon^{\frac{3}{5}}), \beta = \mathcal{O}(\epsilon^{\frac{2}{5}}), M = \mathcal{O}(\epsilon^{-\frac{3}{5}})$ that are stated in the theorem satisfy the above conditions eqs. (3) to (9).

**Proof Sketch.** The proof consists of the following key steps.

1. Develop *preliminary bound for* $\sum_{t=0}^{M-1}\|\theta_t^{(m)} - \theta^*\|^2$ (Lemma D.1).

   We bound $\sum_{t=0}^{M-1}\|\theta_t^{(m)} - \theta^*\|^2$ in terms of $\sum_{t=0}^{M-1}\|z_t^{(m)}\|^2$, $\|\tilde{z}^{(m-1)}\|^2$, and $\|\tilde{\theta}^{(m-1)} - \theta^*\|^2$.

2. Develop *preliminary bound for* $\sum_{t=0}^{M-1}\|z_t^{(m)}\|^2$ (Lemma D.3).

   We bound $\sum_{t=0}^{M-1}\|z_t^{(m)}\|^2$ in terms of $\sum_{t=0}^{M-1}\|\theta_t^{(m)} - \theta^*\|^2$, $\|\tilde{z}^{(m-1)}\|^2$, and $\|\tilde{\theta}^{(m-1)} - \theta^*\|^2$, and plug it into the preliminary bound of $\sum_{t=0}^{M-1}\|\theta_t^{(m)} - \theta^*\|^2$. Then, we obtain an upper bound of $\sum_{t=0}^{M-1}\|\theta_t^{(m)} - \theta^*\|^2$ in terms of $\|\tilde{z}^{(m-1)}\|^2$, and $\|\tilde{\theta}^{(m-1)} - \theta^*\|^2$.

3. Develop *preliminary non-asymptotic bound for* $\|\tilde{z}^{(m)}\|^2$ (Lemma D.2).

   We develop a non-asymptotic bound for $\|\tilde{z}^{(m)}\|^2$.

4. Develop *preliminary non-asymptotic bound for* $\|\tilde{\theta}^m - \theta^*\|^2$ (Lemma D.4).

   We plug the bound in Lemma D.2 into the previous upper bounds. Then, we obtain an inequality between $\mathbb{E}\|\tilde{\theta}^{(m)} - \theta^*\|$ and $\mathbb{E}\|\tilde{\theta}^{(m-1)} - \theta^*\|$. Telescoping this inequality leads to the final result.

5. Develop *refined bound for* $\|\tilde{z}^{(m)}\|^2$ (Lemma D.5).

   We bound $\sum_{t=0}^{M-1}\|z_t^{(m)}\|^2$ in terms of $\sum_{t=0}^{M-1}\|\theta_t^{(m)} - \theta^*\|^2$, $\|\tilde{z}^{(m-1)}\|^2$, and $\|\tilde{\theta}^{(m-1)} - \theta^*\|^2$. Then, we apply Lemma D.1 and obtain an upper bound of $\sum_{t=0}^{M-1}\|z_t^{(m)}\|^2$ in terms of $\|\tilde{z}^{(m-1)}\|^2$ and $\|\tilde{\theta}^{(m-1)} - \theta^*\|^2$. Moreover, we apply Lemma D.2 and the preliminary non-asymptotic bound of $\|\tilde{\theta}^m - \theta^*\|^2$ to obtain an upper bound of $\sum_{t=0}^{M-1}\|z_t^{(m)}\|^2$ in terms of $\|\tilde{z}^{(m-1)}\|^2$. This gives the desired refined bound of $\|\tilde{z}^m\|^2$.

6. Develop *refined bound for* $\|\tilde{\theta}^m - \theta^*\|^2$ (Theorem 3.1).

   We use the refined bound of $\|\tilde{z}^m\|^2$ instead of the preliminary bound obtained in the step 4.

First, based on Lemma D.1, we have the following result

$$
\frac{\lambda_{\widehat{A}}}{6}\alpha \sum_{t=0}^{M-1} \mathbb{E}_{m,0}\|\theta_t^{(m)} - \theta^*\|^2
$$

$$
\leq \Big[1 + \alpha^2 M \cdot 5(1+\gamma)^2 \rho_{\max}^2 \big(1 + \frac{\gamma\rho_{\max}}{\min|\lambda(C)|}\big)^2\Big]\mathbb{E}_{m,0}\|\tilde{\theta}^{(m-1)} - \theta^*\|^2 + \alpha^2 \cdot 5K_1
$$

$$
+ \alpha \cdot \frac{2\rho_{\max}^2\gamma^2}{\lambda_{\widehat{A}}} \sum_{t=0}^{M-1} \mathbb{E}_{m,0}\|z_t^{(m)}\|^2 + \alpha^2 M \cdot 5\gamma^2\rho_{\max}^2\mathbb{E}_{m,0}\|\tilde{z}^{(m-1)}\|^2.
$$

Apply Lemma D.3 to bound the term $\sum_{t=0}^{M-1}\mathbb{E}_{m,0}\|z_t^{(m)}\|^2$ in the above inequality, re-arrange the obtained result and note that $\frac{\lambda_C}{6}\beta - 10\beta^2 - 10\gamma^2\rho_{\max}^2\big(\alpha^2 + \frac{2\alpha^2}{\lambda_C}\frac{1}{\beta}\big)\cdot\big(\rho_{\max}\frac{1+\gamma}{\min|\lambda(C)|}\big)^2 \geq 0$. Then, we obtain the following inequality,

$$
\frac{\lambda_{\widehat{A}}}{12}\alpha \sum_{t=0}^{M-1} \mathbb{E}_{m,0}\|\theta_t^{(m)} - \theta^*\|^2
$$

$$
\leq \Big\{1 + \alpha^2 M \cdot 5(1+\gamma)^2 \rho_{\max}^2 \big(1 + \frac{\gamma\rho_{\max}}{\min|\lambda(C)|}\big)^2 + \alpha M \cdot \big(\frac{\alpha^2}{\beta^2}\cdot C_1 + \beta \cdot C_2\big)\Big\}\mathbb{E}_{m,0}\|\tilde{\theta}^{(m-1)} - \theta^*\|^2
$$

$$
+ \Big\{\frac{\alpha}{\beta}\cdot\frac{2\rho_{\max}^2\gamma^2}{\lambda_{\widehat{A}}}\frac{3}{\lambda_C}\Big[1 + \Big[10\beta^2 + 10\gamma^2\rho_{\max}^2\big(1 + \frac{2}{\lambda_C}\big)\cdot\big(\rho_{\max}\frac{1+\gamma}{\min|\lambda(C)|}\big)^2\frac{\alpha^2}{\beta}\Big]M\Big] + \alpha^2 M \cdot 5\gamma^2\rho_{\max}^2\Big\}
$$

$$
\cdot \mathbb{E}_{m,0}\|\tilde{z}^{(m-1)}\|^2
$$

$$
+ \alpha\beta \cdot \frac{60\rho_{\max}^2\gamma^2}{\lambda_{\widehat{A}}\lambda_C}\cdot K_2 + \frac{\alpha^3}{\beta^2}\cdot\frac{60\rho_{\max}^2\gamma^2}{\lambda_{\widehat{A}}\lambda_C}\cdot\big(\rho_{\max}\frac{1+\gamma}{\min|\lambda(C)|}\big)^2 K_1\big(1 + \frac{2}{\lambda_C}\big) + \alpha^2 \cdot 5K_1.
$$

476   where $C_1$ and $C_2$ are specified in eq.(20) and eq.(21), respectively. Dividing $\frac{\lambda_{\widehat{A}}}{12}\alpha M$ and taking

477   total expectation on both sides of the above inequality, and applying Lemma D.5 to bound the term

478   $\mathbb{E}\|\tilde{z}^{(m-1)}\|^2$, we obtain that

$$\mathbb{E}\|\tilde{\theta}^{(m)} - \theta^*\|^2$$

$$\leq D \cdot \mathbb{E}_{m,0}\|\tilde{\theta}^{(m-1)} - \theta^*\|^2$$

$$+ \frac{12}{\lambda_{\widehat{A}}}\frac{1}{\alpha M}\left\{\frac{\alpha}{\beta} \cdot \frac{2\rho_{\max}^2\gamma^2}{\lambda_{\widehat{A}}}\frac{3}{\lambda_C}\left[1 + \left[10\beta^2 + 10\gamma^2\rho_{\max}^2\left(1 + \frac{2}{\lambda_C}\right) \cdot \left(\rho_{\max}\frac{1+\gamma}{\min|\lambda(C)|}\right)^2\frac{\alpha^2}{\beta}\right]M\right] + \alpha^2 M \cdot 5\gamma^2\rho_{\max}^2\right\}$$

$$\times \left\{F^{m-1} \cdot \mathbb{E}\|\tilde{z}^{(0)}\|^2 + \frac{D^{m-1} - F^{m-1}}{D - F} \cdot \mathbb{E}\|\tilde{\theta}^{(0)} - \theta^*\|^2 + \frac{\frac{D^{m-1}-F^{m-1}}{D-F} - \frac{E^{m-1}-F^{m-1}}{E-F}}{D - E}\mathbb{E}\|\tilde{z}^{(0)}\|^2\right.$$

$$+ \frac{1}{1-F}\frac{1}{1-D}\frac{1152}{\lambda_{\widehat{A}}^2}\frac{\rho_{\max}^2\gamma^2}{\lambda_C^2}\left(\frac{\alpha^2}{\beta^2} \cdot C_3 + \beta \cdot C_4\right)\left[\frac{1}{\beta M} + \beta \cdot 20 + \frac{\alpha^2}{\beta^2} \cdot 20\gamma^2\rho_{\max}^2\left(1 + \frac{2}{\lambda_C}\right) \cdot \left(\rho_{\max}\frac{1+\gamma}{\min|\lambda(C)|}\right)^2\right]$$

$$\times \left[\beta \cdot \frac{24}{\lambda_C}H_{\text{VR}}^2 + \frac{\alpha^2}{\beta^2} \cdot \left(1 + \frac{2}{\lambda_C}\right) \cdot \frac{2}{\lambda_C}\left(\rho_{\max}\frac{1+\gamma}{\min|\lambda(C)|}\right)^2 G_{\text{VR}}^2\right]$$

$$\left. + \frac{1}{1-F}\frac{1}{1-D}\left[\frac{\beta}{M} \cdot \left(\frac{80}{\lambda_C}K_2 + C_4\frac{600}{\lambda_C}\frac{K_1}{\lambda_{\widehat{A}}}\right) + \frac{\alpha^2}{\beta^2}\frac{1}{M} \cdot \left(\frac{80}{\lambda_C}\left(\rho_{\max}\frac{1+\gamma}{\min|\lambda(C)|}\right)^2 K_1\left(1 + \frac{2}{\lambda_C}\right) + C_3\frac{600}{\lambda_C}\frac{K_1}{\lambda_{\widehat{A}}}\right)\right]\right\}$$

$$+ \frac{12}{\lambda_{\widehat{A}}}\frac{1}{\alpha M}\left\{\alpha\beta \cdot \frac{60\rho_{\max}^2\gamma^2}{\lambda_{\widehat{A}}\lambda_C} \cdot K_2 + \frac{\alpha^3}{\beta^2} \cdot \frac{60\rho_{\max}^2\gamma^2}{\lambda_{\widehat{A}}\lambda_C} \cdot \left(\rho_{\max}\frac{1+\gamma}{\min|\lambda(C)|}\right)^2 K_1\left(1 + \frac{2}{\lambda_C}\right) + \alpha^2 \cdot 5K_1\right\}.$$

479   Note that the second coefficient in the above inequality can be simplified as

$$\frac{12}{\lambda_{\widehat{A}}}\frac{1}{\alpha M}\left\{\frac{\alpha}{\beta} \cdot \frac{2\rho_{\max}^2\gamma^2}{\lambda_{\widehat{A}}}\frac{3}{\lambda_C}\left[1 + \left[10\beta^2 + 10\gamma^2\rho_{\max}^2\left(1 + \frac{2}{\lambda_C}\right) \cdot \left(\rho_{\max}\frac{1+\gamma}{\min|\lambda(C)|}\right)^2\frac{\alpha^2}{\beta}\right]M\right] + \alpha^2 M \cdot 5\gamma^2\rho_{\max}^2\right\}$$

$$= \frac{12}{\lambda_{\widehat{A}}}\left\{\frac{\alpha}{\beta} \cdot \frac{2\rho_{\max}^2\gamma^2}{\lambda_{\widehat{A}}}\frac{3}{\lambda_C}\left[\frac{1}{\beta M} + 10\beta + 10\gamma^2\rho_{\max}^2\left(1 + \frac{2}{\lambda_C}\right) \cdot \left(\rho_{\max}\frac{1+\gamma}{\min|\lambda(C)|}\right)^2\frac{\alpha^2}{\beta^2}\right] + \alpha \cdot 5\gamma^2\rho_{\max}^2\right\}$$

$$= \frac{\alpha}{\beta^2 M} \cdot \frac{72\rho_{\max}^2\gamma^2}{\lambda_{\widehat{A}}^2}\frac{1}{\lambda_C} + \beta \cdot \frac{720\rho_{\max}^2\gamma^2}{\lambda_{\widehat{A}}^2}\frac{1}{\lambda_C} + \frac{\alpha^2}{\beta^2} \cdot \frac{720\rho_{\max}^2\gamma^4}{\lambda_{\widehat{A}}^2}\frac{1}{\lambda_C}\rho_{\max}^2\left(1 + \frac{2}{\lambda_C}\right) \cdot \left(\rho_{\max}\frac{1+\gamma}{\min|\lambda(C)|}\right)^2$$

$$+ \alpha \cdot \frac{60}{\lambda_{\widehat{A}}}\gamma^2\rho_{\max}^2,$$

480   and note that we have assumed that

$$\frac{\alpha}{\beta^2 M} \cdot \frac{72\rho_{\max}^2\gamma^2}{\lambda_{\widehat{A}}^2}\frac{1}{\lambda_C} + \beta \cdot \frac{720\rho_{\max}^2\gamma^2}{\lambda_{\widehat{A}}^2}\frac{1}{\lambda_C}$$

$$+ \frac{\alpha^2}{\beta^2} \cdot \frac{720\rho_{\max}^2\gamma^4}{\lambda_{\widehat{A}}^2}\frac{1}{\lambda_C}\rho_{\max}^2\left(1 + \frac{2}{\lambda_C}\right) \cdot \left(\rho_{\max}\frac{1+\gamma}{\min|\lambda(C)|}\right)^2 + \alpha \cdot \frac{60}{\lambda_{\widehat{A}}}\gamma^2\rho_{\max}^2 \leq 1. \quad (10)$$

481   Then, we obtain that

$$\mathbb{E}\|\tilde{\theta}^{(m)} - \theta^*\|^2$$

$$\leq D \cdot \mathbb{E}_{m,0}\|\tilde{\theta}^{(m-1)} - \theta^*\|^2 + F^{m-1} \cdot \mathbb{E}\|\tilde{z}^{(0)}\|^2 + \frac{D^{m-1} - F^{m-1}}{D - F} \cdot \mathbb{E}\|\tilde{\theta}^{(0)} - \theta^*\|^2$$

$$+ \frac{\frac{D^{m-1}-F^{m-1}}{D-F} - \frac{E^{m-1}-F^{m-1}}{E-F}}{D - E}\mathbb{E}\|\tilde{z}^{(0)}\|^2$$

$$+ \left\{\frac{\alpha}{\beta^2 M} \cdot \frac{72\rho_{\max}^2\gamma^2}{\lambda_{\widehat{A}}^2}\frac{1}{\lambda_C} + \beta \cdot \frac{720\rho_{\max}^2\gamma^2}{\lambda_{\widehat{A}}^2}\frac{1}{\lambda_C}\right.$$

$$+ \frac{\alpha^2}{\beta^2} \cdot \frac{720\rho_{\max}^2\gamma^4}{\lambda_{\widehat{A}}^2}\frac{1}{\lambda_C}\rho_{\max}^2\left(1 + \frac{2}{\lambda_C}\right) \cdot \left(\rho_{\max}\frac{1+\gamma}{\min|\lambda(C)|}\right)^2 + \alpha \cdot \frac{60}{\lambda_{\widehat{A}}}\gamma^2\rho_{\max}^2\right\}$$

$$\times \left\{\frac{1}{1-F}\frac{1}{1-D}\frac{1152}{\lambda_{\widehat{A}}^2}\frac{\rho_{\max}^2\gamma^2}{\lambda_C^2}\left(\frac{\alpha^2}{\beta^2} \cdot C_3 + \beta \cdot C_4\right)\left[\frac{1}{\beta M} + \beta \cdot 20 + \frac{\alpha^2}{\beta^2} \cdot 20\gamma^2\rho_{\max}^2\left(1 + \frac{2}{\lambda_C}\right) \cdot \left(\rho_{\max}\frac{1+\gamma}{\min|\lambda(C)|}\right)^2\right]\right.$$

$$\cdot \Big[\beta \cdot \frac{24}{\lambda_C} H_{\mathrm{VR}}^2 + \frac{\alpha^2}{\beta^2} \cdot \big(1 + \frac{2}{\lambda_C}\big) \cdot \frac{2}{\lambda_C} \big(\rho_{\max} \frac{1+\gamma}{\min|\lambda(C)|}\big)^2 G_{\mathrm{VR}}^2\Big]$$

$$+ \frac{1}{1-F} \frac{1}{1-D} \Big[\frac{\beta}{M} \cdot \big(\frac{80}{\lambda_C} K_2 + C_4 \frac{600}{\lambda_C} \frac{K_1}{\lambda_{\widehat{A}}}\big) + \frac{\alpha^2}{\beta^2} \frac{1}{M} \cdot \big(\frac{80}{\lambda_C} \big(\rho_{\max} \frac{1+\gamma}{\min|\lambda(C)|}\big)^2 K_1 \big(1 + \frac{2}{\lambda_C}\big) + C_3 \frac{600}{\lambda_C} \frac{K_1}{\lambda_{\widehat{A}}}\big)\Big]\Big\}$$

$$+ \frac{12}{\lambda_{\widehat{A}}} \frac{1}{\alpha M} \Big\{\alpha\beta \cdot \frac{60\rho_{\max}^2 \gamma^2}{\lambda_{\widehat{A}} \lambda_C} \cdot K_2 + \frac{\alpha^3}{\beta^2} \cdot \frac{60\rho_{\max}^2 \gamma^2}{\lambda_{\widehat{A}} \lambda_C} \cdot \big(\rho_{\max} \frac{1+\gamma}{\min|\lambda(C)|}\big)^2 K_1 \big(1 + \frac{2}{\lambda_C}\big) + \alpha^2 \cdot 5K_1\Big\}$$

$$\leq D \cdot \mathbb{E}_{m,0} \|\tilde{\theta}^{(m-1)} - \theta^*\|^2 + F^{m-1} \cdot \mathbb{E}\|\tilde{z}^{(0)}\|^2 + \frac{D^{m-1} - F^{m-1}}{D - F} \cdot \mathbb{E}\|\tilde{\theta}^{(0)} - \theta^*\|^2$$

$$+ \frac{\frac{D^{m-1} - F^{m-1}}{D-F} - \frac{E^{m-1} - F^{m-1}}{E-F}}{D - E} \mathbb{E}\|\tilde{z}^{(0)}\|^2$$

$$+ \Big\{ \frac{\alpha}{\beta^2 M} \cdot \frac{72\rho_{\max}^2 \gamma^2}{\lambda_{\widehat{A}}^2} \frac{1}{\lambda_C} + \beta \cdot \frac{720\rho_{\max}^2 \gamma^2}{\lambda_{\widehat{A}}^2} \frac{1}{\lambda_C}$$

$$+ \frac{\alpha^2}{\beta^2} \cdot \frac{720\rho_{\max}^2 \gamma^4}{\lambda_{\widehat{A}}^2} \frac{1}{\lambda_C} \rho_{\max}^2 \big(1 + \frac{2}{\lambda_C}\big) \cdot \big(\rho_{\max} \frac{1+\gamma}{\min|\lambda(C)|}\big)^2 + \alpha \cdot \frac{60}{\lambda_{\widehat{A}}} \gamma^2 \rho_{\max}^2 \Big\}$$

$$\cdot \Big\{ \frac{1}{1-F} \frac{1}{1-D} \frac{1152}{\lambda_{\widehat{A}}^2} \frac{\rho_{\max}^2 \gamma^2}{\lambda_C^2} \big(\frac{\alpha^2}{\beta^2} \cdot C_3 + \beta \cdot C_4\big) \Big[\frac{1}{\beta M} + \beta \cdot 20 + \frac{\alpha^2}{\beta^2} \cdot 20\gamma^2 \rho_{\max}^2 \big(1 + \frac{2}{\lambda_C}\big) \cdot \big(\rho_{\max} \frac{1+\gamma}{\min|\lambda(C)|}\big)^2\Big]$$

$$\cdot \Big[\beta \cdot \frac{24}{\lambda_C} H_{\mathrm{VR}}^2 + \frac{\alpha^2}{\beta^2} \cdot \big(1 + \frac{2}{\lambda_C}\big) \cdot \frac{2}{\lambda_C} \big(\rho_{\max} \frac{1+\gamma}{\min|\lambda(C)|}\big)^2 G_{\mathrm{VR}}^2\Big]\Big\}$$

$$+ \frac{1}{1-F} \frac{1}{1-D} \Big[\frac{\beta}{M} \cdot \big(\frac{80}{\lambda_C} K_2 + C_4 \frac{600}{\lambda_C} \frac{K_1}{\lambda_{\widehat{A}}}\big) + \frac{\alpha^2}{\beta^2} \frac{1}{M} \cdot \big(\frac{80}{\lambda_C} \big(\rho_{\max} \frac{1+\gamma}{\min|\lambda(C)|}\big)^2 K_1 \big(1 + \frac{2}{\lambda_C}\big) + C_3 \frac{600}{\lambda_C} \frac{K_1}{\lambda_{\widehat{A}}}\big)\Big]$$

$$+ \frac{1}{1-F} \frac{1}{1-D} \Big\{\frac{\beta}{M} \cdot \frac{720\rho_{\max}^2 \gamma^2}{\lambda_{\widehat{A}}^2 \lambda_C} \cdot K_2 + \frac{\alpha^2}{\beta^2} \frac{1}{M} \cdot \frac{720\rho_{\max}^2 \gamma^2}{\lambda_{\widehat{A}}^2 \lambda_C} \cdot \big(\rho_{\max} \frac{1+\gamma}{\min|\lambda(C)|}\big)^2 K_1 \big(1 + \frac{2}{\lambda_C}\big) + \frac{\alpha}{M} \cdot \frac{60}{\lambda_{\widehat{A}}} K_1\Big\}$$

$$= D \cdot \mathbb{E}_{m,0} \|\tilde{\theta}^{(m-1)} - \theta^*\|^2 + F^{m-1} \cdot \mathbb{E}\|\tilde{z}^{(0)}\|^2 + \frac{D^{m-1} - F^{m-1}}{D - F} \cdot \mathbb{E}\|\tilde{\theta}^{(0)} - \theta^*\|^2$$

$$+ \frac{\frac{D^{m-1} - F^{m-1}}{D-F} - \frac{E^{m-1} - F^{m-1}}{E-F}}{D - E} \mathbb{E}\|\tilde{z}^{(0)}\|^2$$

$$+ \Big\{ \frac{\alpha}{\beta^2 M} \cdot \frac{72\rho_{\max}^2 \gamma^2}{\lambda_{\widehat{A}}^2} \frac{1}{\lambda_C} + \beta \cdot \frac{720\rho_{\max}^2 \gamma^2}{\lambda_{\widehat{A}}^2} \frac{1}{\lambda_C}$$

$$+ \frac{\alpha^2}{\beta^2} \cdot \frac{720\rho_{\max}^2 \gamma^4}{\lambda_{\widehat{A}}^2} \frac{1}{\lambda_C} \rho_{\max}^2 \big(1 + \frac{2}{\lambda_C}\big) \cdot \big(\rho_{\max} \frac{1+\gamma}{\min|\lambda(C)|}\big)^2 + \alpha \cdot \frac{60}{\lambda_{\widehat{A}}} \gamma^2 \rho_{\max}^2 \Big\}$$

$$\cdot \Big\{ \frac{1}{1-F} \frac{1}{1-D} \frac{1152}{\lambda_{\widehat{A}}^2} \frac{\rho_{\max}^2 \gamma^2}{\lambda_C^2} \big(\frac{\alpha^2}{\beta^2} \cdot C_3 + \beta \cdot C_4\big) \Big[\frac{1}{\beta M} + \beta \cdot 20 + \frac{\alpha^2}{\beta^2} \cdot 20\gamma^2 \rho_{\max}^2 \big(1 + \frac{2}{\lambda_C}\big) \cdot \big(\rho_{\max} \frac{1+\gamma}{\min|\lambda(C)|}\big)^2\Big]$$

$$\cdot \Big[\beta \cdot \frac{24}{\lambda_C} H_{\mathrm{VR}}^2 + \frac{\alpha^2}{\beta^2} \cdot \big(1 + \frac{2}{\lambda_C}\big) \cdot \frac{2}{\lambda_C} \big(\rho_{\max} \frac{1+\gamma}{\min|\lambda(C)|}\big)^2 G_{\mathrm{VR}}^2\Big]\Big\}$$

$$+ \frac{1}{1-F} \frac{1}{1-D} \Big\{\frac{\beta}{M} \cdot \Big[\frac{720\rho_{\max}^2 \gamma^2}{\lambda_{\widehat{A}}^2 \lambda_C} K_2 + \big(\frac{80}{\lambda_C} K_2 + C_4 \frac{600}{\lambda_C} \frac{K_1}{\lambda_{\widehat{A}}}\big)\Big]$$

$$+ \frac{\alpha^2}{\beta^2} \frac{1}{M} \cdot \Big[\frac{720\rho_{\max}^2 \gamma^2}{\lambda_{\widehat{A}}^2 \lambda_C} \cdot \big(\rho_{\max} \frac{1+\gamma}{\min|\lambda(C)|}\big)^2 K_1 \big(1 + \frac{2}{\lambda_C}\big) + \big(\frac{80}{\lambda_C} \big(\rho_{\max} \frac{1+\gamma}{\min|\lambda(C)|}\big)^2 K_1 \big(1 + \frac{2}{\lambda_C}\big) + C_3 \frac{600}{\lambda_C} \frac{K_1}{\lambda_{\widehat{A}}}\big)\Big]$$

$$+ \frac{\alpha}{M} \cdot \frac{60}{\lambda_{\widehat{A}}} K_1\Big\},$$

where we use eq.(10) in the first step, use $1 < \frac{1}{1-D} \frac{1}{1-F}$ in the second step, and rearrange the terms in the last step. Next, we telescope the above inequality over $m$. To further simplify the result, we choose the optimal relation between $\alpha$ and $\beta$, i.e., $\beta = \mathcal{O}(\alpha^{2/3})$. Then, for sufficiently small $\alpha$ and $\beta$, we have $D > E$ and $D > F$, and we obtain that

$$\mathbb{E}\|\tilde{\theta}^{(m)} - \theta^*\|^2$$

$$\leq D^m \cdot \|\tilde{\theta}^{(0)} - \theta^*\|^2 + D^{m-1} \cdot \mathbb{E}\|\tilde{z}^{(0)}\|^2 + \frac{mD^{m-1}}{D-F} \cdot \mathbb{E}\|\tilde{\theta}^{(0)} - \theta^*\|^2 + \frac{mD^{m-1}}{(D-E)(D-F)}\mathbb{E}\|\tilde{z}^{(0)}\|^2$$

$$+ \frac{1}{1-D}\Big\{\frac{\alpha}{\beta^2 M} \cdot \frac{72\rho_{\max}^2\gamma^2}{\lambda_{\widehat{A}}^2}\frac{1}{\lambda_C}$$

$$+ \beta \cdot \frac{720\rho_{\max}^2\gamma^2}{\lambda_{\widehat{A}}^2}\frac{1}{\lambda_C} + \frac{\alpha^2}{\beta^2} \cdot \frac{720\rho_{\max}^2\gamma^4}{\lambda_{\widehat{A}}^2}\frac{1}{\lambda_C}\rho_{\max}^2\big(1 + \frac{2}{\lambda_C}\big) \cdot \big(\rho_{\max}\frac{1+\gamma}{\min|\lambda(C)|}\big)^2 + \alpha \cdot \frac{60}{\lambda_{\widehat{A}}}\gamma^2\rho_{\max}^2\Big\}$$

$$\times \Big\{\frac{1}{1-F}\frac{1}{1-D}\frac{1152}{\lambda_{\widehat{A}}^2}\frac{\rho_{\max}^2\gamma^2}{\lambda_C^2}\big(\frac{\alpha^2}{\beta^2} \cdot C_3 + \beta \cdot C_4\big)\Big[\frac{1}{\beta M} + \beta \cdot 20 + \frac{\alpha^2}{\beta^2} \cdot 20\gamma^2\rho_{\max}^2\big(1 + \frac{2}{\lambda_C}\big) \cdot \big(\rho_{\max}\frac{1+\gamma}{\min|\lambda(C)|}\big)^2\Big]$$

$$\times \Big[\beta \cdot \frac{24}{\lambda_C}H_{\text{VR}}^2 + \frac{\alpha^2}{\beta^2} \cdot \big(1 + \frac{2}{\lambda_C}\big) \cdot \frac{2}{\lambda_C}\big(\rho_{\max}\frac{1+\gamma}{\min|\lambda(C)|}\big)^2 G_{\text{VR}}^2\Big]\Big\}$$

$$+ \frac{1}{1-F}\frac{1}{(1-D)^2}\Big\{\frac{\beta}{M} \cdot \Big[\frac{720\rho_{\max}^2\gamma^2}{\lambda_{\widehat{A}}^2\lambda_C}K_2 + \big(\frac{80}{\lambda_C}K_2 + C_4\frac{600}{\lambda_C}\frac{K_1}{\lambda_{\widehat{A}}}\big)\Big]$$

$$+ \frac{\alpha^2}{\beta^2}\frac{1}{M} \cdot \Big[\frac{720\rho_{\max}^2\gamma^2}{\lambda_{\widehat{A}}^2\lambda_C} \cdot \big(\rho_{\max}\frac{1+\gamma}{\min|\lambda(C)|}\big)^2 K_1\big(1 + \frac{2}{\lambda_C}\big) + \big(\frac{80}{\lambda_C}\big(\rho_{\max}\frac{1+\gamma}{\min|\lambda(C)|}\big)^2 K_1\big(1 + \frac{2}{\lambda_C}\big)$$

$$+ C_3\frac{600}{\lambda_C}\frac{K_1}{\lambda_{\widehat{A}}}\big)\Big] + \frac{\alpha}{M} \cdot \frac{60}{\lambda_{\widehat{A}}}K_1\Big\}.$$

The first four terms in the right hand side of the above inequality are dominated by the order $\mathcal{O}(mD^m)$. Also, under the choices of learning rates, the fifth term is in the order of $\mathcal{O}(\beta^4)$ and the last term (in the last three lines) is in the order of $\mathcal{O}(\frac{\beta}{M})$. To elaborate this, we note that the fifth term is a product of two curly brackets: the first one is in the order of $\mathcal{O}(\frac{\alpha}{\beta^2}\frac{1}{M}) + \mathcal{O}(\beta)$, the second one is in the order of $\mathcal{O}(\beta) \times \big(\mathcal{O}(\frac{1}{\beta M}) + \mathcal{O}(\beta)\big) \times \mathcal{O}(\beta) = \mathcal{O}(\frac{\beta}{M}) + \mathcal{O}(\beta^3)$. So, their product is in the order of $\big(\mathcal{O}(\frac{\alpha}{\beta^2}\frac{1}{M}) + \mathcal{O}(\beta)\big) \times \big(\mathcal{O}(\frac{\beta}{M}) + \mathcal{O}(\beta^3)\big) = \mathcal{O}(\frac{\alpha}{\beta}\frac{1}{M^2}) + \mathcal{O}(\frac{\beta^2}{M}) + \mathcal{O}(\frac{\alpha\beta}{M}) + \mathcal{O}(\beta^4) = \mathcal{O}(\frac{\beta^2}{M}) + \mathcal{O}(\beta^4)$. The last term is in the order of $\frac{\beta}{M}$. Therefore, the above inequality implies that

$$\mathbb{E}\|\tilde{\theta}^{(m)} - \theta^*\|^2 \leq \mathcal{O}(mD^m) + \mathcal{O}(\beta^4) + \mathcal{O}(\frac{\beta}{M}).$$

Next, we compute the sample complexity for achieving $\mathbb{E}\|\tilde{\theta}^{(m)} - \theta^*\|^2 \leq \epsilon$. The above convergence rate implies that, for sufficiently small $\beta$ and sufficiently large $M$, there always exists constants $I_1, I_2, I_3$ such that

$$\mathbb{E}\|\tilde{\theta}^{(m)} - \theta^*\|^2 \leq mD^m I_1 + \beta^4 I_2 + \frac{\beta}{M}I_3.$$

We require

(i) $\beta^4 I_2 \leq \epsilon/3 \Rightarrow \beta \leq I_2^{1/4}\epsilon^{1/4} = \mathcal{O}(\epsilon^{1/4})$.

(ii) $\frac{\beta}{M}I_3 \leq \epsilon/3 \Rightarrow M \geq \mathcal{O}(\frac{\beta}{\epsilon})$.

(iii) $mD^m I_1 \leq \epsilon/3$. We notice that this inequality implies $D^m I_1 \leq \epsilon/(3m)$.

We choose $m = \mathcal{O}(\log \epsilon^{-1})$ so that $m \leq \mathcal{O}(\log \epsilon^{-2})$. Using the upper bound of $m$, the requirement in (iii) suffices to require that $D^m \leq \overline{\mathcal{O}}(\epsilon/\log \epsilon^{-2})$, which further implies that $m \geq \mathcal{O}(\log \epsilon^{-1} + \log\log \epsilon^{-2})/\log D^{-1} = \mathcal{O}(\log \epsilon^{-1})$ (note that $D < 1$). Hence, it is valid to choose $m = \mathcal{O}(\log \epsilon^{-1})$. Also, since $\alpha = \mathcal{O}(\beta^{2/3})$, then $D \leq 1$ requires that $M \geq \mathcal{O}(\beta^{-3/2})$, which combines with (ii) further requires that

$$M \geq \max\{\mathcal{O}(\frac{\beta}{\epsilon}), \mathcal{O}(\beta^{-3/2})\}.$$

Let $\mathcal{O}(\frac{\beta}{\epsilon})$ and $\mathcal{O}(\beta^{-3/2})$ be of the same order, i.e., $\beta = \mathcal{O}(\epsilon^{2/5})$, which satisfies (i). So overall we require that $M \geq \mathcal{O}(\epsilon^{-3/5})$, which leads to the sample complexity

$$mM \geq \mathcal{O}(\epsilon^{-3/5}\log \epsilon^{-1}).$$

## C  Proof of Corollary 3.2

**Corollary C.1.** *Under the same assumptions as those of Theorem 3.1, choose the learning rates $\alpha, \beta$ and the batch size $M$ such that all requirements of Theorem 3.1 are satisfied. Then, the following refined bound holds.*

$$\mathbb{E}\|\tilde{z}^{(m)}\|^2$$

$$\leq F^m \cdot \mathbb{E}\|\tilde{z}^{(0)}\|^2 + \frac{D^m - F^m}{D - F} \cdot \mathbb{E}\|\tilde{\theta}^{(0)} - \theta^*\|^2 + \frac{\frac{D^m - F^m}{D - F} - \frac{E^m - F^m}{E - F}}{D - E} \mathbb{E}\|\tilde{z}^{(0)}\|^2$$

$$+ \frac{1}{1 - F}\frac{1}{1 - D}\frac{1152}{\lambda_{\widehat{A}}^2}\frac{\rho_{\max}^2\gamma^2}{\lambda_C^2}\left(\frac{\alpha^2}{\beta^2} \cdot C_3 + \beta \cdot C_4\right)\left[\frac{1}{\beta M} + \beta \cdot 20 + \frac{\alpha^2}{\beta^2} \cdot 20\gamma^2\rho_{\max}^2\left(1 + \frac{2}{\lambda_C}\right) \cdot \left(\rho_{\max}\frac{1 + \gamma}{\min|\lambda(C)|}\right)^2\right]$$

$$\times \left[\beta \cdot \frac{24}{\lambda_C}H_{VR}^2 + \frac{\alpha^2}{\beta^2} \cdot \left(1 + \frac{2}{\lambda_C}\right) \cdot \frac{2}{\lambda_C}\left(\rho_{\max}\frac{1 + \gamma}{\min|\lambda(C)|}\right)^2 G_{VR}^2\right]$$

$$+ \frac{1}{1 - F}\frac{1}{1 - D}\left[\frac{\beta}{M} \cdot \left(\frac{80}{\lambda_C}K_2 + C_4\frac{600}{\lambda_C}\frac{K_1}{\lambda_{\widehat{A}}}\right) + \frac{\alpha^2}{\beta^2}\frac{1}{M} \cdot \left(\frac{80}{\lambda_C}\left(\rho_{\max}\frac{1 + \gamma}{\min|\lambda(C)|}\right)^2 K_1\left(1 + \frac{2}{\lambda_C}\right) + C_3\frac{600}{\lambda_C}\frac{K_1}{\lambda_{\widehat{A}}}\right)\right].$$

*where $K_1$ is specified in eq.(26) in Lemma E.1, $K_2$ is specified in eq.(32) in Lemma E.2, $D$ and $E$ are specified in eq.(18) and eq.(19) in Lemma D.4.*

*Proof.* See Lemma D.5. Next, we derive its asymptotic upper bound under the setting $\beta = \mathcal{O}(\alpha^{2/3})$. We note that all the conditions of Theorem 3.1 on the learning rates $\alpha, \beta$ and the batch size $M$ can be satisfied with a sufficiently small $\alpha$ in this setting. The first three terms are in the order of $D^m$ (because $D > E, D > F$). Here we mainly discuss the order of the fourth and fifth term in the above bound, and note that this term is a product of three brackets. Since we set $\beta = \mathcal{O}(\alpha^{2/3})$, the first bracket of this product is in the order of $\mathcal{O}(\beta)$, and the second bracket of this product is in the order of $\mathcal{O}(\frac{1}{\beta M}) + \mathcal{O}(\beta)$. The last bracket of this product is in the order of $\mathcal{O}(\beta)$. Therefore, the fourth term in the above bound is in the order of $\mathcal{O}(\frac{\beta}{M}) + \mathcal{O}(\beta^3)$. Also, the last term of this upper bound is in the order of $\mathcal{O}(\frac{\beta}{M})$. Overall, we can obtain that

$$\mathbb{E}\|\tilde{z}^{(m)}\|^2 = \mathcal{O}(D^m) + \mathcal{O}(\frac{\beta}{M}) + \mathcal{O}(\beta^3).$$

By following the same proof logic of Theorem 3.1, we obtain the desired complexity result.

$\square$

## D  Key Lemmas for Proving Theorem 3.1

**Lemma D.1** (Preliminary Bound for $\sum_{t=0}^{M-1}\mathbb{E}_{m,0}\|\theta_t^{(m)} - \theta^*\|^2$). *Under the same assumptions as those of Theorem 3.1, choose the learning rate $\alpha$ such that*

$$\alpha \leq \min\left\{\frac{1}{5\lambda_{\widehat{A}}}, \frac{\lambda_{\widehat{A}}}{60}/\left[(1 + \gamma)^2\rho_{\max}^2\left(1 + \frac{\gamma\rho_{\max}}{\min|\lambda(C)|}\right)^2\right]\right\}. \tag{11}$$

*Then, the following preliminary bound holds, where $K_1$ is specified in eq.(26) in Lemma E.1.*

$$\frac{\lambda_{\widehat{A}}}{6}\alpha\sum_{t=0}^{M-1}\mathbb{E}_{m,0}\|\theta_t^{(m)} - \theta^*\|^2$$

$$\leq \left[1 + \alpha^2 M \cdot 5(1 + \gamma)^2\rho_{\max}^2\left(1 + \frac{\gamma\rho_{\max}}{\min|\lambda(C)|}\right)^2\right]\mathbb{E}_{m,0}\|\tilde{\theta}^{(m-1)} - \theta^*\|^2 + \alpha^2 \cdot 5K_1$$

$$+ \alpha \cdot \frac{2\rho_{\max}^2\gamma^2}{\lambda_{\widehat{A}}}\sum_{t=0}^{M-1}\mathbb{E}_{m,0}\|z_t^{(m)}\|^2 + \alpha^2 M \cdot 5\gamma^2\rho_{\max}^2\mathbb{E}_{m,0}\|\tilde{z}^{(m-1)}\|^2.$$

*Proof.* Based on the update rule of VRTDC for i.i.d. samples, we obtain that

$$\theta_{t+1}^{(m)} = \Pi_{R_\theta}\left[\theta_t^{(m)} + \alpha[G_t^{(m)}(\theta_t^{(m)}, z_t^{(m)}) - G_t^{(m)}(\tilde{\theta}^{(m-1)}, \tilde{z}^{(m-1)}) + G^{(m)}(\tilde{\theta}^{(m-1)}, \tilde{z}^{(m-1)})]\right].$$

The above update rule further implies that

$$\|\theta_{t+1}^{(m)} - \theta^*\|^2$$

$$\overset{(i)}{\leq} \|\theta_t^{(m)} - \theta^* + \alpha[G_t^{(m)}(\theta_t^{(m)}, z_t^{(m)}) - G_t^{(m)}(\tilde{\theta}^{(m-1)}, \tilde{z}^{(m-1)}) + G^{(m)}(\tilde{\theta}^{(m-1)}, \tilde{z}^{(m-1)})]\|^2$$

$$= \|\theta_t^{(m)} - \theta^*\|^2 + \alpha^2\|G_t^{(m)}(\theta_t^{(m)}, z_t^{(m)}) - G_t^{(m)}(\tilde{\theta}^{(m-1)}, \tilde{z}^{(m-1)}) + G^{(m)}(\tilde{\theta}^{(m-1)}, \tilde{z}^{(m-1)})\|^2$$

$$+ 2\alpha\langle\theta_t^{(m)} - \theta^*, G_t^{(m)}(\theta_t^{(m)}, z_t^{(m)}) - G_t^{(m)}(\tilde{\theta}^{(m-1)}, \tilde{z}^{(m-1)}) + G^{(m)}(\tilde{\theta}^{(m-1)}, \tilde{z}^{(m-1)})\rangle, \quad (12)$$

where (i) uses the assumption that $R_\theta \geq \|\theta^*\|$ (i.e., $\theta^*$ is in the ball with radius $R_\theta$) and the fact that $\Pi_{R_\theta}$ is 1-Lipschitz. Then, we take the expectation $\mathbb{E}_{m,0}$ on both sides. In particular, an upper bound for the second variance term is given in Lemma E.1. Next, we bound the last term. Note that $\theta_t^{(m)} \in \mathcal{F}_{m,t-1}$ by the definition of the given filtration. Also, the i.i.d. sampling implies that $\mathbb{E}_{m,t-1}[-G_t^{(m)}(\tilde{\theta}^{(m-1)}, \tilde{z}^{(m-1)}) + G^{(m)}(\tilde{\theta}^{(m-1)}, \tilde{z}^{(m-1)})] = 0$. Therefore, for the last term of the above equation, we obtain that

$$\mathbb{E}_{m,0}\langle\theta_t^{(m)} - \theta^*, G_t^{(m)}(\theta_t^{(m)}, z_t^{(m)}) - G_t^{(m)}(\tilde{\theta}^{(m-1)}, \tilde{z}^{(m-1)}) + G^{(m)}(\tilde{\theta}^{(m-1)}, \tilde{z}^{(m-1)})\rangle$$

$$=\mathbb{E}_{m,0}\langle\theta_t^{(m)} - \theta^*, \mathbb{E}_{m,t-1}[G_t^{(m)}(\theta_t^{(m)}, z_t^{(m)}) - G_t^{(m)}(\tilde{\theta}^{(m-1)}, \tilde{z}^{(m-1)}) + G^{(m)}(\tilde{\theta}^{(m-1)}, \tilde{z}^{(m-1)})]\rangle$$

$$=\mathbb{E}_{m,0}\langle\theta_t^{(m)} - \theta^*, \mathbb{E}_{m,t-1}G_t^{(m)}(\theta_t^{(m)}, z_t^{(m)})\rangle$$

$$+ \mathbb{E}_{m,0}\langle\theta_t^{(m)} - \theta^*, \mathbb{E}_{m,t-1}[-G_t^{(m)}(\tilde{\theta}^{(m-1)}, \tilde{z}^{(m-1)}) + G^{(m)}(\tilde{\theta}^{(m-1)}, \tilde{z}^{(m-1)})]\rangle$$

$$=\mathbb{E}_{m,0}\langle\theta_t^{(m)} - \theta^*, \mathbb{E}_{m,t-1}G_t^{(m)}(\theta_t^{(m)}, z_t^{(m)})\rangle$$

$$=\mathbb{E}_{m,0}\langle\theta_t^{(m)} - \theta^*, \mathbb{E}_{m,t-1}[\widehat{A}_t^{(m)}\theta_t^{(m)} + \widehat{b}_t^{(m)} + B_t^{(m)}z_t^{(m)}]\rangle$$

$$=\mathbb{E}_{m,0}\langle\theta_t^{(m)} - \theta^*, \mathbb{E}_{m,t-1}[\widehat{A}_t^{(m)}\theta_t^{(m)} + \widehat{b}_t^{(m)}]\rangle + \mathbb{E}_{m,0}\langle\theta_t^{(m)} - \theta^*, B_t^{(m)}z_t^{(m)}\rangle. \quad (13)$$

Note that $\mathbb{E}_{m,t-1}\widehat{A}_t^{(m)} = \widehat{A}$ and $\widehat{A}\theta^* + \widehat{b} = 0$, the first term of eq. (13) above can be simplified as

$$\mathbb{E}_{m,0}\langle\theta_t^{(m)} - \theta^*, \mathbb{E}_{m,t-1}[\widehat{A}_t^{(m)}\theta_t^{(m)} + \widehat{b}_t^{(m)}]\rangle = \mathbb{E}_{m,0}\langle\theta_t^{(m)} - \theta^*, \widehat{A}\theta_t^{(m)} + \widehat{b}\rangle$$

$$= \mathbb{E}_{m,0}\langle\theta_t^{(m)} - \theta^*, \widehat{A}(\theta_t^{(m)} - \theta^*) + \widehat{A}\theta^* + \widehat{b}\rangle$$

$$= \mathbb{E}_{m,0}\langle\theta_t^{(m)} - \theta^*, \widehat{A}(\theta_t^{(m)} - \theta^*)\rangle$$

$$\leq -\frac{\lambda_{\widehat{A}}}{2}\mathbb{E}_{m,0}\|\theta_t^{(m)} - \theta^*\|^2,$$

where the last inequality uses the property of negative definite matrix $(\theta - \theta^*)^T\widehat{A}(\theta - \theta^*) \leq \lambda_{\max}(\widehat{A})\|\theta - \theta^*\|^2$ and the definition that $\lambda_{\widehat{A}} := -\lambda_{\max}(\widehat{A} + \widehat{A}^\top)$. The last term of eq. (13) can be bounded using the inequality $2\langle u, v\rangle \leq \|u\|^2 + \|v\|^2$ as

$$\mathbb{E}_{m,0}\langle\theta_t^{(m)} - \theta^*, B^{(m)}z_t^{(m)}\rangle \leq \frac{1}{2}\cdot\frac{\lambda_{\widehat{A}}}{2}\mathbb{E}_{m,0}\|\theta_t^{(m)} - \theta^*\|^2 + \frac{1}{2}\cdot\frac{2}{\lambda_{\widehat{A}}}\rho_{\max}^2\gamma^2 \cdot \mathbb{E}_{m,0}\|z_t^{(m)}\|^2,$$

where we have used the fact that $\|B_t^{(m)}\| \leq \rho_{\max}\gamma$. Substituting these inequalities into eq. (13), we obtain that

$$\mathbb{E}_{m,0}\langle\theta_t^{(m)} - \theta^*, G_t^{(m)}(\theta_t^{(m)}, z_t^{(m)}) - G_t^{(m)}(\tilde{\theta}^{(m-1)}, \tilde{z}^{(m-1)}) + G^{(m)}(\tilde{\theta}^{(m-1)}, \tilde{z}^{(m-1)})\rangle$$

$$\leq -\frac{\lambda_{\widehat{A}}}{4}\mathbb{E}_{m,0}\|\theta_t^{(m)} - \theta^*\|^2 + \frac{\rho_{\max}^2\gamma^2}{\lambda_{\widehat{A}}}\mathbb{E}_{m,0}\|z_t^{(m)}\|^2.$$

Substituting the above inequality into eq. (12) yields that

$$\mathbb{E}_{m,0}\|\theta_{t+1}^{(m)} - \theta^*\|^2$$

$$\leq\mathbb{E}_{m,0}\|\theta_t^{(m)} - \theta^*\|^2 + \alpha\Big[-\frac{\lambda_{\widehat{A}}}{4}\mathbb{E}_{m,0}\|\theta_t^{(m)} - \theta^*\|^2 + \frac{\rho_{\max}^2\gamma^2}{\lambda_{\widehat{A}}}\mathbb{E}_{m,0}\|z_t^{(m)}\|^2\Big]$$

$$+ \alpha^2\Big[5(1+\gamma)^2\rho_{\max}^2\Big(1 + \frac{\gamma\rho_{\max}}{\min|\lambda(C)|}\Big)^2\big(\mathbb{E}_{m,0}\|\theta_t^{(m)} - \theta^*\|^2 + \mathbb{E}_{m,0}\|\tilde{\theta}^{(m-1)} - \theta^*\|^2\big)\Big]$$

$$+ \alpha^2 \big[ 5\gamma^2 \rho_{\max}^2 \big( \mathbb{E}_{m,0} \| z_t^{(m)} \|^2 + \mathbb{E}_{m,0} \| \tilde{z}^{(m-1)} \|^2 \big) + \frac{5K_1}{M} \big]$$

$$= \mathbb{E}_{m,0} \| \theta_t^{(m)} - \theta^* \|^2 - \big( \frac{\lambda_{\widehat{A}}}{4} \alpha - \alpha^2 \cdot 5(1+\gamma)^2 \rho_{\max}^2 \big( 1 + \frac{\gamma \rho_{\max}}{\min |\lambda(C)|} \big)^2 \big) \mathbb{E}_{m,0} \| \theta_t^{(m)} - \theta^* \|^2$$

$$+ \alpha^2 \cdot 5(1+\gamma)^2 \rho_{\max}^2 \big( 1 + \frac{\gamma \rho_{\max}}{\min |\lambda(C)|} \big)^2 \mathbb{E}_{m,0} \| \tilde{\theta}^{(m-1)} - \theta^* \|^2 + \frac{\alpha^2}{M} \cdot 5K_1$$

$$+ \big( \alpha \cdot \frac{\rho_{\max}^2 \gamma^2}{\lambda_{\widehat{A}}} + \alpha^2 \cdot 5\gamma^2 \rho_{\max}^2 \big) \mathbb{E}_{m,0} \| z_t^{(m)} \|^2 + \alpha^2 \cdot 5\gamma^2 \rho_{\max}^2 \mathbb{E}_{m,0} \| \tilde{z}^{(m-1)} \|^2 .$$

542    Summing the above inequality over $t = 0, \ldots, M-1$ yields that

$$\big( \frac{\lambda_{\widehat{A}}}{4} \alpha - \alpha^2 \cdot 5(1+\gamma)^2 \rho_{\max}^2 \big( 1 + \frac{\gamma \rho_{\max}}{\min |\lambda(C)|} \big)^2 \big) \sum_{t=0}^{M-1} \mathbb{E}_{m,0} \| \theta_t^{(m)} - \theta^* \|^2$$

$$\leq \big[ 1 + \alpha^2 M \cdot 5(1+\gamma)^2 \rho_{\max}^2 \big( 1 + \frac{\gamma \rho_{\max}}{\min |\lambda(C)|} \big)^2 \big] \mathbb{E}_{m,0} \| \tilde{\theta}^{(m-1)} - \theta^* \|^2 + \alpha^2 \cdot 5K_1$$

$$+ \big( \alpha \cdot \frac{\rho_{\max}^2 \gamma^2}{\lambda_{\widehat{A}}} + \alpha^2 \cdot 5\gamma^2 \rho_{\max}^2 \big) \sum_{t=0}^{M-1} \mathbb{E}_{m,0} \| z_t^{(m)} \|^2 + \alpha^2 M \cdot 5\gamma^2 \rho_{\max}^2 \mathbb{E}_{m,0} \| \tilde{z}^{(m-1)} \|^2 .$$

543    To further simplify the above inequality, we choose a sufficiently small $\alpha$ such that $\alpha^2 \cdot 5\gamma^2 \rho_{\max}^2 \leq$
544    $\alpha \cdot \frac{\rho_{\max}^2 \gamma^2}{\lambda_{\widehat{A}}}$ and $\frac{\lambda_{\widehat{A}}}{4} \alpha - \alpha^2 \cdot 5(1+\gamma)^2 \rho_{\max}^2 \big( 1 + \frac{\gamma \rho_{\max}}{\min |\lambda(C)|} \big)^2 \geq \frac{\lambda_{\widehat{A}}}{6} \alpha$. Then, the above inequality can
545    be rewritten as

$$\frac{\lambda_{\widehat{A}}}{6} \alpha \sum_{t=0}^{M-1} \mathbb{E}_{m,0} \| \theta_t^{(m)} - \theta^* \|^2$$

$$\leq \big[ 1 + \alpha^2 M \cdot 5(1+\gamma)^2 \rho_{\max}^2 \big( 1 + \frac{\gamma \rho_{\max}}{\min |\lambda(C)|} \big)^2 \big] \mathbb{E}_{m,0} \| \tilde{\theta}^{(m-1)} - \theta^* \|^2 + \alpha^2 \cdot 5K_1$$

$$+ \alpha \cdot \frac{2\rho_{\max}^2 \gamma^2}{\lambda_{\widehat{A}}} \sum_{t=0}^{M-1} \mathbb{E}_{m,0} \| z_t^{(m)} \|^2 + \alpha^2 M \cdot 5\gamma^2 \rho_{\max}^2 \mathbb{E}_{m,0} \| \tilde{z}^{(m-1)} \|^2 .$$

546    $\qquad\qquad\qquad\qquad\qquad\qquad\qquad\qquad\qquad\qquad\qquad\qquad\qquad\qquad\qquad\qquad$ $\square$

547    **Lemma D.2** (Preliminary bound for $\mathbb{E} \| \tilde{z}^{(m)} \|^2$)**.** *Under the same assumptions as those of Theorem 3.1,*
548    *choose the learning rate $\beta$ and the batch size $M$ such that $\beta < 1$ and $M\beta > \frac{4}{\lambda_C}$. Then, the following*
549    *preliminary bound holds.*

$$\mathbb{E} \| \tilde{z}^{(m)} \|^2 \leq \big( \frac{1}{M\beta} \cdot \frac{2}{\lambda_C} \big)^m \mathbb{E} \| \tilde{z}^{(0)} \|^2$$

$$+ 2 \cdot \big[ \beta \cdot \frac{24}{\lambda_C} H_{VR}^2 + \frac{\alpha^2}{\beta^2} \cdot \big( 1 + \frac{2}{\lambda_C} \big) \cdot \frac{2}{\lambda_C} \big( \rho_{\max} \frac{1+\gamma}{\min |\lambda(C)|} \big)^2 G_{VR}^2 \big].$$

550    *where $H_{VR}, G_{VR}$ is defined in Lemma J.5 and Lemma J.4.*

551    *Proof.* First, based on the update rule of $w_t^{(m)}$, we have

$$w_{t+1}^{(m)} = \Pi_{R_w} \big[ w_t^{(m)} + A_t^{(m)} \theta_t^{(m)} + b_t^{(m)} + C_t^{(m)} w_t^{(m)} \big],$$

552    which further implies the following one-step update rule of the tracking error $z_t^{(m)}$.

$$z_{t+1}^{(m)} = \Pi_{R_w} \big[ w_t^{(m)} + A_t^{(m)} \theta_t^{(m)} + b_t^{(m)} + C_t^{(m)} w_t^{(m)} \big] + C^{-1} (b + A(\tilde{\theta}^{(m)})).$$

553    Then, its square norm can be bounded as

$$\| z_{t+1}^{(m)} \|^2 \overset{(i)}{\leq} \| z_t^{(m)} + \beta \big[ H_t^{(m)} (\theta_t^{(m)}, z_t^{(m)}) - H_t^{(m)} (\tilde{\theta}^{(m-1)}, \tilde{z}^{(m-1)}) + H^{(m)} (\tilde{\theta}^{(m-1)}, \tilde{z}^{(m-1)}) \big]$$

$$+ C^{-1} A (\theta_{t+1}^{(m)} - \theta_t^{(m)}) \|^2$$

$$
= \|z_t^{(m)}\|^2 + 2\beta^2 \|H_t^{(m)}(\theta_t^{(m)}, z_t^{(m)}) - H_t^{(m)}(\tilde{\theta}^{(m-1)}, \tilde{z}^{(m-1)}) + H^{(m)}(\tilde{\theta}^{(m-1)}, \tilde{z}^{(m-1)})\|^2
$$
$$
+ 2\|C^{-1}A(\theta_{t+1}^{(m)} - \theta_t^{(m)})\|^2
$$
$$
+ 2\beta\langle z_t^{(m)}, H_t^{(m)}(\theta_t^{(m)}, z_t^{(m)}) - H_t^{(m)}(\tilde{\theta}^{(m-1)}, \tilde{z}^{(m-1)}) + H^{(m)}(\tilde{\theta}^{(m-1)}, \tilde{z}^{(m-1)})\rangle
$$
$$
+ 2\langle z_t^{(m)}, C^{-1}A(\theta_{t+1}^{(m)} - \theta_t^{(m)})\rangle, \tag{14}
$$

where (i) uses the assumption that $R_w \geq 2\|C^{-1}\|\|A\|R_\theta$ (i.e., $C^{-1}(b + A\theta_t^{(m)})$ is in the ball with radius $R_w$) and the fact that $\Pi_{R_w}$ is 1-Lipschitz. For the last term of eq. (14), it can be bounded as

$$
2\langle z_t^{(m)}, C^{-1}A(\theta_{t+1}^{(m)} - \theta_t^{(m)})\rangle \leq \frac{\lambda_C}{2}\beta\|z_t^{(m)}\|^2 + \frac{2}{\lambda_C}\frac{1}{\beta}\|C^{-1}A(\theta_{t+1}^{(m)} - \theta_t^{(m)})\|^2.
$$

554    Substituting the above inequality, Lemma J.4 and Lemma J.5 into eq. (14), we obtain that

$$
\|z_{t+1}^{(m)}\|^2 \leq \|z_t^{(m)}\|^2 + 2\beta^2 H_{\text{VR}}^2 + \left(\alpha^2 + \frac{2\alpha^2}{\lambda_C}\frac{1}{\beta}\right) \cdot 2\left(\rho_{\max}\frac{1+\gamma}{\min|\lambda(C)|}\right)^2 G_{\text{VR}}^2 + \frac{\lambda_C}{2}\beta\|z_t^{(m)}\|^2
$$
$$
+ 2\beta\langle z_t^{(m)}, H_t^{(m)}(\theta_t^{(m)}, z_t^{(m)}) - H_t^{(m)}(\tilde{\theta}^{(m-1)}, \tilde{z}^{(m-1)}) + H^{(m)}(\tilde{\theta}^{(m-1)}, \tilde{z}^{(m-1)})\rangle. \tag{15}
$$

555    Next, we bound the inner product term in the above inequality. Notice that $z_t^{(m)} \in \mathcal{F}_{m,t-1}$ and by
556    i.i.d. sampling we have $\mathbb{E}_{m,t-1}\bar{A}_t^{(m)} = \bar{A}$. Therefore,

$$
\mathbb{E}_{m,0}\langle z_t^{(m)}, H_t^{(m)}(\theta_t^{(m)}, z_t^{(m)}) - H_t^{(m)}(\tilde{\theta}^{(m-1)}, \tilde{z}^{(m-1)}) + H^{(m)}(\tilde{\theta}^{(m-1)}, \tilde{z}^{(m-1)})\rangle
$$
$$
= \mathbb{E}_{m,0}\langle z_t^{(m)}, H_t^{(m)}(\theta_t^{(m)}, z_t^{(m)})\rangle
$$
$$
= \mathbb{E}_{m,0}\langle z_t^{(m)}, \bar{A}_t^{(m)}\theta_t^{(m)} + \bar{b}_t^{(m)} + C_t^{(m)}z_t^{(m)}\rangle
$$
$$
= \mathbb{E}_{m,0}\langle z_t^{(m)}, \mathbb{E}_{m,t-1}(\bar{A}_t^{(m)})\theta_t^{(m)} + \mathbb{E}_{m,t-1}(\bar{b}^{(m)})\rangle + \mathbb{E}_{m,0}\langle z_t^{(m)}, \mathbb{E}_{m,t-1}(C_t^{(m)} - C)z_t^{(m)}\rangle + \mathbb{E}_{m,0}\langle z_t^{(m)}, Cz_t^{(m)}\rangle
$$
$$
= \mathbb{E}_{m,0}\langle z_t^{(m)}, Cz_t^{(m)}\rangle
$$
$$
\leq -\frac{\lambda_C}{2}\mathbb{E}_{m,0}\|z_t^{(m)}\|^2
$$

557    where the last inequality utilizes the negative definiteness of $C$ (recall that $\lambda_C := -\lambda_{\max}(C + C^\top)$).
558    Substituting the above inequality into eq. (15) (after taking expectation) yields that

$$
\mathbb{E}_{m,0}\|z_{t+1}^{(m)}\|^2 \leq \mathbb{E}_{m,0}\|z_t^{(m)}\|^2 + 2\beta^2 H_{\text{VR}}^2 + \left(\alpha^2 + \frac{2\alpha^2}{\lambda_C}\frac{1}{\beta}\right) \cdot 2\left(\rho_{\max}\frac{1+\gamma}{\min|\lambda(C)|}\right)^2 G_{\text{VR}}^2
$$
$$
- \frac{\lambda_C}{2}\beta\mathbb{E}_{m,0}\|z_t^{(m)}\|^2.
$$

559    Summing the above inequality over one batch yields that

$$
\mathbb{E}_{m,0}\|z_M^{(m)}\|^2 \leq \mathbb{E}_{m,0}\|z_0^{(m)}\|^2 + 2\beta^2 M H_{\text{VR}}^2 + \left(\alpha^2 + \frac{2\alpha^2}{\lambda_C}\frac{1}{\beta}\right)M \cdot 2\left(\rho_{\max}\frac{1+\gamma}{\min|\lambda(C)|}\right)^2 G_{\text{VR}}^2
$$
$$
- \frac{\lambda_C}{2}\beta \sum_{t=0}^{M-1} \mathbb{E}_{m,0}\|z_t^{(m)}\|^2.
$$

560    Re-arranging the above inequality and omitting $\mathbb{E}_{m,0}\|z_M^{(m)}\|^2$ further yields that

$$
\frac{\lambda_C}{2}\beta M \mathbb{E}_{m,0}\|\tilde{z}^{(m)}\|^2 \leq \|\tilde{z}^{(m-1)}\|^2 + 2\beta^2 M H_{\text{VR}}^2 + \left(\alpha^2 + \frac{2\alpha^2}{\lambda_C}\frac{1}{\beta}\right)M \cdot 2\left(\rho_{\max}\frac{1+\gamma}{\min|\lambda(C)|}\right)^2 G_{\text{VR}}^2.
$$

561    Dividing $\frac{\lambda_C}{2}\beta M$ on both sides of the above inequality, we obtain the following one-batch bound.

$$
\mathbb{E}\|\tilde{z}^{(m)}\|^2 \leq \frac{1}{M\beta} \cdot \frac{2}{\lambda_C}\mathbb{E}\|\tilde{z}^{(m-1)}\|^2 + \beta \cdot \frac{4}{\lambda_C}H_{\text{VR}}^2 + \left(\frac{\alpha^2}{\beta} + \frac{2}{\lambda_C}\frac{\alpha^2}{\beta^2}\right) \cdot \frac{4}{\lambda_C}\left(\rho_{\max}\frac{1+\gamma}{\min|\lambda(C)|}\right)^2 G_{\text{VR}}^2.
$$

562  Finally, we recursively unroll the above inequality and obtain

$$\mathbb{E}\|\tilde{z}^{(m)}\|^2 \le \big(\frac{1}{M\beta}\cdot\frac{2}{\lambda_C}\big)^m \mathbb{E}\|\tilde{z}^{(0)}\|^2$$
$$+\frac{1}{1-\frac{1}{M\beta}\cdot\frac{2}{\lambda_C}}\Big[\beta\cdot\frac{24}{\lambda_C}H_{\mathrm{VR}}^2 + \big(\frac{\alpha^2}{\beta}+\frac{2}{\lambda_C}\frac{\alpha^2}{\beta^2}\big)\cdot\frac{2}{\lambda_C}\big(\rho_{\max}\frac{1+\gamma}{\min|\lambda(C)|}\big)^2 G_{\mathrm{VR}}^2\Big].$$

563  To further simplify the above inequality, we assume $\beta < 1$ and $M\beta > \frac{4}{\lambda_C}$. Then, we have

$$\mathbb{E}\|\tilde{z}^{(m)}\|^2 \le \big(\frac{1}{M\beta}\cdot\frac{2}{\lambda_C}\big)^m \mathbb{E}\|\tilde{z}^{(0)}\|^2$$
$$+2\cdot\Big[\beta\cdot\frac{24}{\lambda_C}H_{\mathrm{VR}}^2 + \frac{\alpha^2}{\beta^2}\cdot\big(1+\frac{2}{\lambda_C}\big)\cdot\frac{2}{\lambda_C}\big(\rho_{\max}\frac{1+\gamma}{\min|\lambda(C)|}\big)^2 G_{\mathrm{VR}}^2\Big].$$

564  $\square$

565  **Lemma D.3** (Preliminary Bound for $\sum_{t=0}^{M-1}\|z_t^{(m)}\|^2$). *Under the same assumptions as those of*
566  *Theorem 3.1, choose the learning rate $\beta$ and the batch size $M$ such that $\beta < 1$ and*

$$\frac{\lambda_C}{2}\beta - 10\beta^2 - 10\gamma^2\rho_{\max}^2\big(\alpha^2+\frac{2\alpha^2}{\lambda_C}\frac{1}{\beta}\big)\cdot\big(\rho_{\max}\frac{1+\gamma}{\min|\lambda(C)|}\big)^2 \ge \frac{\lambda_C}{3}\beta. \qquad (16)$$

567  *Then the following preliminary bound holds.*

$$\frac{\lambda_C}{3}\beta\sum_{t=0}^{M-1}\mathbb{E}_{m,0}\|z_t^{(m)}\|^2$$
$$\le\Big[1+\Big[10\beta^2 + 10\gamma^2\rho_{\max}^2\big(1+\frac{2}{\lambda_C}\big)\cdot\big(\rho_{\max}\frac{1+\gamma}{\min|\lambda(C)|}\big)^2\frac{\alpha^2}{\beta}\Big]M\Big]\mathbb{E}_{m,0}\|\tilde{z}^{(m-1)}\|^2$$
$$+10(1+\gamma)^2\rho_{\max}^2\cdot\Big[\big(1+\frac{\gamma\rho_{\max}}{\min|\lambda(C)|}\big)^2\big(1+\frac{2}{\lambda_C}\big)\cdot\big(\rho_{\max}\frac{1+\gamma}{\min|\lambda(C)|}\big)^2\frac{\alpha^2}{\beta}$$
$$+\big(1+\frac{1}{\min|\lambda(C)|}\big)^2\beta^2\Big]\Big(\sum_{t=0}^{M-1}\mathbb{E}_{m,0}\|\theta_t^{(m)}-\theta^*\|^2 + M\mathbb{E}_{m,0}\|\tilde{\theta}^{(m-1)}-\theta^*\|^2\Big)$$
$$+10K_2\beta^2 + 10\big(\rho_{\max}\frac{1+\gamma}{\min|\lambda(C)|}\big)^2 K_1\big(1+\frac{2}{\lambda_C}\big)\frac{\alpha^2}{\beta},$$

568  *where $K_1$ is specified in eq.(26) in Lemma E.1.*

569  *Proof.* Following the proof of Lemma D.2, the one-step update of $z_t^{(m)}$ implies that

$$\|z_{t+1}^{(m)}\|^2 \le \|z_t^{(m)}\|^2 + 2\beta^2\|H_t^{(m)}(\theta_t^{(m)},z_t^{(m)}) - H_t^{(m)}(\tilde{\theta}^{(m-1)},\tilde{z}^{(m-1)}) + H^{(m)}(\tilde{\theta}^{(m-1)},\tilde{z}^{(m-1)})\|^2$$
$$+2\|C^{-1}A(\theta_{t+1}^{(m)}-\theta_t^{(m)})\|^2 + 2\langle z_t^{(m)}, C^{-1}A(\theta_{t+1}^{(m)}-\theta_t^{(m)})\rangle$$
$$+2\beta\langle z_t^{(m)}, H_t^{(m)}(\theta_t^{(m)},z_t^{(m)}) - H_t^{(m)}(\tilde{\theta}^{(m-1)},\tilde{z}^{(m-1)}) + H^{(m)}(\tilde{\theta}^{(m-1)},\tilde{z}^{(m-1)})\rangle$$
$$\le\|z_t^{(m)}\|^2 + 2\beta^2\|H_t^{(m)}(\theta_t^{(m)},z_t^{(m)}) - H_t^{(m)}(\tilde{\theta}^{(m-1)},\tilde{z}^{(m-1)}) + H^{(m)}(\tilde{\theta}^{(m-1)},\tilde{z}^{(m-1)})\|^2$$
$$+2\|C^{-1}A(\theta_{t+1}^{(m)}-\theta_t^{(m)})\|^2 + \frac{\lambda_C}{2}\beta\|z_t^{(m)}\|^2 + \frac{2}{\lambda_C}\frac{1}{\beta}\|C^{-1}A(\theta_{t+1}-\theta_t)\|^2.$$
$$+2\beta\langle z_t^{(m)}, H_t^{(m)}(\theta_t^{(m)},z_t^{(m)}) - H_t^{(m)}(\tilde{\theta}^{(m-1)},\tilde{z}^{(m-1)}) + H^{(m)}(\tilde{\theta}^{(m-1)},\tilde{z}^{(m-1)})\rangle$$
$$\le\|z_t^{(m)}\|^2 + \frac{\lambda_C}{2}\beta\|z_t^{(m)}\|^2 + 2\beta^2\|H_t^{(m)}(\theta_t^{(m)},z_t^{(m)}) - H_t^{(m)}(\tilde{\theta}^{(m-1)},\tilde{z}^{(m-1)}) + H^{(m)}(\tilde{\theta}^{(m-1)},\tilde{z}^{(m-1)})\|^2$$
$$+\big(\alpha^2+\frac{2\alpha^2}{\lambda_C}\frac{1}{\beta}\big)\cdot 2\big(\rho_{\max}\frac{1+\gamma}{\min|\lambda(C)|}\big)^2\|G_t^{(m)}(\theta_t^{(m)},z_t^{(m)})$$
$$-G_t^{(m)}(\tilde{\theta}^{(m-1)},\tilde{z}^{(m-1)}) + G^{(m)}(\tilde{\theta}^{(m-1)},\tilde{z}^{(m-1)})\|^2$$

$$+ 2\beta\langle z_t^{(m)}, H_t^{(m)}(\theta_t^{(m)}, z_t^{(m)}) - H_t^{(m)}(\tilde{\theta}^{(m-1)}, \tilde{z}^{(m-1)}) + H^{(m)}(\tilde{\theta}^{(m-1)}, \tilde{z}^{(m-1)})\rangle. \tag{17}$$

570 For the last inner product term, we still have

$$\mathbb{E}_{m,0}\langle z_t^{(m)}, H_t^{(m)}(\theta_t^{(m)}, z_t^{(m)}) - H_t^{(m)}(\tilde{\theta}^{(m-1)}, \tilde{z}^{(m-1)}) + H^{(m)}(\tilde{\theta}^{(m-1)}, \tilde{z}^{(m-1)})\rangle \le -\frac{\lambda_C}{2}\mathbb{E}_{m,0}\|z_t^{(m)}\|^2.$$

571 Instead of bounding the variance term in eq. (17) using Lemma J.4 and Lemma J.5, we apply Lemma
572 E.1 and Lemma E.2 to get a refined bound. Combining these together, we obtain from eq. (17) that

$$\mathbb{E}_{m,0}\|z_{t+1}^{(m)}\|^2$$

$$\le \mathbb{E}_{m,0}\|z_t^{(m)}\|^2 - \frac{\lambda_C}{2}\beta\mathbb{E}_{m,0}\|z_t^{(m)}\|^2 + 2\beta^2\Big[5\big(\mathbb{E}_{m,0}\|z_t^{(m)}\|^2 + \mathbb{E}_{m,0}\|\tilde{z}^{(m-1)}\|^2\big) + \frac{5K_2}{M}$$

$$+ 5(1+\gamma)^2\rho_{\max}^2\big(1 + \frac{1}{\min|\lambda(C)|}\big)^2\big(\mathbb{E}_{m,0}\|\theta_t^{(m)} - \theta^*\|^2 + \mathbb{E}_{m,0}\|\tilde{\theta}^{(m-1)} - \theta^*\|^2\big)\Big]$$

$$+ \big(\alpha^2 + \frac{2\alpha^2}{\lambda_C}\frac{1}{\beta}\big)\cdot 2\big(\rho_{\max}\frac{1+\gamma}{\min|\lambda(C)|}\big)^2\Big[5\gamma^2\rho_{\max}^2\big(\mathbb{E}_{m,0}\|z_t^{(m)}\|^2 + \mathbb{E}_{m,0}\|\tilde{z}^{(m-1)}\|^2\big) + \frac{5K_1}{M}$$

$$+ 5(1+\gamma)^2\rho_{\max}^2\big(1 + \frac{\gamma\rho_{\max}}{\min|\lambda(C)|}\big)^2\big(\mathbb{E}_{m,0}\|\theta_t^{(m)} - \theta^*\|^2 + \mathbb{E}_{m,0}\|\tilde{\theta}^{(m-1)} - \theta^*\|^2\big)\Big].$$

573 Re-arranging the above inequality yields that

$$\mathbb{E}_{m,0}\|z_{t+1}^{(m)}\|^2$$

$$\le \mathbb{E}_{m,0}\|z_t^{(m)}\|^2 - \Big[\frac{\lambda_C}{2}\beta - 10\beta^2 - 10\gamma^2\rho_{\max}^2\big(\alpha^2 + \frac{2\alpha^2}{\lambda_C}\frac{1}{\beta}\big)\cdot\big(\rho_{\max}\frac{1+\gamma}{\min|\lambda(C)|}\big)^2\Big]\mathbb{E}_{m,0}\|z_t^{(m)}\|^2$$

$$+ \Big[10\beta^2 + 10\gamma^2\rho_{\max}^2\big(\alpha^2 + \frac{2\alpha^2}{\lambda_C}\frac{1}{\beta}\big)\cdot\big(\rho_{\max}\frac{1+\gamma}{\min|\lambda(C)|}\big)^2\Big]\mathbb{E}_{m,0}\|\tilde{z}^{(m-1)}\|^2$$

$$+ 10(1+\gamma)^2\rho_{\max}^2\cdot\Big[\big(1 + \frac{\gamma\rho_{\max}}{\min|\lambda(C)|}\big)^2\big(\alpha^2 + \frac{2\alpha^2}{\lambda_C}\frac{1}{\beta}\big)\cdot\big(\rho_{\max}\frac{1+\gamma}{\min|\lambda(C)|}\big)^2$$

$$+ \big(1 + \frac{1}{\min|\lambda(C)|}\big)^2\beta^2\Big]\big(\mathbb{E}_{m,0}\|\theta_t^{(m)} - \theta^*\|^2 + \mathbb{E}_{m,0}\|\tilde{\theta}^{(m-1)} - \theta^*\|^2\big)$$

$$+ \frac{1}{M}\cdot\Big[10K_2\beta^2 + 10\big(\rho_{\max}\frac{1+\gamma}{\min|\lambda(C)|}\big)^2 K_1\big(\alpha^2 + \frac{2\alpha^2}{\lambda_C}\frac{1}{\beta}\big)\Big].$$

574 Telescoping the above inequality over one batch yields that

$$\Big[\frac{\lambda_C}{2}\beta - 10\beta^2 - 10\gamma^2\rho_{\max}^2\big(\alpha^2 + \frac{2\alpha^2}{\lambda_C}\frac{1}{\beta}\big)\cdot\big(\rho_{\max}\frac{1+\gamma}{\min|\lambda(C)|}\big)^2\Big]\sum_{t=0}^{M-1}\mathbb{E}_{m,0}\|z_t^{(m)}\|^2$$

$$\le \Big[1 + \Big[10\beta^2 + 10\gamma^2\rho_{\max}^2\big(\alpha^2 + \frac{2\alpha^2}{\lambda_C}\frac{1}{\beta}\big)\cdot\big(\rho_{\max}\frac{1+\gamma}{\min|\lambda(C)|}\big)^2\Big]M\Big]\mathbb{E}_{m,0}\|\tilde{z}^{(m-1)}\|^2$$

$$+ 10(1+\gamma)^2\rho_{\max}^2\cdot\Big[\big(1 + \frac{\gamma\rho_{\max}}{\min|\lambda(C)|}\big)^2\big(\alpha^2 + \frac{2\alpha^2}{\lambda_C}\frac{1}{\beta}\big)\cdot\big(\rho_{\max}\frac{1+\gamma}{\min|\lambda(C)|}\big)^2$$

$$+ \big(1 + \frac{1}{\min|\lambda(C)|}\big)^2\beta^2\Big]\big(\sum_{t=0}^{M-1}\mathbb{E}_{m,0}\|\theta_t^{(m)} - \theta^*\|^2 + M\mathbb{E}_{m,0}\|\tilde{\theta}^{(m-1)} - \theta^*\|^2\big)$$

$$+ 10K_2\beta^2 + 10\big(\rho_{\max}\frac{1+\gamma}{\min|\lambda(C)|}\big)^2 K_1\big(\alpha^2 + \frac{2\alpha^2}{\lambda_C}\frac{1}{\beta}\big).$$

To further simplify the above inequality, we let $\beta < 1$ and

$$\frac{\lambda_C}{2}\beta - 10\beta^2 - 10\gamma^2\rho_{\max}^2\big(\alpha^2 + \frac{2\alpha^2}{\lambda_C}\frac{1}{\beta}\big)\cdot\big(\rho_{\max}\frac{1+\gamma}{\min|\lambda(C)|}\big)^2 \ge \frac{\lambda_C}{3}\beta.$$

575 Then, we finally obtain that

$$\frac{\lambda_C}{3}\beta\sum_{t=0}^{M-1}\mathbb{E}_{m,0}\|z_t^{(m)}\|^2$$

$$\leq \Big[1 + \Big[10\beta^2 + 10\gamma^2\rho_{\max}^2\big(1 + \frac{2}{\lambda_C}\big)\cdot\big(\rho_{\max}\frac{1+\gamma}{\min|\lambda(C)|}\big)^2\frac{\alpha^2}{\beta}\Big]M\Big]\mathbb{E}_{m,0}\|\tilde{z}^{(m-1)}\|^2$$

$$+ 10(1+\gamma)^2\rho_{\max}^2\cdot\Big[\big(1 + \frac{\gamma\rho_{\max}}{\min|\lambda(C)|}\big)^2\big(1 + \frac{2}{\lambda_C}\big)\cdot\big(\rho_{\max}\frac{1+\gamma}{\min|\lambda(C)|}\big)^2\frac{\alpha^2}{\beta}$$

$$+ \big(1 + \frac{1}{\min|\lambda(C)|}\big)^2\beta^2\Big]\big(\sum_{t=0}^{M-1}\mathbb{E}_{m,0}\|\theta_t^{(m)} - \theta^*\|^2 + M\mathbb{E}_{m,0}\|\tilde{\theta}^{(m-1)} - \theta^*\|^2\big)$$

$$+ 10K_2\beta^2 + 10\big(\rho_{\max}\frac{1+\gamma}{\min|\lambda(C)|}\big)^2K_1\big(1 + \frac{2}{\lambda_C}\big)\frac{\alpha^2}{\beta}.$$

□

**Lemma D.4** (Preliminary bound for $\sum_{t=0}^{M-1}\mathbb{E}_{m,0}\|\theta_t^{(m)} - \theta^*\|^2$). *Under the same assumptions as those of Theorem 3.1, Lemma D.3, Lemma D.1, and Lemma D.2, choose the learning rates $\alpha, \beta$ and the batch size $M$ such that*

$$D := \frac{12}{\lambda_{\widehat{A}}}\Big\{\frac{1}{\alpha M} + \alpha\cdot 5(1+\gamma)^2\rho_{\max}^2\big(1 + \frac{\gamma\rho_{\max}}{\min|\lambda(C)|}\big)^2 + \frac{\alpha^2}{\beta^2}\cdot C_1 + \beta\cdot C_2\Big\} < 1, \quad (18)$$

*and*

$$E := \frac{1}{M\beta}\cdot\frac{2}{\lambda_C} < 1, \quad (19)$$

*where*

$$C_1 = \frac{2\rho_{\max}^2\gamma^2}{\lambda_{\widehat{A}}}\frac{3}{\lambda_C}\cdot 10(1+\gamma)^2\rho_{\max}^2\cdot\big(1 + \frac{\gamma\rho_{\max}}{\min|\lambda(C)|}\big)^2\big(1 + \frac{2}{\lambda_C}\big)\cdot\big(\rho_{\max}\frac{1+\gamma}{\min|\lambda(C)|}\big)^2, \quad (20)$$

*and*

$$C_2 = \frac{2\rho_{\max}^2\gamma^2}{\lambda_{\widehat{A}}}\frac{3}{\lambda_C}\cdot 10(1+\gamma)^2\rho_{\max}^2\cdot\big(1 + \frac{1}{\min|\lambda(C)|}\big)^2. \quad (21)$$

*Then, the following preliminary bound holds.*

$$\mathbb{E}\|\tilde{\theta}^{(m)} - \theta^*\|^2$$

$$\leq D^m\cdot\mathbb{E}\|\tilde{\theta}^{(0)} - \theta^*\|^2$$

$$+ \frac{12}{\lambda_{\widehat{A}}}\Big\{\frac{2\rho_{\max}^2\gamma^2}{\lambda_{\widehat{A}}}\frac{3}{\lambda_C}\Big[\frac{1}{\beta M} + \beta\cdot 10 + \frac{\alpha^2}{\beta^2}\cdot 10\gamma^2\rho_{\max}^2\big(1 + \frac{2}{\lambda_C}\big)\cdot\big(\rho_{\max}\frac{1+\gamma}{\min|\lambda(C)|}\big)^2\Big] + \alpha\cdot 5\gamma^2\rho_{\max}^2\Big\}$$

$$\times\Big\{\frac{D^m - E^m}{D - E}\mathbb{E}\|\tilde{z}^{(0)}\|^2 + \frac{2}{1-D}\cdot\Big[\beta\cdot\frac{24}{\lambda_C}H_{VR}^2 + \frac{\alpha^2}{\beta^2}\cdot\big(1 + \frac{2}{\lambda_C}\big)\cdot\frac{2}{\lambda_C}\big(\rho_{\max}\frac{1+\gamma}{\min|\lambda(C)|}\big)^2G_{VR}^2\Big]\Big\}$$

$$+ \frac{1}{1-D}\Big\{\frac{\beta}{M}\cdot\frac{12}{\lambda_{\widehat{A}}}\frac{60\rho_{\max}^2\gamma^2}{\lambda_{\widehat{A}}\lambda_C}\cdot K_2 + \frac{\alpha^2}{\beta^2}\frac{1}{M}\cdot\frac{12}{\lambda_{\widehat{A}}}\frac{60\rho_{\max}^2\gamma^2}{\lambda_{\widehat{A}}\lambda_C}\cdot\big(\rho_{\max}\frac{1+\gamma}{\min|\lambda(C)|}\big)^2K_1\big(1 + \frac{2}{\lambda_C}\big) + \frac{\alpha}{M}\cdot\frac{60K_1}{\lambda_{\widehat{A}}}\Big\}$$

*where $K_1$ is specified in eq.(26) in Lemma E.1, and $K_2$ is specified in eq.(32) in Lemma E.2.*

*Proof.* First, recall that Lemma D.1 gives the following preliminary bound for $\sum_{t=0}^{M-1}\|\theta_t^{(m)} - \theta^*\|^2$.

$$\frac{\lambda_{\widehat{A}}}{6}\alpha\sum_{t=0}^{M-1}\mathbb{E}_{m,0}\|\theta_t^{(m)} - \theta^*\|^2$$

$$\leq\Big[1 + \alpha^2 M\cdot 5(1+\gamma)^2\rho_{\max}^2\big(1 + \frac{\gamma\rho_{\max}}{\min|\lambda(C)|}\big)^2\Big]\mathbb{E}_{m,0}\|\tilde{\theta}^{(m-1)} - \theta^*\|^2 + \alpha^2\cdot 5K_1$$

$$+ \alpha\cdot\frac{2\rho_{\max}^2\gamma^2}{\lambda_{\widehat{A}}}\sum_{t=0}^{M-1}\mathbb{E}_{m,0}\|z_t^{(m)}\|^2 + \alpha^2 M\cdot 5\gamma^2\rho_{\max}^2\mathbb{E}_{m,0}\|\tilde{z}^{(m-1)}\|^2.$$

Then, we combine the above preliminary bound with Lemma D.3 and obtain that

$$\frac{\lambda_{\widehat{A}}}{6}\alpha\sum_{t=0}^{M-1}\mathbb{E}_{m,0}\|\theta_t^{(m)} - \theta^*\|^2$$

$$\leq \Big[1+\alpha^2 M\cdot 5(1+\gamma)^2\rho_{\max}^2\big(1+\frac{\gamma\rho_{\max}}{\min|\lambda(C)|}\big)^2\Big]\mathbb{E}_{m,0}\|\tilde{\theta}^{(m-1)}-\theta^*\|^2+\alpha^2\cdot 5K_1$$

$$+\frac{\alpha}{\beta}\cdot\frac{2\rho_{\max}^2\gamma^2}{\lambda_{\widehat{A}}}\frac{3}{\lambda_C}\Big\{\Big[1+\Big[10\beta^2+10\gamma^2\rho_{\max}^2\big(1+\frac{2}{\lambda_C}\big)\cdot\big(\rho_{\max}\frac{1+\gamma}{\min|\lambda(C)|}\big)^2\frac{\alpha^2}{\beta}\Big]M\Big]\mathbb{E}_{m,0}\|\tilde{z}^{(m-1)}\|^2$$

$$+10(1+\gamma)^2\rho_{\max}^2\cdot\Big[\big(1+\frac{\gamma\rho_{\max}}{\min|\lambda(C)|}\big)^2\big(1+\frac{2}{\lambda_C}\big)\cdot\big(\rho_{\max}\frac{1+\gamma}{\min|\lambda(C)|}\big)^2\frac{\alpha^2}{\beta}$$

$$+\big(1+\frac{1}{\min|\lambda(C)|}\big)^2\beta^2\Big]\big(\sum_{t=0}^{M-1}\mathbb{E}_{m,0}\|\theta_t^{(m)}-\theta^*\|^2+M\mathbb{E}_{m,0}\|\tilde{\theta}^{(m-1)}-\theta^*\|^2\big)$$

$$+10K_2\beta^2+10\big(\rho_{\max}\frac{1+\gamma}{\min|\lambda(C)|}\big)^2K_1\big(1+\frac{2}{\lambda_C}\big)\frac{\alpha^2}{\beta}\Big\}$$

$$+\alpha^2 M\cdot 5\gamma^2\rho_{\max}^2\mathbb{E}_{m,0}\|\tilde{z}^{(m-1)}\|^2$$

$$=\Big[1+\alpha^2 M\cdot 5(1+\gamma)^2\rho_{\max}^2\big(1+\frac{\gamma\rho_{\max}}{\min|\lambda(C)|}\big)^2\Big]\mathbb{E}_{m,0}\|\tilde{\theta}^{(m-1)}-\theta^*\|^2+\alpha^2\cdot 5K_1$$

$$+\frac{\alpha}{\beta}\cdot\frac{2\rho_{\max}^2\gamma^2}{\lambda_{\widehat{A}}}\frac{3}{\lambda_C}\Big[1+\Big[10\beta^2+10\gamma^2\rho_{\max}^2\big(1+\frac{2}{\lambda_C}\big)\cdot\big(\rho_{\max}\frac{1+\gamma}{\min|\lambda(C)|}\big)^2\frac{\alpha^2}{\beta}\Big]M\Big]\mathbb{E}_{m,0}\|\tilde{z}^{(m-1)}\|^2$$

$$+\frac{\alpha}{\beta}\cdot\frac{2\rho_{\max}^2\gamma^2}{\lambda_{\widehat{A}}}\frac{3}{\lambda_C}\cdot 10(1+\gamma)^2\rho_{\max}^2\cdot\Big[\big(1+\frac{\gamma\rho_{\max}}{\min|\lambda(C)|}\big)^2\big(1+\frac{2}{\lambda_C}\big)\cdot\big(\rho_{\max}\frac{1+\gamma}{\min|\lambda(C)|}\big)^2\frac{\alpha^2}{\beta}$$

$$+\big(1+\frac{1}{\min|\lambda(C)|}\big)^2\beta^2\Big]\big(\sum_{t=0}^{M-1}\mathbb{E}_{m,0}\|\theta_t^{(m)}-\theta^*\|^2+M\mathbb{E}_{m,0}\|\tilde{\theta}^{(m-1)}-\theta^*\|^2\big)$$

$$+\frac{\alpha}{\beta}\cdot\frac{2\rho_{\max}^2\gamma^2}{\lambda_{\widehat{A}}}\frac{3}{\lambda_C}\cdot 10K_2\beta^2+\frac{\alpha}{\beta}\cdot\frac{2\rho_{\max}^2\gamma^2}{\lambda_{\widehat{A}}}\frac{3}{\lambda_C}\cdot 10\big(\rho_{\max}\frac{1+\gamma}{\min|\lambda(C)|}\big)^2K_1\big(1+\frac{2}{\lambda_C}\big)\frac{\alpha^2}{\beta}$$

$$+\alpha^2 M\cdot 5\gamma^2\rho_{\max}^2\mathbb{E}_{m,0}\|\tilde{z}^{(m-1)}\|^2$$

$$=\Big[1+\alpha^2 M\cdot 5(1+\gamma)^2\rho_{\max}^2\big(1+\frac{\gamma\rho_{\max}}{\min|\lambda(C)|}\big)^2\Big]\mathbb{E}_{m,0}\|\tilde{\theta}^{(m-1)}-\theta^*\|^2$$

$$+\Big\{\frac{\alpha}{\beta}\cdot\frac{2\rho_{\max}^2\gamma^2}{\lambda_{\widehat{A}}}\frac{3}{\lambda_C}\Big[1+\Big[10\beta^2+10\gamma^2\rho_{\max}^2\big(1+\frac{2}{\lambda_C}\big)\cdot\big(\rho_{\max}\frac{1+\gamma}{\min|\lambda(C)|}\big)^2\frac{\alpha^2}{\beta}\Big]M\Big]$$

$$+\alpha^2 M\cdot 5\gamma^2\rho_{\max}^2\Big\}\mathbb{E}_{m,0}\|\tilde{z}^{(m-1)}\|^2$$

$$+\frac{\alpha}{\beta}\cdot\frac{2\rho_{\max}^2\gamma^2}{\lambda_{\widehat{A}}}\frac{3}{\lambda_C}\cdot 10(1+\gamma)^2\rho_{\max}^2\cdot\Big[\big(1+\frac{\gamma\rho_{\max}}{\min|\lambda(C)|}\big)^2\big(1+\frac{2}{\lambda_C}\big)\cdot\big(\rho_{\max}\frac{1+\gamma}{\min|\lambda(C)|}\big)^2\frac{\alpha^2}{\beta}$$

$$+\big(1+\frac{1}{\min|\lambda(C)|}\big)^2\beta^2\Big]\big(\sum_{t=0}^{M-1}\mathbb{E}_{m,0}\|\theta_t^{(m)}-\theta^*\|^2+M\mathbb{E}_{m,0}\|\tilde{\theta}^{(m-1)}-\theta^*\|^2\big)$$

$$+\alpha\beta\cdot\frac{60\rho_{\max}^2\gamma^2}{\lambda_{\widehat{A}}\lambda_C}\cdot K_2+\frac{\alpha^3}{\beta^2}\cdot\frac{60\rho_{\max}^2\gamma^2}{\lambda_{\widehat{A}}\lambda_C}\cdot\big(\rho_{\max}\frac{1+\gamma}{\min|\lambda(C)|}\big)^2K_1\big(1+\frac{2}{\lambda_C}\big)+\alpha^2\cdot 5K_1,$$

587 where in the first equality we expand the curly bracket and in the last equality we combine and
588 re-arrange the terms. Then, we move the term $\sum_{t=0}^{M-1}\mathbb{E}_{m,0}\|\theta_t^{(m)}-\theta^*\|^2$ in the last equality to the
589 left-hand side and obtain that

$$\Big\{\frac{\lambda_{\widehat{A}}}{6}\alpha-\frac{\alpha}{\beta}\cdot\frac{2\rho_{\max}^2\gamma^2}{\lambda_{\widehat{A}}}\frac{3}{\lambda_C}\cdot 10(1+\gamma)^2\rho_{\max}^2\cdot\Big[\big(1+\frac{\gamma\rho_{\max}}{\min|\lambda(C)|}\big)^2\big(1+\frac{2}{\lambda_C}\big)\cdot\big(\rho_{\max}\frac{1+\gamma}{\min|\lambda(C)|}\big)^2\frac{\alpha^2}{\beta}$$

$$+\big(1+\frac{1}{\min|\lambda(C)|}\big)^2\beta^2\Big]\Big\}\sum_{t=0}^{M-1}\mathbb{E}_{m,0}\|\theta_t^{(m)}-\theta^*\|^2$$

$$\leq\Big\{\Big[1+\alpha^2 M\cdot 5(1+\gamma)^2\rho_{\max}^2\big(1+\frac{\gamma\rho_{\max}}{\min|\lambda(C)|}\big)^2\Big]$$

$$+\frac{\alpha}{\beta}\cdot\frac{2\rho_{\max}^2\gamma^2}{\lambda_{\widehat{A}}}\frac{3}{\lambda_C}\cdot 10(1+\gamma)^2\rho_{\max}^2\cdot\Big[\big(1+\frac{\gamma\rho_{\max}}{\min|\lambda(C)|}\big)^2\big(1+\frac{2}{\lambda_C}\big)\cdot\big(\rho_{\max}\frac{1+\gamma}{\min|\lambda(C)|}\big)^2\frac{\alpha^2}{\beta}$$

$$+ \Big(1 + \frac{1}{\min|\lambda(C)|}\Big)^2 \beta^2\Big] \cdot M\Big\} \mathbb{E}_{m,0}\|\tilde{\theta}^{(m-1)} - \theta^*\|^2$$

$$+ \Big\{\frac{\alpha}{\beta} \cdot \frac{2\rho_{\max}^2 \gamma^2}{\lambda_{\widehat{A}}} \frac{3}{\lambda_C}\Big[1 + \Big[10\beta^2 + 10\gamma^2 \rho_{\max}^2\big(1 + \frac{2}{\lambda_C}\big) \cdot \big(\rho_{\max}\frac{1+\gamma}{\min|\lambda(C)|}\big)^2 \frac{\alpha^2}{\beta}\Big]M\Big]$$

$$+ \alpha^2 M \cdot 5\gamma^2 \rho_{\max}^2\Big\} \mathbb{E}_{m,0}\|\tilde{z}^{(m-1)}\|^2$$

$$+ \alpha\beta \cdot \frac{60\rho_{\max}^2 \gamma^2}{\lambda_{\widehat{A}}\lambda_C} \cdot K_2 + \frac{\alpha^3}{\beta^2} \cdot \frac{60\rho_{\max}^2 \gamma^2}{\lambda_{\widehat{A}}\lambda_C} \cdot \big(\rho_{\max}\frac{1+\gamma}{\min|\lambda(C)|}\big)^2 K_1\big(1 + \frac{2}{\lambda_C}\big) + \alpha^2 \cdot 5K_1 \quad (22)$$

590   Now we define the following constants to further simplify the result above.

591   • $C_1 = \frac{2\rho_{\max}^2 \gamma^2}{\lambda_{\widehat{A}}} \frac{3}{\lambda_C} \cdot 10(1+\gamma)^2 \rho_{\max}^2 \cdot \big(1 + \frac{\gamma\rho_{\max}}{\min|\lambda(C)|}\big)^2 \big(1 + \frac{2}{\lambda_C}\big) \cdot \big(\rho_{\max}\frac{1+\gamma}{\min|\lambda(C)|}\big)^2$,

592   • $C_2 = \frac{2\rho_{\max}^2 \gamma^2}{\lambda_{\widehat{A}}} \frac{3}{\lambda_C} \cdot 10(1+\gamma)^2 \rho_{\max}^2 \cdot \big(1 + \frac{1}{\min|\lambda(C)|}\big)^2$.

593   Then, eq.(22) can be rewritten as

$$\Big\{\frac{\lambda_{\widehat{A}}}{6}\alpha - \alpha \cdot \big(\frac{\alpha^2}{\beta^2} \cdot C_1 + \beta \cdot C_2\big)\Big\} \sum_{t=0}^{M-1} \mathbb{E}_{m,0}\|\theta_t^{(m)} - \theta^*\|^2$$

$$\leq \Big\{1 + \alpha^2 M \cdot 5(1+\gamma)^2 \rho_{\max}^2\big(1 + \frac{\gamma\rho_{\max}}{\min|\lambda(C)|}\big)^2 + \alpha M \cdot \big(\frac{\alpha^2}{\beta^2} \cdot C_1 + \beta \cdot C_2\big)\Big\} \mathbb{E}_{m,0}\|\tilde{\theta}^{(m-1)} - \theta^*\|^2$$

$$+ \Big\{\frac{\alpha}{\beta} \cdot \frac{2\rho_{\max}^2 \gamma^2}{\lambda_{\widehat{A}}} \frac{3}{\lambda_C}\Big[1 + \Big[10\beta^2 + 10\gamma^2 \rho_{\max}^2\big(1 + \frac{2}{\lambda_C}\big) \cdot \big(\rho_{\max}\frac{1+\gamma}{\min|\lambda(C)|}\big)^2 \frac{\alpha^2}{\beta}\Big]M\Big] + \alpha^2 M \cdot 5\gamma^2 \rho_{\max}^2\Big\} \mathbb{E}_{m,0}\|\tilde{z}^{(m-1)}\|^2$$

$$+ \alpha\beta \cdot \frac{60\rho_{\max}^2 \gamma^2}{\lambda_{\widehat{A}}\lambda_C} \cdot K_2 + \frac{\alpha^3}{\beta^2} \cdot \frac{60\rho_{\max}^2 \gamma^2}{\lambda_{\widehat{A}}\lambda_C} \cdot \big(\rho_{\max}\frac{1+\gamma}{\min|\lambda(C)|}\big)^2 K_1\big(1 + \frac{2}{\lambda_C}\big) + \alpha^2 \cdot 5K_1.$$

594   Apply Lemma D.2 to the inequality above and taking total expectation on both sides, we obtain that

$$\Big\{\frac{\lambda_{\widehat{A}}}{6}\alpha - \alpha \cdot \big(\frac{\alpha^2}{\beta^2} \cdot C_1 + \beta \cdot C_2\big)\Big\} \sum_{t=0}^{M-1} \mathbb{E}\|\theta_t^{(m)} - \theta^*\|^2$$

$$\leq \Big\{1 + \alpha^2 M \cdot 5(1+\gamma)^2 \rho_{\max}^2\big(1 + \frac{\gamma\rho_{\max}}{\min|\lambda(C)|}\big)^2 + \alpha M \cdot \big(\frac{\alpha^2}{\beta^2} \cdot C_1 + \beta \cdot C_2\big)\Big\} \mathbb{E}\|\tilde{\theta}^{(m-1)} - \theta^*\|^2$$

$$+ \Big\{\frac{\alpha}{\beta} \cdot \frac{2\rho_{\max}^2 \gamma^2}{\lambda_{\widehat{A}}} \frac{3}{\lambda_C}\Big[1 + \Big[10\beta^2 + 10\gamma^2 \rho_{\max}^2\big(1 + \frac{2}{\lambda_C}\big) \cdot \big(\rho_{\max}\frac{1+\gamma}{\min|\lambda(C)|}\big)^2 \frac{\alpha^2}{\beta}\Big]M\Big] + \alpha^2 M \cdot 5\gamma^2 \rho_{\max}^2\Big\}$$

$$\times \Big\{\big(\frac{1}{M\beta} \cdot \frac{2}{\lambda_C}\big)^m \mathbb{E}\|\tilde{z}^{(0)}\|^2 + 2 \cdot \Big[\beta \cdot \frac{24}{\lambda_C}H_{\mathrm{VR}}^2 + \frac{\alpha^2}{\beta^2} \cdot \big(1 + \frac{2}{\lambda_C}\big) \cdot \frac{2}{\lambda_C}\big(\rho_{\max}\frac{1+\gamma}{\min|\lambda(C)|}\big)^2 G_{\mathrm{VR}}^2\Big]\Big\}$$

$$+ \alpha\beta \cdot \frac{60\rho_{\max}^2 \gamma^2}{\lambda_{\widehat{A}}\lambda_C} \cdot K_2 + \frac{\alpha^3}{\beta^2} \cdot \frac{60\rho_{\max}^2 \gamma^2}{\lambda_{\widehat{A}}\lambda_C} \cdot \big(\rho_{\max}\frac{1+\gamma}{\min|\lambda(C)|}\big)^2 K_1\big(1 + \frac{2}{\lambda_C}\big) + \alpha^2 \cdot 5K_1.$$

595   Let $\frac{\lambda_{\widehat{A}}}{6}\alpha - \alpha \cdot \big(\frac{\alpha^2}{\beta^2} \cdot C_1 + \beta \cdot C_2\big) \geq \frac{\lambda_{\widehat{A}}}{12}\alpha$ and divide $\frac{\lambda_{\widehat{A}}}{12}\alpha M$ on both sides of the above inequality.

596   Then, apply Jensen's inequality to the left-hand side of the inequality above, we obtain that

$$\mathbb{E}\|\tilde{\theta}^{(m)} - \theta^*\|^2$$

$$\leq \frac{12}{\lambda_{\widehat{A}}}\Big\{\frac{1}{\alpha M} + \alpha \cdot 5(1+\gamma)^2 \rho_{\max}^2\big(1 + \frac{\gamma\rho_{\max}}{\min|\lambda(C)|}\big)^2 + \frac{\alpha^2}{\beta^2} \cdot C_1 + \beta \cdot C_2\Big\} \mathbb{E}\|\tilde{\theta}^{(m-1)} - \theta^*\|^2$$

$$+ \frac{12}{\lambda_{\widehat{A}}}\Big\{\frac{1}{\beta M} \cdot \frac{2\rho_{\max}^2 \gamma^2}{\lambda_{\widehat{A}}} \frac{3}{\lambda_C}\Big[1 + \Big[10\beta^2 + 10\gamma^2 \rho_{\max}^2\big(1 + \frac{2}{\lambda_C}\big) \cdot \big(\rho_{\max}\frac{1+\gamma}{\min|\lambda(C)|}\big)^2 \frac{\alpha^2}{\beta}\Big]M\Big] + \alpha \cdot 5\gamma^2 \rho_{\max}^2\Big\}$$

$$\times \Big\{\big(\frac{1}{M\beta} \cdot \frac{2}{\lambda_C}\big)^{m-1} \mathbb{E}\|\tilde{z}^{(0)}\|^2 + 2 \cdot \Big[\beta \cdot \frac{24}{\lambda_C}H_{\mathrm{VR}}^2 + \frac{\alpha^2}{\beta^2} \cdot \big(1 + \frac{2}{\lambda_C}\big) \cdot \frac{2}{\lambda_C}\big(\rho_{\max}\frac{1+\gamma}{\min|\lambda(C)|}\big)^2 G_{\mathrm{VR}}^2\Big]\Big\}$$

$$+ \frac{\beta}{M} \cdot \frac{12}{\lambda_{\widehat{A}}}\frac{60\rho_{\max}^2 \gamma^2}{\lambda_{\widehat{A}}\lambda_C} \cdot K_2 + \frac{\alpha^2}{\beta^2}\frac{1}{M} \cdot \frac{12}{\lambda_{\widehat{A}}}\frac{60\rho_{\max}^2 \gamma^2}{\lambda_{\widehat{A}}\lambda_C} \cdot \big(\rho_{\max}\frac{1+\gamma}{\min|\lambda(C)|}\big)^2 K_1\big(1 + \frac{2}{\lambda_C}\big) + \frac{\alpha}{M} \cdot \frac{60K_1}{\lambda_{\widehat{A}}}.$$

597  Next, we define $D := \frac{12}{\lambda_{\widehat{A}}} \left\{ \frac{1}{\alpha M} + \alpha \cdot 5(1+\gamma)^2 \rho_{\max}^2 \left(1 + \frac{\gamma \rho_{\max}}{\min |\lambda(C)|}\right)^2 + \frac{\alpha^2}{\beta^2} \cdot C_1 + \beta \cdot C_2 \right\}$ and

598  $E := \frac{1}{M\beta} \cdot \frac{2}{\lambda_C}$. Telescoping the above inequality yields that

$$\mathbb{E}\|\tilde{\theta}^{(m)} - \theta^*\|^2$$

$$\leq D^m \cdot \mathbb{E}\|\tilde{\theta}^{(0)} - \theta^*\|^2$$

$$+ \frac{12}{\lambda_{\widehat{A}}} \left\{ \frac{1}{\beta M} \cdot \frac{2\rho_{\max}^2 \gamma^2}{\lambda_{\widehat{A}}} \frac{3}{\lambda_C} \left[ 1 + \left[ 10\beta^2 + 10\gamma^2 \rho_{\max}^2 \left(1 + \frac{2}{\lambda_C}\right) \cdot \left(\rho_{\max} \frac{1+\gamma}{\min |\lambda(C)|}\right)^2 \frac{\alpha^2}{\beta} \right] M \right] + \alpha \cdot 5\gamma^2 \rho_{\max}^2 \right\}$$

$$\times \left\{ \frac{D^m - E^m}{D - E} \mathbb{E}\|\tilde{z}^{(0)}\|^2 + \frac{2}{1 - D} \cdot \left[ \beta \cdot \frac{24}{\lambda_C} H_{\mathrm{VR}}^2 + \frac{\alpha^2}{\beta^2} \cdot \left(1 + \frac{2}{\lambda_C}\right) \cdot \frac{2}{\lambda_C} \left(\rho_{\max} \frac{1+\gamma}{\min |\lambda(C)|}\right)^2 G_{\mathrm{VR}}^2 \right] \right\}$$

$$+ \frac{1}{1-D} \left\{ \frac{\beta}{M} \cdot \frac{12}{\lambda_{\widehat{A}}} \frac{60\rho_{\max}^2 \gamma^2}{\lambda_{\widehat{A}} \lambda_C} \cdot K_2 + \frac{\alpha^2}{\beta^2} \frac{1}{M} \cdot \frac{12}{\lambda_{\widehat{A}}} \frac{60\rho_{\max}^2 \gamma^2}{\lambda_{\widehat{A}} \lambda_C} \left(\rho_{\max} \frac{1+\gamma}{\min |\lambda(C)|}\right)^2 K_1 \left(1 + \frac{2}{\lambda_C}\right) + \frac{\alpha}{M} \cdot \frac{60K_1}{\lambda_{\widehat{A}}} \right\}$$

$$= D^m \cdot \mathbb{E}\|\tilde{\theta}^{(0)} - \theta^*\|^2$$

$$+ \frac{12}{\lambda_{\widehat{A}}} \left\{ \frac{2\rho_{\max}^2 \gamma^2}{\lambda_{\widehat{A}}} \frac{3}{\lambda_C} \left[ \frac{1}{\beta M} + \beta \cdot 10 + \frac{\alpha^2}{\beta^2} \cdot 10\gamma^2 \rho_{\max}^2 \left(1 + \frac{2}{\lambda_C}\right) \cdot \left(\rho_{\max} \frac{1+\gamma}{\min |\lambda(C)|}\right)^2 \right] + \alpha \cdot 5\gamma^2 \rho_{\max}^2 \right\}$$

$$\times \left\{ \frac{D^m - E^m}{D - E} \mathbb{E}\|\tilde{z}^{(0)}\|^2 + \frac{2}{1 - D} \cdot \left[ \beta \cdot \frac{24}{\lambda_C} H_{\mathrm{VR}}^2 + \frac{\alpha^2}{\beta^2} \cdot \left(1 + \frac{2}{\lambda_C}\right) \cdot \frac{2}{\lambda_C} \left(\rho_{\max} \frac{1+\gamma}{\min |\lambda(C)|}\right)^2 G_{\mathrm{VR}}^2 \right] \right\}$$

$$+ \frac{1}{1-D} \left\{ \frac{\beta}{M} \cdot \frac{12}{\lambda_{\widehat{A}}} \frac{60\rho_{\max}^2 \gamma^2}{\lambda_{\widehat{A}} \lambda_C} \cdot K_2 + \frac{\alpha^2}{\beta^2} \frac{1}{M} \cdot \frac{12}{\lambda_{\widehat{A}}} \frac{60\rho_{\max}^2 \gamma^2}{\lambda_{\widehat{A}} \lambda_C} \left(\rho_{\max} \frac{1+\gamma}{\min |\lambda(C)|}\right)^2 K_1 \left(1 + \frac{2}{\lambda_C}\right) + \frac{\alpha}{M} \cdot \frac{60K_1}{\lambda_{\widehat{A}}} \right\},$$

599  where in the last equality we re-arrange and simplify the upper bound to get the desired bound.

600                                                                                                       □

601  **Lemma D.5** (Refined Bound for $\mathbb{E}\|\tilde{z}^{(m)}\|^2$). *Under the same assumptions as those of Theorem 3.1,*
602  *Lemma D.3, Lemma D.1, Lemma D.4 and Lemma D.2, choose the learning rates $\alpha, \beta$ and the batch*
603  *size $M$ such that*

$$F := \frac{4}{\lambda_C} \left[ \frac{1}{\beta M} + \beta \cdot 10 + \frac{\alpha^2}{\beta^2} \cdot 10\gamma^2 \rho_{\max}^2 \left(1 + \frac{2}{\lambda_C}\right) \cdot \left(\rho_{\max} \frac{1+\gamma}{\min |\lambda(C)|}\right)^2 \right.$$

$$\left. + \left( \frac{\alpha^3}{\beta^2} \cdot C_3 + \alpha\beta \cdot C_4 \right) \frac{30}{\lambda_{\widehat{A}}} \gamma^2 \rho_{\max}^2 \right] < 1 \tag{23}$$

604  *and*

605      • $\frac{1}{\alpha M} + \alpha \cdot 5(1+\gamma)^2 \rho_{\max}^2 \left(1 + \frac{\gamma \rho_{\max}}{\min |\lambda(C)|}\right)^2 \leq \frac{\lambda_{\widehat{A}}}{6}$,

606      • $\frac{\alpha^2}{\beta^2} \cdot C_3 + \beta \cdot C_4 \leq \frac{1-D}{144} \frac{\lambda_{\widehat{A}}^2 \lambda_C}{\rho_{\max}^2 \gamma^2}$,

607      • $\frac{\alpha^2}{\beta^2} \cdot C_3 + \beta \cdot C_4 \leq 5(1 - D)$,

608  *where*

$$C_3 := 10(1+\gamma)^2 \rho_{\max}^2 \cdot \left(1 + \frac{\gamma \rho_{\max}}{\min |\lambda(C)|}\right)^2 \left(1 + \frac{2}{\lambda_C}\right) \cdot \left(\rho_{\max} \frac{1+\gamma}{\min |\lambda(C)|}\right)^2 \tag{24}$$

609  *and*

$$C_4 := 10(1+\gamma)^2 \rho_{\max}^2 \cdot \left(1 + \frac{1}{\min |\lambda(C)|}\right)^2 \tag{25}$$

610  *and $D$ is specified in eq.(18) in Lemma D.4. Then, the following refined bound holds.*

$$\mathbb{E}\|\tilde{z}^{(m)}\|^2$$

$$\leq F^m \cdot \mathbb{E}\|\tilde{z}^{(0)}\|^2 + \frac{D^m - F^m}{D - F} \cdot \mathbb{E}\|\tilde{\theta}^{(0)} - \theta^*\|^2 + \frac{\frac{D^m - F^m}{D - F} - \frac{E^m - F^m}{E - F}}{D - E} \mathbb{E}\|\tilde{z}^{(0)}\|^2$$

$$+ \frac{1}{1-F}\frac{1}{1-D}\frac{1152}{\lambda_{\widehat{A}}^2}\frac{\rho_{\max}^2\gamma^2}{\lambda_C^2}\Big(\frac{\alpha^2}{\beta^2}\cdot C_3 + \beta\cdot C_4\Big)\Big[\frac{1}{\beta M} + \beta\cdot 20 + \frac{\alpha^2}{\beta^2}\cdot 20\gamma^2\rho_{\max}^2\Big(1+\frac{2}{\lambda_C}\Big)\cdot\Big(\rho_{\max}\frac{1+\gamma}{\min|\lambda(C)|}\Big)^2\Big]$$

$$\times \Big[\beta\cdot\frac{24}{\lambda_C}H_{VR}^2 + \frac{\alpha^2}{\beta^2}\cdot\Big(1+\frac{2}{\lambda_C}\Big)\cdot\frac{2}{\lambda_C}\Big(\rho_{\max}\frac{1+\gamma}{\min|\lambda(C)|}\Big)^2 G_{VR}^2\Big]$$

$$+ \frac{1}{1-F}\frac{1}{1-D}\Big[\frac{\beta}{M}\cdot\Big(\frac{80}{\lambda_C}K_2 + C_4\frac{600}{\lambda_C}\frac{K_1}{\lambda_{\widehat{A}}}\Big) + \frac{\alpha^2}{\beta^2}\frac{1}{M}\cdot\Big(\frac{80}{\lambda_C}\Big(\rho_{\max}\frac{1+\gamma}{\min|\lambda(C)|}\Big)^2 K_1\Big(1+\frac{2}{\lambda_C}\Big) + C_3\frac{600}{\lambda_C}\frac{K_1}{\lambda_{\widehat{A}}}\Big)\Big].$$

where $K_1$ is specified in eq.(26) in Lemma E.1, $K_2$ is specified in eq.(32) in Lemma E.2, $D$ and $E$ are specified in eq.(18) and eq.(19) in Lemma D.4.

*Proof.* From Lemma D.3, we have the following inequality

$$\frac{\lambda_C}{3}\beta\sum_{t=0}^{M-1}\mathbb{E}_{m,0}\|z_t^{(m)}\|^2$$

$$\leq\Big[1 + \Big[10\beta^2 + 10\gamma^2\rho_{\max}^2\Big(1+\frac{2}{\lambda_C}\Big)\cdot\Big(\rho_{\max}\frac{1+\gamma}{\min|\lambda(C)|}\Big)^2\frac{\alpha^2}{\beta}\Big]M\Big]\mathbb{E}_{m,0}\|\tilde{z}^{(m-1)}\|^2$$

$$+ 10(1+\gamma)^2\rho_{\max}^2\cdot\Big[\Big(1+\frac{\gamma\rho_{\max}}{\min|\lambda(C)|}\Big)^2\Big(1+\frac{2}{\lambda_C}\Big)\cdot\Big(\rho_{\max}\frac{1+\gamma}{\min|\lambda(C)|}\Big)^2\frac{\alpha^2}{\beta}$$

$$+ \Big(1+\frac{1}{\min|\lambda(C)|}\Big)^2\beta^2\Big]\Big(\sum_{t=0}^{M-1}\mathbb{E}_{m,0}\|\theta_t^{(m)}-\theta^*\|^2 + M\mathbb{E}_{m,0}\|\tilde{\theta}^{(m-1)}-\theta^*\|^2\Big)$$

$$+ 10K_2\beta^2 + 10\Big(\rho_{\max}\frac{1+\gamma}{\min|\lambda(C)|}\Big)^2 K_1\Big(1+\frac{2}{\lambda_C}\Big)\frac{\alpha^2}{\beta}.$$

Note that we have already bounded $\sum_{t=0}^{M-1}\mathbb{E}_{m,0}\|\theta_t^{(m)}-\theta^*\|^2$ in Lemma D.1. Then, we plug the result of Lemma D.1 into the above inequality and obtain that

$$\frac{\lambda_C}{3}\beta\sum_{t=0}^{M-1}\mathbb{E}_{m,0}\|z_t^{(m)}\|^2$$

$$\leq\Big[1 + \Big[10\beta^2 + 10\gamma^2\rho_{\max}^2\Big(1+\frac{2}{\lambda_C}\Big)\cdot\Big(\rho_{\max}\frac{1+\gamma}{\min|\lambda(C)|}\Big)^2\frac{\alpha^2}{\beta}\Big]M\Big]\mathbb{E}_{m,0}\|\tilde{z}^{(m-1)}\|^2$$

$$+ 10(1+\gamma)^2\rho_{\max}^2\cdot\Big[\Big(1+\frac{\gamma\rho_{\max}}{\min|\lambda(C)|}\Big)^2\Big(1+\frac{2}{\lambda_C}\Big)\cdot\Big(\rho_{\max}\frac{1+\gamma}{\min|\lambda(C)|}\Big)^2\frac{\alpha^2}{\beta}$$

$$+ \Big(1+\frac{1}{\min|\lambda(C)|}\Big)^2\beta^2\Big]\Big(\frac{6}{\lambda_{\widehat{A}}}\frac{1}{\alpha}\Big\{\Big[1 + \alpha^2 M\cdot 5(1+\gamma)^2\rho_{\max}^2\Big(1+\frac{\gamma\rho_{\max}}{\min|\lambda(C)|}\Big)^2\Big]\mathbb{E}_{m,0}\|\tilde{\theta}^{(m-1)}-\theta^*\|^2 + \alpha^2\cdot 5K_1$$

$$+ \alpha\cdot\frac{2\rho_{\max}^2\gamma^2}{\lambda_{\widehat{A}}}\sum_{t=0}^{M-1}\mathbb{E}_{m,0}\|z_t^{(m)}\|^2 + \alpha^2 M\cdot 5\gamma^2\rho_{\max}^2\mathbb{E}_{m,0}\|\tilde{z}^{(m-1)}\|^2\Big\} + M\mathbb{E}_{m,0}\|\tilde{\theta}^{(m-1)}-\theta^*\|^2\Big)$$

$$+ 10K_2\beta^2 + 10\Big(\rho_{\max}\frac{1+\gamma}{\min|\lambda(C)|}\Big)^2 K_1\Big(1+\frac{2}{\lambda_C}\Big)\frac{\alpha^2}{\beta}.$$

Define $C_3 = 10(1+\gamma)^2\rho_{\max}^2\cdot\Big(1+\frac{\gamma\rho_{\max}}{\min|\lambda(C)|}\Big)^2\Big(1+\frac{2}{\lambda_C}\Big)\cdot\Big(\rho_{\max}\frac{1+\gamma}{\min|\lambda(C)|}\Big)^2$ and $C_4 = 10(1+\gamma)^2\rho_{\max}^2\cdot\Big(1+\frac{1}{\min|\lambda(C)|}\Big)^2$, then the above inequality can be re-written as

$$\frac{\lambda_C}{3}\beta\sum_{t=0}^{M-1}\mathbb{E}_{m,0}\|z_t^{(m)}\|^2$$

$$\leq\Big[1 + \Big[10\beta^2 + 10\gamma^2\rho_{\max}^2\Big(1+\frac{2}{\lambda_C}\Big)\cdot\Big(\rho_{\max}\frac{1+\gamma}{\min|\lambda(C)|}\Big)^2\frac{\alpha^2}{\beta}\Big]M\Big]\mathbb{E}_{m,0}\|\tilde{z}^{(m-1)}\|^2$$

$$+ \Big(\frac{\alpha^2}{\beta}\cdot C_3 + \beta^2\cdot C_4\Big)\Big(\frac{6}{\lambda_{\widehat{A}}}\frac{1}{\alpha}\Big\{\Big[1 + \alpha^2 M\cdot 5(1+\gamma)^2\rho_{\max}^2\Big(1+\frac{\gamma\rho_{\max}}{\min|\lambda(C)|}\Big)^2\Big]\mathbb{E}_{m,0}\|\tilde{\theta}^{(m-1)}-\theta^*\|^2 + \alpha^2\cdot 5K_1$$

$$+ \alpha \cdot \frac{2\rho_{\max}^2 \gamma^2}{\lambda_{\widehat{A}}} \sum_{t=0}^{M-1} \mathbb{E}_{m,0} \|z_t^{(m)}\|^2 + \alpha^2 M \cdot 5\gamma^2 \rho_{\max}^2 \mathbb{E}_{m,0} \|\tilde{z}^{(m-1)}\|^2 \Big\} + M\mathbb{E}_{m,0} \|\tilde{\theta}^{(m-1)} - \theta^*\|^2 \Big)$$

$$+ 10K_2 \beta^2 + 10\big(\rho_{\max} \frac{1+\gamma}{\min|\lambda(C)|}\big)^2 K_1 \big(1 + \frac{2}{\lambda_C}\big) \frac{\alpha^2}{\beta}.$$

Simplifying the above inequality yields that

$$\Big[\frac{\lambda_C}{3}\beta - \big(\frac{\alpha^2}{\beta} \cdot C_3 + \beta^2 \cdot C_4\big) \frac{12\rho_{\max}^2 \gamma^2}{\lambda_{\widehat{A}}^2}\Big] \sum_{t=0}^{M-1} \mathbb{E}_{m,0} \|z_t^{(m)}\|^2$$

$$\leq \Big[1 + \Big[10\beta^2 + 10\gamma^2 \rho_{\max}^2 \big(1 + \frac{2}{\lambda_C}\big) \cdot \big(\rho_{\max} \frac{1+\gamma}{\min|\lambda(C)|}\big)^2 \frac{\alpha^2}{\beta}\Big] M + \big(\frac{\alpha^2}{\beta} \cdot C_3 + \beta^2 \cdot C_4\big) \frac{30}{\lambda_{\widehat{A}}} \gamma^2 \rho_{\max}^2 \alpha M\Big] \mathbb{E}_{m,0} \|\tilde{z}^{(m-1)}\|^2$$

$$+ \big(\frac{\alpha^2}{\beta} \cdot C_3 + \beta^2 \cdot C_4\big) \big(\frac{6}{\lambda_{\widehat{A}}} \frac{1}{\alpha} \big[1 + \alpha^2 M \cdot 5(1+\gamma)^2 \rho_{\max}^2 \big(1 + \frac{\gamma\rho_{\max}}{\min|\lambda(C)|}\big)^2\big] + M\big) \mathbb{E}_{m,0} \|\tilde{\theta}^{(m-1)} - \theta^*\|^2$$

$$+ 10K_2\beta^2 + 10\big(\rho_{\max} \frac{1+\gamma}{\min|\lambda(C)|}\big)^2 K_1 \big(1 + \frac{2}{\lambda_C}\big) \frac{\alpha^2}{\beta} + \big(\frac{\alpha^2}{\beta} \cdot C_3 + \beta^2 \cdot C_4\big)\alpha \cdot \frac{30}{\lambda_{\widehat{A}}} K_1.$$

Let $\frac{\lambda_C}{3}\beta - \big(\frac{\alpha^2}{\beta} \cdot C_3 + \beta^2 \cdot C_4\big) \frac{12\rho_{\max}^2 \gamma^2}{\lambda_{\widehat{A}}^2} \geq \frac{\lambda_C}{4}\beta$. Dividing $\frac{\lambda_C}{4}\beta M$ and taking total expectation on both sides, and applying Jensen's inequality to the left-hand side, we obtain that

$$\mathbb{E}\|\tilde{z}^{(m)}\|^2$$

$$\leq \frac{4}{\lambda_C}\Big[\frac{1}{\beta M} + \beta \cdot 10 + \frac{\alpha^2}{\beta^2} \cdot 10\gamma^2 \rho_{\max}^2 \big(1 + \frac{2}{\lambda_C}\big) \cdot \big(\rho_{\max} \frac{1+\gamma}{\min|\lambda(C)|}\big)^2 + \big(\frac{\alpha^3}{\beta^2} \cdot C_3 + \alpha\beta \cdot C_4\big) \frac{30}{\lambda_{\widehat{A}}} \gamma^2 \rho_{\max}^2\Big] \mathbb{E}\|\tilde{z}^{(m-1)}\|^2$$

$$+ \frac{4}{\lambda_C}\big(\frac{\alpha^2}{\beta^2} \cdot C_3 + \beta \cdot C_4\big) \big(\frac{6}{\lambda_{\widehat{A}}} \frac{1}{\alpha M} \big[1 + \alpha^2 M \cdot 5(1+\gamma)^2 \rho_{\max}^2 \big(1 + \frac{\gamma\rho_{\max}}{\min|\lambda(C)|}\big)^2\big] + 1\big) \mathbb{E}\|\tilde{\theta}^{(m-1)} - \theta^*\|^2$$

$$+ \frac{40}{\lambda_C} K_2 \frac{\beta}{M} + \frac{40}{\lambda_C} \big(\rho_{\max} \frac{1+\gamma}{\min|\lambda(C)|}\big)^2 K_1 \big(1 + \frac{2}{\lambda_C}\big) \frac{\alpha^2}{\beta^2} \frac{1}{M} + \frac{\alpha}{M} \big(\frac{\alpha^2}{\beta^2} \cdot C_3 + \beta \cdot C_4\big) \cdot \frac{4}{\lambda_C} \frac{30}{\lambda_{\widehat{A}}} K_1.$$

Define $F := \frac{4}{\lambda_C}\Big[\frac{1}{\beta M} + \beta \cdot 10 + \frac{\alpha^2}{\beta^2} \cdot 10\gamma^2 \rho_{\max}^2 \big(1 + \frac{2}{\lambda_C}\big) \cdot \big(\rho_{\max} \frac{1+\gamma}{\min|\lambda(C)|}\big)^2 + \big(\frac{\alpha^3}{\beta^2} \cdot C_3 + \alpha\beta \cdot$ $C_4\big) \frac{30}{\lambda_{\widehat{A}}} \gamma^2 \rho_{\max}^2\Big]$. The above inequality can be simplified as

$$\mathbb{E}\|\tilde{z}^{(m)}\|^2$$

$$\leq F \cdot \mathbb{E}\|\tilde{z}^{(m-1)}\|^2$$

$$+ \frac{4}{\lambda_C}\big(\frac{\alpha^2}{\beta^2} \cdot C_3 + \beta \cdot C_4\big) \big(\frac{6}{\lambda_{\widehat{A}}} \frac{1}{\alpha M} \big[1 + \alpha^2 M \cdot 5(1+\gamma)^2 \rho_{\max}^2 \big(1 + \frac{\gamma\rho_{\max}}{\min|\lambda(C)|}\big)^2\big] + 1\big) \mathbb{E}\|\tilde{\theta}^{(m-1)} - \theta^*\|^2$$

$$+ \frac{40}{\lambda_C} K_2 \frac{\beta}{M} + \frac{40}{\lambda_C} \big(\rho_{\max} \frac{1+\gamma}{\min|\lambda(C)|}\big)^2 K_1 \big(1 + \frac{2}{\lambda_C}\big) \frac{\alpha^2}{\beta^2} \frac{1}{M} + \frac{\alpha}{M} \big(\frac{\alpha^2}{\beta^2} \cdot C_3 + \beta \cdot C_4\big) \cdot \frac{4}{\lambda_C} \frac{30}{\lambda_{\widehat{A}}} K_1.$$

Lastly, recall that we already have the preliminary convergence bound of $\mathbb{E}\|\tilde{\theta}^{(m-1)} - \theta^*\|^2$ in Lemma D.4. Apply this result to the above inequality yields that

$$\mathbb{E}\|\tilde{z}^{(m)}\|^2$$

$$\leq F \cdot \mathbb{E}\|\tilde{z}^{(m-1)}\|^2$$

$$+ \frac{4}{\lambda_C}\big(\frac{\alpha^2}{\beta^2} \cdot C_3 + \beta \cdot C_4\big) \big(\frac{6}{\lambda_{\widehat{A}}} \frac{1}{\alpha M} \big[1 + \alpha^2 M \cdot 5(1+\gamma)^2 \rho_{\max}^2 \big(1 + \frac{\gamma\rho_{\max}}{\min|\lambda(C)|}\big)^2\big] + 1\big) \Big[D^{m-1} \cdot \mathbb{E}\|\tilde{\theta}^{(0)} - \theta^*\|^2$$

$$+ \frac{12}{\lambda_{\widehat{A}}} \Big\{\frac{2\rho_{\max}^2 \gamma^2}{\lambda_{\widehat{A}}} \frac{3}{\lambda_C} \big[\frac{1}{\beta M} + \beta \cdot 10 + \frac{\alpha^2}{\beta^2} \cdot 10\gamma^2 \rho_{\max}^2 \big(1 + \frac{2}{\lambda_C}\big) \cdot \big(\rho_{\max} \frac{1+\gamma}{\min|\lambda(C)|}\big)^2\big] + \alpha \cdot 5\gamma^2 \rho_{\max}^2 \Big\}$$

$$\times \Big\{\frac{D^{m-1} - E^{m-1}}{D - E} \mathbb{E}\|\tilde{z}^{(0)}\|^2 + \frac{2}{1-D} \cdot \big[\beta \cdot \frac{24}{\lambda_C} H_{\text{VR}}^2 + \frac{\alpha^2}{\beta^2} \cdot \big(1 + \frac{2}{\lambda_C}\big) \cdot \frac{2}{\lambda_C} \big(\rho_{\max} \frac{1+\gamma}{\min|\lambda(C)|}\big)^2 G_{\text{VR}}^2\big]\Big\}$$

$$+ \frac{1}{1-D}\Big\{\frac{\beta}{M} \cdot \frac{12}{\lambda_{\widehat{A}}} \frac{60\rho_{\max}^2 \gamma^2}{\lambda_{\widehat{A}} \lambda_C} \cdot K_2 + \frac{\alpha^2}{\beta^2} \frac{1}{M} \cdot \frac{12}{\lambda_{\widehat{A}}} \frac{60\rho_{\max}^2 \gamma^2}{\lambda_{\widehat{A}} \lambda_C} \cdot \big(\rho_{\max} \frac{1+\gamma}{\min|\lambda(C)|}\big)^2 K_1 \big(1 + \frac{2}{\lambda_C}\big) + \frac{\alpha}{M} \cdot \frac{60K_1}{\lambda_{\widehat{A}}}\Big\}\Big]$$

$$+ \frac{40}{\lambda_C} K_2 \frac{\beta}{M} + \frac{40}{\lambda_C} \Big(\rho_{\max} \frac{1+\gamma}{\min |\lambda(C)|}\Big)^2 K_1 \Big(1 + \frac{2}{\lambda_C}\Big) \frac{\alpha^2}{\beta^2} \frac{1}{M} + \frac{\alpha}{M} \Big(\frac{\alpha^2}{\beta^2} \cdot C_3 + \beta \cdot C_4\Big) \cdot \frac{4}{\lambda_C} \frac{30}{\lambda_{\widehat{A}}} K_1.$$

625    Telescoping the above inequality, we obtain the final non-asymptotic bound of $\mathbb{E}\|\tilde{z}^{(m)}\|^2$ as

$$\mathbb{E}\|\tilde{z}^{(m)}\|^2$$
$$\leq F^m \cdot \mathbb{E}\|\tilde{z}^{(0)}\|^2$$
$$+ \frac{4}{\lambda_C} \Big(\frac{\alpha^2}{\beta^2} \cdot C_3 + \beta \cdot C_4\Big) \Big(\frac{6}{\lambda_{\widehat{A}}} \frac{1}{\alpha M}\Big[1 + \alpha^2 M \cdot 5(1+\gamma)^2 \rho_{\max}^2 \Big(1 + \frac{\gamma \rho_{\max}}{\min |\lambda(C)|}\Big)^2\Big] + 1\Big) \Big[\frac{D^m - F^m}{D - F} \cdot \mathbb{E}\|\tilde{\theta}^{(0)} - \theta^*\|^2$$
$$+ \frac{12}{\lambda_{\widehat{A}}} \Big\{ \frac{2\rho_{\max}^2 \gamma^2}{\lambda_{\widehat{A}}} \frac{3}{\lambda_C} \Big[\frac{1}{\beta M} + \beta \cdot 10 + \frac{\alpha^2}{\beta^2} \cdot 10\gamma^2 \rho_{\max}^2 \Big(1 + \frac{2}{\lambda_C}\Big) \cdot \Big(\rho_{\max} \frac{1+\gamma}{\min |\lambda(C)|}\Big)^2\Big] + \alpha \cdot 5\gamma^2 \rho_{\max}^2 \Big\}$$
$$\times \Big\{ \frac{\frac{D^m - F^m}{D - F} - \frac{E^m - F^m}{E - F}}{D - E} \mathbb{E}\|\tilde{z}^{(0)}\|^2 + \frac{1}{1 - F} \frac{2}{1 - D} \cdot \Big[\beta \cdot \frac{24}{\lambda_C} H_{\mathrm{VR}}^2 + \frac{\alpha^2}{\beta^2} \cdot \Big(1 + \frac{2}{\lambda_C}\Big) \cdot \frac{2}{\lambda_C} \Big(\rho_{\max} \frac{1+\gamma}{\min |\lambda(C)|}\Big)^2 G_{\mathrm{VR}}^2\Big] \Big\}$$
$$+ \frac{1}{1 - F} \frac{1}{1 - D} \Big\{ \frac{\beta}{M} \cdot \frac{12}{\lambda_{\widehat{A}}} \frac{60\rho_{\max}^2 \gamma^2}{\lambda_{\widehat{A}} \lambda_C} \cdot K_2 + \frac{\alpha^2}{\beta^2} \frac{1}{M} \cdot \frac{12}{\lambda_{\widehat{A}}} \frac{60\rho_{\max}^2 \gamma^2}{\lambda_{\widehat{A}} \lambda_C} \cdot \Big(\rho_{\max} \frac{1+\gamma}{\min |\lambda(C)|}\Big)^2 K_1 \Big(1 + \frac{2}{\lambda_C}\Big) + \frac{\alpha}{M} \cdot \frac{60 K_1}{\lambda_{\widehat{A}}} \Big\} \Big]$$
$$+ \frac{1}{1 - F} \Big[\frac{\beta}{M} \cdot \frac{40}{\lambda_C} K_2 + \frac{\alpha^2}{\beta^2} \frac{1}{M} \cdot \frac{40}{\lambda_C} \Big(\rho_{\max} \frac{1+\gamma}{\min |\lambda(C)|}\Big)^2 K_1 \Big(1 + \frac{2}{\lambda_C}\Big) + \frac{\alpha}{M} \Big(\frac{\alpha^2}{\beta^2} \cdot C_3 + \beta \cdot C_4\Big) \cdot \frac{4}{\lambda_C} \frac{30}{\lambda_{\widehat{A}}} K_1\Big].$$

626    Lastly, we make the following assumption to further simplify the inequality above. We note that these
627    requirements for the learning rate $\alpha, \beta$ and the batch size $M$ are not necessary; they are only used for
628    simplification.

629      • $\frac{6}{\lambda_{\widehat{A}}} \frac{1}{\alpha M} \Big[1 + \alpha^2 M \cdot 5(1+\gamma)^2 \rho_{\max}^2 \Big(1 + \frac{\gamma \rho_{\max}}{\min |\lambda(C)|}\Big)^2\Big] + 1 \leq 2$,

630      • $\frac{8}{\lambda_C} \Big(\frac{\alpha^2}{\beta^2} \cdot C_3 + \beta \cdot C_4\Big) \cdot \frac{1}{1-F} \frac{1}{1-D} \frac{\beta}{M} \cdot \frac{12}{\lambda_{\widehat{A}}} \frac{60\rho_{\max}^2 \gamma^2}{\lambda_{\widehat{A}} \lambda_C} \cdot K_2 \leq \frac{1}{1-F} \frac{\beta}{M} \cdot \frac{40}{\lambda_C} K_2$,

631      • $\frac{8}{\lambda_C} \Big(\frac{\alpha^2}{\beta^2} \cdot C_3 + \beta \cdot C_4\Big) \cdot \frac{1}{1-F} \frac{1}{1-D} \frac{\alpha^2}{\beta^2} \frac{1}{M} \cdot \frac{12}{\lambda_{\widehat{A}}} \frac{60\rho_{\max}^2 \gamma^2}{\lambda_{\widehat{A}} \lambda_C} \cdot \Big(\rho_{\max} \frac{1+\gamma}{\min |\lambda(C)|}\Big)^2 K_1 \Big(1 + \frac{2}{\lambda_C}\Big) \leq$
632        $\frac{1}{1-F} \frac{\alpha^2}{\beta^2} \frac{1}{M} \cdot \frac{40}{\lambda_C} \Big(\rho_{\max} \frac{1+\gamma}{\min |\lambda(C)|}\Big)^2 K_1 \Big(1 + \frac{2}{\lambda_C}\Big)$.

633    Apply these conditions to the above inequality yields that

$$\mathbb{E}\|\tilde{z}^{(m)}\|^2$$
$$\leq F^m \cdot \mathbb{E}\|\tilde{z}^{(0)}\|^2 + \frac{8}{\lambda_C} \Big(\frac{\alpha^2}{\beta^2} \cdot C_3 + \beta \cdot C_4\Big) \Big[\frac{D^m - F^m}{D - F} \cdot \mathbb{E}\|\tilde{\theta}^{(0)} - \theta^*\|^2\Big]$$
$$+ \frac{8}{\lambda_C} \Big(\frac{\alpha^2}{\beta^2} \cdot C_3 + \beta \cdot C_4\Big) \frac{12}{\lambda_{\widehat{A}}} \Big\{ \frac{2\rho_{\max}^2 \gamma^2}{\lambda_{\widehat{A}}} \frac{3}{\lambda_C} \Big[\frac{1}{\beta M} + \beta \cdot 10 + \frac{\alpha^2}{\beta^2} \cdot 10\gamma^2 \rho_{\max}^2 \Big(1 + \frac{2}{\lambda_C}\Big) \cdot \Big(\rho_{\max} \frac{1+\gamma}{\min |\lambda(C)|}\Big)^2\Big]$$
$$+ \alpha \cdot 5\gamma^2 \rho_{\max}^2 \Big\}$$
$$\times \Big\{ \frac{\frac{D^m - F^m}{D - F} - \frac{E^m - F^m}{E - F}}{D - E} \mathbb{E}\|\tilde{z}^{(0)}\|^2 + \frac{1}{1 - F} \frac{2}{1 - D} \cdot \Big[\beta \cdot \frac{24}{\lambda_C} H_{\mathrm{VR}}^2 + \frac{\alpha^2}{\beta^2} \cdot \Big(1 + \frac{2}{\lambda_C}\Big) \cdot \frac{2}{\lambda_C} \Big(\rho_{\max} \frac{1+\gamma}{\min |\lambda(C)|}\Big)^2 G_{\mathrm{VR}}^2\Big] \Big\}$$
$$+ \frac{\alpha}{M} \Big(\frac{\alpha^2}{\beta^2} \cdot C_3 + \beta \cdot C_4\Big) \cdot \frac{1}{1 - F} \Big(\frac{1}{1 - D} \frac{8}{\lambda_C} \frac{60 K_1}{\lambda_{\widehat{A}}} + \frac{4}{\lambda_C} \frac{30}{\lambda_{\widehat{A}}} K_1\Big)$$
$$+ \frac{1}{1 - F} \Big[\frac{\beta}{M} \cdot \frac{80}{\lambda_C} K_2 + \frac{\alpha^2}{\beta^2} \frac{1}{M} \cdot \frac{80}{\lambda_C} \Big(\rho_{\max} \frac{1+\gamma}{\min |\lambda(C)|}\Big)^2 K_1 \Big(1 + \frac{2}{\lambda_C}\Big)\Big].$$

634    Further note that $1 < \frac{1}{1-D}$ and $\alpha \leq 1$, the above inequality implies that

$$\mathbb{E}\|\tilde{z}^{(m)}\|^2$$
$$\leq F^m \cdot \mathbb{E}\|\tilde{z}^{(0)}\|^2 + \frac{8}{\lambda_C} \Big(\frac{\alpha^2}{\beta^2} \cdot C_3 + \beta \cdot C_4\Big) \Big[\frac{D^m - F^m}{D - F} \cdot \mathbb{E}\|\tilde{\theta}^{(0)} - \theta^*\|^2\Big]$$

$$+ \frac{8}{\lambda_C}\Big(\frac{\alpha^2}{\beta^2}\cdot C_3 + \beta\cdot C_4\Big)\frac{12}{\lambda_{\widehat{A}}}\Big\{\frac{2\rho_{\max}^2\gamma^2}{\lambda_{\widehat{A}}}\frac{3}{\lambda_C}\Big[\frac{1}{\beta M} + \beta\cdot 10 + \frac{\alpha^2}{\beta^2}\cdot 10\gamma^2\rho_{\max}^2\Big(1+\frac{2}{\lambda_C}\Big)\cdot\Big(\rho_{\max}\frac{1+\gamma}{\min|\lambda(C)|}\Big)^2\Big]$$

$$+ \alpha\cdot 5\gamma^2\rho_{\max}^2\Big\}$$

$$\times\Big\{\frac{\frac{D^m-F^m}{D-F}-\frac{E^m-F^m}{E-F}}{D-E}\mathbb{E}\|\tilde{z}^{(0)}\|^2 + \frac{1}{1-F}\frac{2}{1-D}\cdot\Big[\beta\cdot\frac{24}{\lambda_C}H_{\mathrm{VR}}^2 + \frac{\alpha^2}{\beta^2}\cdot\Big(1+\frac{2}{\lambda_C}\Big)\cdot\frac{2}{\lambda_C}\Big(\rho_{\max}\frac{1+\gamma}{\min|\lambda(C)|}\Big)^2 G_{\mathrm{VR}}^2\Big]\Big\}$$

$$+ \frac{1}{1-F}\frac{1}{1-D}\Big[\frac{\beta}{M}\cdot\Big(\frac{80}{\lambda_C}K_2 + C_4\frac{600}{\lambda_C}\frac{K_1}{\lambda_{\widehat{A}}}\Big) + \frac{\alpha^2}{\beta^2}\frac{1}{M}\cdot\Big(\frac{80}{\lambda_C}\Big(\rho_{\max}\frac{1+\gamma}{\min|\lambda(C)|}\Big)^2 K_1\Big(1+\frac{2}{\lambda_C}\Big) + C_3\frac{600}{\lambda_C}\frac{K_1}{\lambda_{\widehat{A}}}\Big)\Big].$$

Assume that $\alpha\cdot 5\gamma^2\rho_{\max}^2 \leq \frac{2\rho_{\max}^2\gamma^2}{\lambda_{\widehat{A}}}\frac{3}{\lambda_C}\Big[\beta\cdot 10 + \frac{\alpha^2}{\beta^2}\cdot 10\gamma^2\rho_{\max}^2\Big(1+\frac{2}{\lambda_C}\Big)\cdot\Big(\rho_{\max}\frac{1+\gamma}{\min|\lambda(C)|}\Big)^2\Big]$. Then, we further obtain from the above inequality that

$$\mathbb{E}\|\tilde{z}^{(m)}\|^2$$

$$\leq F^m\cdot\mathbb{E}\|\tilde{z}^{(0)}\|^2 + \frac{8}{\lambda_C}\Big(\frac{\alpha^2}{\beta^2}\cdot C_3 + \beta\cdot C_4\Big)\Big[\frac{D^m-F^m}{D-F}\cdot\mathbb{E}\|\tilde{\theta}^{(0)}-\theta^*\|^2\Big]$$

$$+ \frac{8}{\lambda_C}\Big(\frac{\alpha^2}{\beta^2}\cdot C_3 + \beta\cdot C_4\Big)\frac{12}{\lambda_{\widehat{A}}}\frac{2\rho_{\max}^2\gamma^2}{\lambda_{\widehat{A}}}\frac{3}{\lambda_C}\Big[\frac{1}{\beta M} + \beta\cdot 20 + \frac{\alpha^2}{\beta^2}\cdot 20\gamma^2\rho_{\max}^2\Big(1+\frac{2}{\lambda_C}\Big)\cdot\Big(\rho_{\max}\frac{1+\gamma}{\min|\lambda(C)|}\Big)^2\Big]$$

$$\times\Big\{\frac{\frac{D^m-F^m}{D-F}-\frac{E^m-F^m}{E-F}}{D-E}\mathbb{E}\|\tilde{z}^{(0)}\|^2 + \frac{1}{1-F}\frac{2}{1-D}\cdot\Big[\beta\cdot\frac{24}{\lambda_C}H_{\mathrm{VR}}^2 + \frac{\alpha^2}{\beta^2}\cdot\Big(1+\frac{2}{\lambda_C}\Big)\cdot\frac{2}{\lambda_C}\Big(\rho_{\max}\frac{1+\gamma}{\min|\lambda(C)|}\Big)^2 G_{\mathrm{VR}}^2\Big]\Big\}$$

$$+ \frac{1}{1-F}\frac{1}{1-D}\Big[\frac{\beta}{M}\cdot\Big(\frac{80}{\lambda_C}K_2 + C_4\frac{600}{\lambda_C}\frac{K_1}{\lambda_{\widehat{A}}}\Big) + \frac{\alpha^2}{\beta^2}\frac{1}{M}\cdot\Big(\frac{80}{\lambda_C}\Big(\rho_{\max}\frac{1+\gamma}{\min|\lambda(C)|}\Big)^2 K_1\Big(1+\frac{2}{\lambda_C}\Big) + C_3\frac{600}{\lambda_C}\frac{K_1}{\lambda_{\widehat{A}}}\Big)\Big]$$

$$= F^m\cdot\mathbb{E}\|\tilde{z}^{(0)}\|^2 + \frac{8}{\lambda_C}\Big(\frac{\alpha^2}{\beta^2}\cdot C_3 + \beta\cdot C_4\Big)\Big[\frac{D^m-F^m}{D-F}\cdot\mathbb{E}\|\tilde{\theta}^{(0)}-\theta^*\|^2\Big]$$

$$+ \frac{576}{\lambda_{\widehat{A}}^2}\frac{\rho_{\max}^2\gamma^2}{\lambda_C^2}\Big(\frac{\alpha^2}{\beta^2}\cdot C_3 + \beta\cdot C_4\Big)\Big[\frac{1}{\beta M} + \beta\cdot 20 + \frac{\alpha^2}{\beta^2}\cdot 20\gamma^2\rho_{\max}^2\Big(1+\frac{2}{\lambda_C}\Big)\cdot\Big(\rho_{\max}\frac{1+\gamma}{\min|\lambda(C)|}\Big)^2\Big]$$

$$\times\Big\{\frac{\frac{D^m-F^m}{D-F}-\frac{E^m-F^m}{E-F}}{D-E}\mathbb{E}\|\tilde{z}^{(0)}\|^2 + \frac{1}{1-F}\frac{2}{1-D}\cdot\Big[\beta\cdot\frac{24}{\lambda_C}H_{\mathrm{VR}}^2 + \frac{\alpha^2}{\beta^2}\cdot\Big(1+\frac{2}{\lambda_C}\Big)\cdot\frac{2}{\lambda_C}\Big(\rho_{\max}\frac{1+\gamma}{\min|\lambda(C)|}\Big)^2 G_{\mathrm{VR}}^2\Big]\Big\}$$

$$+ \frac{1}{1-F}\frac{1}{1-D}\Big[\frac{\beta}{M}\cdot\Big(\frac{80}{\lambda_C}K_2 + C_4\frac{600}{\lambda_C}\frac{K_1}{\lambda_{\widehat{A}}}\Big) + \frac{\alpha^2}{\beta^2}\frac{1}{M}\cdot\Big(\frac{80}{\lambda_C}\Big(\rho_{\max}\frac{1+\gamma}{\min|\lambda(C)|}\Big)^2 K_1\Big(1+\frac{2}{\lambda_C}\Big) + C_3\frac{600}{\lambda_C}\frac{K_1}{\lambda_{\widehat{A}}}\Big)\Big].$$

Lastly, assume that $\frac{576}{\lambda_{\widehat{A}}^2}\frac{\rho_{\max}^2\gamma^2}{\lambda_C^2}\Big(\frac{\alpha^2}{\beta^2}\cdot C_3 + \beta\cdot C_4\Big)\Big[\frac{1}{\beta M} + \beta\cdot 20 + \frac{\alpha^2}{\beta^2}\cdot 20\gamma^2\rho_{\max}^2\Big(1+\frac{2}{\lambda_C}\Big)\cdot$ $\Big(\rho_{\max}\frac{1+\gamma}{\min|\lambda(C)|}\Big)^2\Big] < 1$ and $\frac{8}{\lambda_C}\Big(\frac{\alpha^2}{\beta^2}\cdot C_3 + \beta\cdot C_4\Big) < 1$, the above inequality further implies that

$$\mathbb{E}\|\tilde{z}^{(m)}\|^2$$

$$\leq F^m\cdot\mathbb{E}\|\tilde{z}^{(0)}\|^2 + \frac{D^m-F^m}{D-F}\cdot\mathbb{E}\|\tilde{\theta}^{(0)}-\theta^*\|^2 + \frac{\frac{D^m-F^m}{D-F}-\frac{E^m-F^m}{E-F}}{D-E}\mathbb{E}\|\tilde{z}^{(0)}\|^2$$

$$+ \frac{1}{1-F}\frac{1}{1-D}\frac{1152}{\lambda_{\widehat{A}}^2}\frac{\rho_{\max}^2\gamma^2}{\lambda_C^2}\Big(\frac{\alpha^2}{\beta^2}\cdot C_3 + \beta\cdot C_4\Big)\Big[\frac{1}{\beta M} + \beta\cdot 20 + \frac{\alpha^2}{\beta^2}\cdot 20\gamma^2\rho_{\max}^2\Big(1+\frac{2}{\lambda_C}\Big)\cdot\Big(\rho_{\max}\frac{1+\gamma}{\min|\lambda(C)|}\Big)^2\Big]$$

$$\times\Big[\beta\cdot\frac{24}{\lambda_C}H_{\mathrm{VR}}^2 + \frac{\alpha^2}{\beta^2}\cdot\Big(1+\frac{2}{\lambda_C}\Big)\cdot\frac{2}{\lambda_C}\Big(\rho_{\max}\frac{1+\gamma}{\min|\lambda(C)|}\Big)^2 G_{\mathrm{VR}}^2\Big]$$

$$+ \frac{1}{1-F}\frac{1}{1-D}\Big[\frac{\beta}{M}\cdot\Big(\frac{80}{\lambda_C}K_2 + C_4\frac{600}{\lambda_C}\frac{K_1}{\lambda_{\widehat{A}}}\Big) + \frac{\alpha^2}{\beta^2}\frac{1}{M}\cdot\Big(\frac{80}{\lambda_C}\Big(\rho_{\max}\frac{1+\gamma}{\min|\lambda(C)|}\Big)^2 K_1\Big(1+\frac{2}{\lambda_C}\Big) + C_3\frac{600}{\lambda_C}\frac{K_1}{\lambda_{\widehat{A}}}\Big)\Big].$$

$\square$

 # E   Other Supporting Lemmas for Proving Theorem 3.1

**Lemma E.1** (One-Step Update of $\theta_t^{(m)}$). *Under the same assumptions as those of Theorem 3.1, the square norm of one-step update of $\theta_t^{(m)}$ in Algorithm 1 can be bounded as*

$$\mathbb{E}_{m,0}\|G_t^{(m)}(\theta_t^{(m)}, z_t^{(m)}) - G_t^{(m)}(\tilde{\theta}^{(m-1)}, \tilde{z}^{(m-1)}) + G^{(m)}(\tilde{\theta}^{(m-1)}, \tilde{z}^{(m-1)})\|^2$$

$$\leq 5(1+\gamma)^2\rho_{\max}^2\big(1 + \frac{\gamma\rho_{\max}}{\min|\lambda(C)|}\big)^2\big(\mathbb{E}_{m,0}\|\theta_t^{(m)} - \theta^*\|^2 + \mathbb{E}_{m,0}\|\tilde{\theta}^{(m-1)} - \theta^*\|^2\big)$$

$$+ 5\gamma^2\rho_{\max}^2\big(\mathbb{E}_{m,0}\|z_t^{(m)}\|^2 + \mathbb{E}_{m,0}\|\tilde{z}^{(m-1)}\|^2\big) + \frac{1}{M}\cdot 5K_1,$$

*where*

$$K_1 := \big[(1+\gamma)R_\theta + r_{\max}\big]^2\rho_{\max}^2\big(1 + \frac{\gamma\rho_{\max}}{\min|\lambda(C)|}\big)^2. \tag{26}$$

*Proof.* Substituting the definitions of $G_t^{(m)}(\cdot)$ and $G^{(m)}(\cdot)$ into the update of $\theta_t^{(m)}$ yields that

$$\|G_t^{(m)}(\theta_t^{(m)}, z_t^{(m)}) - G_t^{(m)}(\tilde{\theta}^{(m-1)}, \tilde{z}^{(m-1)}) + G^{(m)}(\tilde{\theta}^{(m-1)}, \tilde{z}^{(m-1)})\|^2$$

$$=\|\widehat{A}_t^{(m)}\theta_t^{(m)} + \widehat{b}_t^{(m)} + B_t^{(m)}z_t^{(m)} - \widehat{A}_t^{(m)}\tilde{\theta}^{(m-1)} - \widehat{b}_t^{(m)} - B_t^{(m)}\tilde{z}^{(m-1)} + \widehat{A}^{(m)}\tilde{\theta}^{(m-1)} + \widehat{b}^{(m)} + B^{(m)}\tilde{z}^{(m-1)}\|^2$$

$$=\|\big(\widehat{A}_t^{(m)}\theta_t^{(m)} - \widehat{A}_t^{(m)}\theta^*\big) + \big(\widehat{A}_t^{(m)}\theta^* - \widehat{A}_t^{(m)}\tilde{\theta}^{(m-1)} + \widehat{A}^{(m)}\tilde{\theta}^{(m-1)} - \widehat{A}^{(m)}\theta^*\big) + \big(\widehat{A}^{(m)}\theta^* + \widehat{b}^{(m)}\big)$$

$$+ B_t^{(m)}z_t^{(m)} - B_t^{(m)}\tilde{z}^{(m-1)} + B^{(m)}\tilde{z}^{(m-1)}\|^2$$

$$\leq 5\|\widehat{A}_t^{(m)}\|^2\|\theta_t^{(m)} - \theta^*\|^2 + 5\|\big(\widehat{A}_t^{(m)}\tilde{\theta}^{(m-1)} - \widehat{A}_t^{(m)}\theta^*\big) - \big(\widehat{A}^{(m)}\tilde{\theta}^{(m-1)} - \widehat{A}^{(m)}\theta^*\big)\|^2 + 5\|\widehat{A}^{(m)}\theta^* + \widehat{b}^{(m)}\|^2$$

$$+ 5\|B_t^{(m)}\|^2\|z_t^{(m)}\|^2 + 5\|B_t^{(m)}\tilde{z}^{(m-1)} - B^{(m)}\tilde{z}^{(m-1)}\|^2, \tag{27}$$

where the last inequality uses Jensen's inequality $\|\sum_{i=1}^n a_i\|^2 = \|\frac{1}{n}\sum_{i=1}^n (na_i)\|^2 \leq n\sum_{i=1}^n \|a_i\|^2$.
Next, we bound the third term of the right hand side of the above inequality as follows:

$$\|\widehat{A}^{(m)}\theta^* + \widehat{b}^{(m)}\|^2$$

$$\leq \|\sum_{t=0}^{M-1}\widehat{A}_t^{(m)}\theta^* + \sum_{t=0}^{M-1}\widehat{b}_t^{(m)}\|^2$$

$$= \frac{1}{M^2}\Big[\sum_{i=j}\|\widehat{A}_i^{(m)}\theta^* + \widehat{b}_i^{(m)}\|^2 + \sum_{i\neq j}\langle\widehat{A}_i^{(m)}\theta^* + \widehat{b}_i^{(m)}, \widehat{A}_j^{(m)}\theta^* + \widehat{b}_j^{(m)}\rangle\Big]$$

$$\leq \frac{1}{M}\cdot\big[(1+\gamma)R_\theta + r_{\max}\big]^2\rho_{\max}^2\big(1 + \frac{\gamma\rho_{\max}}{\min|\lambda(C)|}\big)^2 + \frac{1}{M^2}\sum_{i\neq j}\langle\widehat{A}_i^{(m)}\theta^* + \widehat{b}_i^{(m)}, \widehat{A}_j^{(m)}\theta^* + \widehat{b}_j^{(m)}\rangle, \tag{28}$$

where in the last inequality we use Lemma J.2 to bound $\|\widehat{A}_i^{(m)}\theta^* + \widehat{b}_i^{(m)}\|$. Next, consider the conditional expectation of the second term of the above inequality, and, without loss of generality, assume $i < j$, we obtain that

$$\mathbb{E}_{m,0}\langle\widehat{A}_i^{(m)}\theta^* + \widehat{b}_i^{(m)}, \widehat{A}_j^{(m)}\theta^* + \widehat{b}_j^{(m)}\rangle$$

$$=\mathbb{E}_{m,0}\langle\widehat{A}_i^{(m)}\theta^* + \widehat{b}_i^{(m)}, \mathbb{E}_{m,i}\big[\widehat{A}_j^{(m)}\theta^* + \widehat{b}_j^{(m)}\big]\rangle$$

$$=\mathbb{E}_{m,0}\langle\widehat{A}_i^{(m)}\theta^* + \widehat{b}_i^{(m)}, \widehat{A}\theta^* + \widehat{b}\rangle$$

$$=0,$$

which follows from the i.i.d. sampling scheme. Substituting the above result into (28), we obtain that

$$\mathbb{E}_{m,0}\|\widehat{A}^{(m)}\theta^* + \widehat{b}^{(m)}\|^2 \leq \frac{1}{M}\cdot\big[(1+\gamma)R_\theta + r_{\max}\big]^2\rho_{\max}^2\big(1 + \frac{\gamma\rho_{\max}}{\min|\lambda(C)|}\big)^2. \tag{29}$$

On the other hand, note that $\mathrm{Var}X \leq \mathbb{E}X^2$, we have

$$\mathbb{E}_{m,t-1}\|\big(\widehat{A}_t^{(m)}\tilde{\theta}^{(m-1)} - \widehat{A}_t^{(m)}\theta^*\big) - \big(\widehat{A}^{(m)}\tilde{\theta}^{(m-1)} - \widehat{A}^{(m)}\theta^*\big)\|^2$$

$$
\begin{aligned}
&= \mathrm{Var}_{m,t-1}\big(\widehat{A}_t^{(m)}\tilde{\theta}^{(m-1)} - \widehat{A}_t^{(m)}\theta^*\big)\\
&\leq \mathbb{E}_{m,t-1}\big(\widehat{A}_t^{(m)}\tilde{\theta}^{(m-1)} - \widehat{A}_t^{(m)}\theta^*\big)^2\\
&\leq \mathbb{E}_{m,t-1}\|\widehat{A}_t^{(m)}\|^2\|\tilde{\theta}^{(m-1)} - \theta^*\|^2
\end{aligned}
\tag{30}
$$

and similarly,

$$
\mathbb{E}_{m,t-1}\|B_t^{(m)}\tilde{z}^{(m-1)} - B^{(m)}\tilde{z}^{(m-1)}\|^2 \leq \mathbb{E}_{m,t-1}\|B_t^{(m)}\|^2\|\tilde{z}^{(m-1)}\|^2.
\tag{31}
$$

Substituting eqs. (29), (30), (31) into (27) yields that

$$
\begin{aligned}
&\mathbb{E}_{m,0}\|G_t^{(m)}(\theta_t^{(m)}, z_t^{(m)}) - G_t^{(m)}(\tilde{\theta}^{(m-1)}, \tilde{z}^{(m-1)}) + G^{(m)}(\tilde{\theta}^{(m-1)}, \tilde{z}^{(m-1)})\|^2\\
&\leq 5(1+\gamma)^2\rho_{\max}^2\big(1 + \frac{\gamma\rho_{\max}}{\min|\lambda(C)|}\big)^2\big(\mathbb{E}_{m,0}\|\theta_t^{(m)} - \theta^*\|^2 + \mathbb{E}_{m,0}\|\tilde{\theta}^{(m-1)} - \theta^*\|^2\big)\\
&\quad + 5\gamma^2\rho_{\max}^2\big(\mathbb{E}_{m,0}\|z_t^{(m)}\|^2 + \mathbb{E}_{m,0}\|\tilde{z}^{(m-1)}\|^2\big) + \frac{1}{M}\cdot 5\big[(1+\gamma)R_\theta + r_{\max}\big]^2\rho_{\max}^2\big(1 + \frac{\gamma\rho_{\max}}{\min|\lambda(C)|}\big)^2.
\end{aligned}
$$

$\square$

**Lemma E.2** (One-Step Update of $z_t^{(m)}$). *Under the same assumptions as those of Theorem 3.1, the square norm of one-step update of $z_t^{(m)}$ in Algorithm 1 is bounded as*

$$
\begin{aligned}
&\mathbb{E}_{m,0}\|H_t^{(m)}(\theta_t^{(m)}, z_t^{(m)}) - H_t^{(m)}(\tilde{\theta}^{(m-1)}, \tilde{z}^{(m-1)}) + H^{(m)}(\tilde{\theta}^{(m-1)}, \tilde{z}^{(m-1)})\|^2\\
&\leq 5(1+\gamma)^2\rho_{\max}^2\big(1 + \frac{1}{\min|\lambda(C)|}\big)^2\big(\mathbb{E}_{m,0}\|\theta_t^{(m)} - \theta^*\|^2 + \mathbb{E}_{m,0}\|\tilde{\theta}^{(m-1)} - \theta^*\|^2\big)\\
&\quad + 5\big(\mathbb{E}_{m,0}\|z_t^{(m)}\|^2 + \mathbb{E}_{m,0}\|\tilde{z}^{(m-1)}\|^2\big) + \frac{5K_2}{M},
\end{aligned}
$$

*where*

$$
K_2 := \big[(1+\gamma)R_\theta + r_{\max}\big]^2\big(1 + \frac{1}{\min|\lambda_C|}\big)^2.
\tag{32}
$$

*Proof.* Follow a similar proof logic as that of Lemma E.1, we obtain the following bound for the square norm of the one-step update of $z_t^{(m)}$,

$$
\begin{aligned}
&\|H_t^{(m)}(\theta_t^{(m)}, z_t^{(m)}) - H_t^{(m)}(\tilde{\theta}^{(m-1)}, \tilde{z}^{(m-1)}) + H^{(m)}(\tilde{\theta}^{(m-1)}, \tilde{z}^{(m-1)})\|^2\\
&\leq 5\|\bar{A}_t^{(m)}\|^2\|\theta_t^{(m)} - \theta^*\|^2 + 5\|\big(\bar{A}_t^{(m)}\tilde{\theta}^{(m-1)} - \bar{A}_t^{(m)}\theta^*\big) - \big(\bar{A}^{(m)}\tilde{\theta}^{(m-1)} - \bar{A}^{(m)}\theta^*\big)\|^2 + 5\|\bar{A}^{(m)}\theta^* + \bar{b}^{(m)}\|^2\\
&\quad + 5\|C_t^{(m)}\|^2\|z_t^{(m)}\|^2 + 5\|C_t^{(m)}\tilde{z}^{(m-1)} - C^{(m)}\tilde{z}^{(m-1)}\|^2.
\end{aligned}
\tag{33}
$$

Then, we take $\mathbb{E}_{m,0}$ on both sides, follow the same steps in the proof of Lemma E.1 and notice that $\mathbb{E}_{m,0}\|\bar{A}^{(m)}\theta^* + \bar{b}^{(m)}\|^2$ in eq. (33) is bounded by

$$
\begin{aligned}
\mathbb{E}_{m,0}\|\bar{A}^{(m)}\theta^* + \bar{b}^{(m)}\|^2 &\leq \frac{1}{M^2}\sum_{i=j}\|\bar{A}_i^{(m)}\theta^* + \bar{b}_i^{(m)}\|^2\\
&\leq \frac{1}{M}\cdot\big[(1+\gamma)R_\theta + r_{\max}\big]^2\big(1 + \frac{1}{\min|\lambda_C|}\big)^2
\end{aligned}
$$

using Lemma J.3. Finally, we obtain that

$$
\begin{aligned}
&\mathbb{E}_{m,0}\|H_t^{(m)}(\theta_t^{(m)}, z_t^{(m)}) - H_t^{(m)}(\tilde{\theta}^{(m-1)}, \tilde{z}^{(m-1)}) + H^{(m)}(\tilde{\theta}^{(m-1)}, \tilde{z}^{(m-1)})\|^2\\
&\leq 5(1+\gamma)^2\rho_{\max}^2\big(1 + \frac{1}{\min|\lambda(C)|}\big)^2\big(\mathbb{E}_{m,0}\|\theta_t^{(m)} - \theta^*\|^2 + \mathbb{E}_{m,0}\|\tilde{\theta}^{(m-1)} - \theta^*\|^2\big)\\
&\quad + 5\big(\mathbb{E}_{m,0}\|z_t^{(m)}\|^2 + \mathbb{E}_{m,0}\|\tilde{z}^{(m-1)}\|^2\big) + \frac{5}{M}\cdot\big[(1+\gamma)R_\theta + r_{\max}\big]^2\big(1 + \frac{1}{\min|\lambda_C|}\big)^2.
\end{aligned}
$$

$\square$

# F   Proof of Theorem 4.1

We assume the learning rates $\alpha, \beta$ and the batch size $M$ satisfy the following conditions.

$$\alpha \leq \min\Big\{\frac{\lambda_{\widehat{A}}}{30}\Big/\Big[(1+\gamma)^2\rho_{\max}^2\Big(1+\frac{\gamma\rho_{\max}}{\min|\lambda(C)|}\Big)^2\Big], \frac{3}{5}\frac{1}{\lambda_{\widehat{A}}}\Big\}, \tag{34}$$

$$\beta \leq 1, \tag{35}$$

$$M\beta > \frac{12}{\lambda_C}, \tag{36}$$

$$\frac{\lambda_C}{48}\beta - 10\beta^2 - 10\gamma^2\rho_{\max}^2\Big(\rho_{\max}\frac{1+\gamma}{\min|\lambda(C)|}\Big)^2\Big(\alpha^2 + \frac{2\alpha^2}{\lambda_C}\frac{1}{\beta}\Big) \geq 0, \tag{37}$$

$$\frac{16}{\lambda_{\widehat{A}}}\Big\{\frac{96}{\lambda_{\widehat{A}}\lambda_C}\gamma^2\rho_{\max}^2\Big[\frac{1}{\beta M} + 10\beta + 10\gamma^2\rho_{\max}^2\Big(1+\frac{2}{\lambda_C}\Big)\Big(\rho_{\max}\frac{1+\gamma}{\min|\lambda(C)|}\Big)^2\frac{\alpha^2}{\beta^2}\Big] + 5\gamma^2\rho_{\max}^2\alpha\Big\} \leq 1, \tag{38}$$

$$\Big(1+\frac{\gamma\rho_{\max}}{\min|\lambda(C)|}\Big)^2\Big(1+\frac{21}{\lambda_C}\Big)\Big(\rho_{\max}\frac{1+\gamma}{\min|\lambda(C)|}\Big)^2\frac{\alpha^2}{\beta^2} + \Big(1+\frac{1}{\min|\lambda(C)|}\Big)^2\beta^2$$
$$\leq \min\Big\{\frac{\lambda_{\widehat{A}}}{48}\Big/\Big[\frac{96}{\lambda_{\widehat{A}}\lambda_C}\gamma^2\rho_{\max}^2 \cdot 10(1+\gamma)^2\rho_{\max}^2\Big], \frac{\lambda_C}{48}\Big/\Big[120(1+\gamma)^2\rho_{\max}^2\frac{1}{\lambda_{\widehat{A}}}\Big]\Big\}, \tag{39}$$

$$\max\{D, E, F\} < 1, \tag{40}$$

where $D, E, F$ are specified in eq.(44), eq.(46), and eq.(49), respectively. We note that under the above conditions, all the supporting lemmas for proving the theorem are satisfied. We also note that for a sufficiently small $\epsilon$, our choices of learning rates and batch size $\alpha = \mathcal{O}(\epsilon^{\frac{3}{4}}), \beta = \mathcal{O}(\epsilon^{\frac{1}{2}})$, $M = \mathcal{O}(\epsilon^{-1})$ that are stated in the theorem satisfy eqs. (35) to (40).

**Proof Sketch**   The proof consists of the following key steps.

1. Develop *preliminary bound for* $\sum_{t=0}^{M-1}\|\theta_t^{(m)} - \theta^*\|^2$. (Lemma H.1)
   We first bound $\sum_{t=0}^{M-1}\|\theta_t^{(m)} - \theta^*\|^2$ in terms of $\sum_{t=0}^{M-1}\|z_t^{(m)}\|^2$, $\|\tilde{z}^{(m-1)}\|^2$, and $\|\tilde{\theta}^{(m-1)} - \theta^*\|^2$.

2. Develop *preliminary bound for* $\sum_{t=0}^{M-1}\|z_t^{(m)}\|^2$. (Lemma H.3)
   Then we bound $\sum_{t=0}^{M-1}\|z_t^{(m)}\|^2$ in terms of $\sum_{t=0}^{M-1}\|\theta_t^{(m)} - \theta^*\|^2$, $\|\tilde{z}^{(m-1)}\|^2$, and $\|\tilde{\theta}^{(m-1)} - \theta^*\|^2$, and plug it into the preliminary bound of $\sum_{t=0}^{M-1}\|\theta_t^{(m)} - \theta^*\|^2$. Then, we obtain an upper bound of $\sum_{t=0}^{M-1}\|\theta_t^{(m)} - \theta^*\|^2$ in terms of $\|\tilde{z}^{(m-1)}\|^2$, and $\|\tilde{\theta}^{(m-1)} - \theta^*\|^2$.

3. Develop *non-asymptotic bound for* $\|\tilde{z}^m\|^2$. (Lemma H.2)
   Lastly, we develop a non-asymptotic bound for $\|\tilde{z}^m\|^2$ and plug it into the previous upper bounds. Then, we obtain a relation between $\mathbb{E}\|\tilde{\theta}^{(m)} - \theta^*\|$ and $\mathbb{E}\|\tilde{\theta}^{(m-1)} - \theta^*\|$. Recursively telescoping this inequality leads to our final result.

By Lemma H.1, we have the following result:

$$\frac{\lambda_{\widehat{A}}}{12}\alpha\sum_{t=0}^{M-1}\mathbb{E}_{m,0}\|\theta_t^{(m)} - \theta^*\|^2$$

$$\leq\Big[1 + \alpha^2 M \cdot 5(1+\gamma)^2\rho_{\max}^2\Big(1+\frac{\gamma\rho_{\max}}{\min|\lambda(C)|}\Big)^2\Big]\mathbb{E}_{m,0}\|\tilde{\theta}^{(m-1)} - \theta^*\|^2 + \alpha \cdot 2K_2 + \alpha^2 \cdot 5K_1$$

$$+ \alpha \cdot \frac{6}{\lambda_{\widehat{A}}}\gamma^2\rho_{\max}^2\sum_{t=0}^{M-1}\mathbb{E}_{m,0}\|z_t^{(m)}\|^2 + \alpha^2 M \cdot 5\gamma^2\rho_{\max}^2\mathbb{E}_{m,0}\|\tilde{z}^{(m-1)}\|^2, \tag{41}$$

where $K_1$ is specified in eq. (63) of Lemma I.1, and $K_2$ is specified in eq. (67) of Lemma I.2. By Lemma H.3, we have that

$$\frac{\lambda_C}{16}\beta\sum_{t=0}^{M-1}\mathbb{E}_{m,0}\|z_t^{(m)}\|^2$$

$$\leq \left[1 + \left[10\beta^2 + 10\gamma^2\rho_{\max}^2\left(\alpha^2 + \frac{2\alpha^2}{\lambda_C}\frac{1}{\beta}\right)\cdot\left(\rho_{\max}\frac{1+\gamma}{\min|\lambda(C)|}\right)^2\right]M\right]\mathbb{E}_{m,0}\|\tilde{z}^{(m-1)}\|^2$$

$$+ 10(1+\gamma)^2\rho_{\max}^2\cdot\left[\left(1 + \frac{\gamma\rho_{\max}}{\min|\lambda(C)|}\right)^2\left(\alpha^2 + \frac{2\alpha^2}{\lambda_C}\frac{1}{\beta}\right)\cdot\left(\rho_{\max}\frac{1+\gamma}{\min|\lambda(C)|}\right)^2\right.$$

$$\left. + \left(1 + \frac{1}{\min|\lambda(C)|}\right)^2\beta^2\right]\left(\sum_{t=0}^{M-1}\mathbb{E}_{m,0}\|\theta_t^{(m)} - \theta^*\|^2 + M\mathbb{E}_{m,0}\|\tilde{\theta}^{(m-1)} - \theta^*\|^2\right)$$

$$+ (2K_3 + 2K_4)\beta + 10K_5\beta^2 + 10\left(\rho_{\max}\frac{1+\gamma}{\min|\lambda(C)|}\right)^2 K_1\left(\alpha^2 + \frac{2\alpha^2}{\lambda_C}\frac{1}{\beta}\right). \tag{42}$$

686 Combining eq.(41) and eq.(42), we obtain the following upper bound of $\sum_{t=0}^{M-1}\|\theta_t^{(m)} - \theta^*\|^2$ in
687 terms of $\|\tilde{z}^{(m-1)}\|^2$ and $\|\tilde{\theta}^{(m-1)} - \theta^*\|^2$.

$$\frac{\lambda_{\widehat{A}}}{12}\alpha\sum_{t=0}^{M-1}\mathbb{E}_{m,0}\|\theta_t^{(m)} - \theta^*\|^2$$

$$\leq\left[1 + \alpha^2 M\cdot 5(1+\gamma)^2\rho_{\max}^2\left(1 + \frac{\gamma\rho_{\max}}{\min|\lambda(C)|}\right)^2\right]\mathbb{E}_{m,0}\|\tilde{\theta}^{(m-1)} - \theta^*\|^2 + \alpha\cdot 2K_2 + \alpha^2\cdot 5K_1$$

$$+ \frac{\alpha}{\beta}\cdot\frac{96}{\lambda_{\widehat{A}}\lambda_C}\gamma^2\rho_{\max}^2\left\{\left[1 + \left[10\beta^2 + 10\gamma^2\rho_{\max}^2\left(\alpha^2 + \frac{2\alpha^2}{\lambda_C}\frac{1}{\beta}\right)\cdot\left(\rho_{\max}\frac{1+\gamma}{\min|\lambda(C)|}\right)^2\right]M\right]\mathbb{E}_{m,0}\|\tilde{z}^{(m-1)}\|^2\right.$$

$$+ 10(1+\gamma)^2\rho_{\max}^2\cdot\left[\left(1 + \frac{\gamma\rho_{\max}}{\min|\lambda(C)|}\right)^2\left(\alpha^2 + \frac{2\alpha^2}{\lambda_C}\frac{1}{\beta}\right)\cdot\left(\rho_{\max}\frac{1+\gamma}{\min|\lambda(C)|}\right)^2\right.$$

$$\left. + \left(1 + \frac{1}{\min|\lambda(C)|}\right)^2\beta^2\right]\left(\sum_{t=0}^{M-1}\mathbb{E}_{m,0}\|\theta_t^{(m)} - \theta^*\|^2 + M\mathbb{E}_{m,0}\|\tilde{\theta}^{(m-1)} - \theta^*\|^2\right)$$

$$\left. + (2K_3 + 2K_4)\beta + 10K_5\beta^2 + 10\left(\rho_{\max}\frac{1+\gamma}{\min|\lambda(C)|}\right)^2 K_1\left(\alpha^2 + \frac{2\alpha^2}{\lambda_C}\frac{1}{\beta}\right)\right\} + \alpha^2 M\cdot 5\gamma^2\rho_{\max}^2\mathbb{E}_{m,0}\|\tilde{z}^{(m-1)}\|^2.$$

688 Re-arranging the above inequality yields that

$$\left\{\frac{\lambda_{\widehat{A}}}{12}\alpha - \frac{\alpha}{\beta}\cdot\frac{96}{\lambda_{\widehat{A}}\lambda_C}\gamma^2\rho_{\max}^2\cdot 10(1+\gamma)^2\rho_{\max}^2\cdot\left[\left(1 + \frac{\gamma\rho_{\max}}{\min|\lambda(C)|}\right)^2\left(\alpha^2 + \frac{2\alpha^2}{\lambda_C}\frac{1}{\beta}\right)\cdot\left(\rho_{\max}\frac{1+\gamma}{\min|\lambda(C)|}\right)^2\right.\right.$$

$$\left.\left. + \left(1 + \frac{1}{\min|\lambda(C)|}\right)^2\beta^2\right]\right\}\sum_{t=0}^{M-1}\mathbb{E}_{m,0}\|\theta_t^{(m)} - \theta^*\|^2$$

$$\leq\left[1 + \alpha^2 M\cdot 5(1+\gamma)^2\rho_{\max}^2\left(1 + \frac{\gamma\rho_{\max}}{\min|\lambda(C)|}\right)^2\right.$$

$$+ \frac{\alpha M}{\beta}\cdot\frac{96}{\lambda_{\widehat{A}}\lambda_C}\gamma^2\rho_{\max}^2\cdot 10(1+\gamma)^2\rho_{\max}^2\cdot\left[\left(1 + \frac{\gamma\rho_{\max}}{\min|\lambda(C)|}\right)^2\left(\alpha^2 + \frac{2\alpha^2}{\lambda_C}\frac{1}{\beta}\right)\cdot\left(\rho_{\max}\frac{1+\gamma}{\min|\lambda(C)|}\right)^2\right.$$

$$\left.\left. + \left(1 + \frac{1}{\min|\lambda(C)|}\right)^2\beta^2\right]\right]\mathbb{E}_{m,0}\|\tilde{\theta}^{(m-1)} - \theta^*\|^2 + \alpha\cdot 2K_2 + \alpha^2\cdot 5K_1$$

$$+ \left\{\frac{\alpha}{\beta}\cdot\frac{96}{\lambda_{\widehat{A}}\lambda_C}\gamma^2\rho_{\max}^2\left[1 + \left[10\beta^2 + 10\gamma^2\rho_{\max}^2\left(\alpha^2 + \frac{2\alpha^2}{\lambda_C}\frac{1}{\beta}\right)\cdot\left(\rho_{\max}\frac{1+\gamma}{\min|\lambda(C)|}\right)^2\right]M\right]\right.$$

$$\left. + \alpha^2 M\cdot 5\gamma^2\rho_{\max}^2\right\}\mathbb{E}_{m,0}\|\tilde{z}^{(m-1)}\|^2$$

$$+ \alpha\cdot\frac{96}{\lambda_{\widehat{A}}\lambda_C}\gamma^2\rho_{\max}^2(2K_3 + 2K_4) + \alpha\beta\cdot\frac{96}{\lambda_{\widehat{A}}\lambda_C}\gamma^2\rho_{\max}^2 10K_5$$

$$+ \frac{\alpha}{\beta}\left(\alpha^2 + \frac{2\alpha^2}{\lambda_C}\frac{1}{\beta}\right)\cdot\frac{96}{\lambda_{\widehat{A}}\lambda_C}\gamma^2\rho_{\max}^2 10\left(\rho_{\max}\frac{1+\gamma}{\min|\lambda(C)|}\right)^2 K_1. \tag{43}$$

689 To simplify the above inequality, note that we assume that $\frac{\lambda_{\widehat{A}}}{12}\alpha - \frac{\alpha}{\beta}\cdot\frac{96}{\lambda_{\widehat{A}}\lambda_C}\gamma^2\rho_{\max}^2\cdot 10(1+\gamma)^2\rho_{\max}^2\cdot$
690 $\left[\left(1 + \frac{\gamma\rho_{\max}}{\min|\lambda(C)|}\right)^2\left(\alpha^2 + \frac{2\alpha^2}{\lambda_C}\frac{1}{\beta}\right)\cdot\left(\rho_{\max}\frac{1+\gamma}{\min|\lambda(C)|}\right)^2 + \left(1 + \frac{1}{\min|\lambda(C)|}\right)^2\beta^2\right] \geq \frac{\lambda_{\widehat{A}}}{16}\alpha$ and $\beta \leq 1$.

Applying Jensen's inequality to the left-hand side of the above inequality, we obtain the following simplified inequality.

$$
\frac{\lambda_{\widehat{A}}}{16}\alpha M \mathbb{E}_{m,0}\|\tilde{\theta}^{(m)} - \theta^*\|^2
$$

$$
\leq \Big[1 + \alpha^2 M \cdot 5(1+\gamma)^2 \rho_{\max}^2 \Big(1 + \frac{\gamma \rho_{\max}}{\min|\lambda(C)|}\Big)^2
$$

$$
+ \alpha M \Big[\Big(1 + \frac{\gamma \rho_{\max}}{\min|\lambda(C)|}\Big)^2 \Big(1 + \frac{2}{\lambda_C}\Big) \cdot \Big(\rho_{\max}\frac{1+\gamma}{\min|\lambda(C)|}\Big)^2 \frac{\alpha^2}{\beta^2}
$$

$$
+ \Big(1 + \frac{1}{\min|\lambda(C)|}\Big)^2 \beta\Big] \cdot \frac{96}{\lambda_{\widehat{A}}\lambda_C}\gamma^2\rho_{\max}^2 \cdot 10(1+\gamma)^2\rho_{\max}^2\Big]\mathbb{E}_{m,0}\|\tilde{\theta}^{(m-1)} - \theta^*\|^2
$$

$$
+ \Big\{\frac{\alpha}{\beta}\cdot\frac{96}{\lambda_{\widehat{A}}\lambda_C}\gamma^2\rho_{\max}^2\Big[1 + \Big[10\beta^2 + 10\gamma^2\rho_{\max}^2\Big(1 + \frac{2}{\lambda_C}\Big)\cdot\Big(\rho_{\max}\frac{1+\gamma}{\min|\lambda(C)|}\Big)^2\frac{\alpha^2}{\beta}\Big]M\Big]
$$

$$
+ \alpha^2 M \cdot 5\gamma^2\rho_{\max}^2\Big\}\mathbb{E}_{m,0}\|\tilde{z}^{(m-1)}\|^2
$$

$$
+ \alpha\cdot\Big[\frac{96}{\lambda_{\widehat{A}}\lambda_C}\gamma^2\rho_{\max}^2(2K_3 + 2K_4) + 2K_2\Big] + \alpha\beta\cdot\frac{96}{\lambda_{\widehat{A}}\lambda_C}\gamma^2\rho_{\max}^2 10K_5
$$

$$
+ \frac{\alpha^3}{\beta}\Big(1 + \frac{2}{\lambda_C}\Big)\cdot\frac{96}{\lambda_{\widehat{A}}\lambda_C}\gamma^2\rho_{\max}^2 10\Big(\rho_{\max}\frac{1+\gamma}{\min|\lambda(C)|}\Big)^2 K_1 + \alpha^2\cdot 5K_1.
$$

Define $C_1 = \Big(1 + \frac{\gamma\rho_{\max}}{\min|\lambda(C)|}\Big)^2\Big(1 + \frac{2}{\lambda_C}\Big)\cdot\Big(\rho_{\max}\frac{1+\gamma}{\min|\lambda(C)|}\Big)^2\cdot\frac{96}{\lambda_{\widehat{A}}\lambda_C}\gamma^2\rho_{\max}^2\cdot 10(1+\gamma)^2\rho_{\max}^2$ and

$C_2 = \Big(1 + \frac{1}{\min|\lambda(C)|}\Big)^2\cdot\frac{96}{\lambda_{\widehat{A}}\lambda_C}\gamma^2\rho_{\max}^2\cdot 10(1+\gamma)^2\rho_{\max}^2$, and assume that

$$
D := \frac{16}{\lambda_{\widehat{A}}}\Big[\frac{1}{\alpha M} + \alpha\cdot 5(1+\gamma)^2\rho_{\max}^2\Big(1 + \frac{\gamma\rho_{\max}}{\min|\lambda(C)|}\Big)^2 + \frac{\alpha^2}{\beta^2}\cdot C_1 + \beta\cdot C_2\Big] < 1. \tag{44}
$$

Taking total expectation and dividing $\frac{\lambda_{\widehat{A}}}{16}\alpha M$ on both sides of the previous simplified inequality, we obtain that

$$
\mathbb{E}\|\tilde{\theta}^{(m)} - \theta^*\|^2
$$

$$
\leq D\cdot\mathbb{E}_{m,0}\|\tilde{\theta}^{(m-1)} - \theta^*\|^2
$$

$$
+ \frac{16}{\lambda_{\widehat{A}}}\Big\{\frac{96}{\lambda_{\widehat{A}}\lambda_C}\gamma^2\rho_{\max}^2\Big[\frac{1}{\beta M} + 10\beta + 10\gamma^2\rho_{\max}^2\Big(1 + \frac{2}{\lambda_C}\Big)\cdot\Big(\rho_{\max}\frac{1+\gamma}{\min|\lambda(C)|}\Big)^2\frac{\alpha^2}{\beta^2}\Big] + \alpha\cdot 5\gamma^2\rho_{\max}^2\Big\}\mathbb{E}\|\tilde{z}^{(m-1)}\|^2
$$

$$
+ \frac{1}{M}\cdot\frac{16}{\lambda_{\widehat{A}}}\Big[\frac{96}{\lambda_{\widehat{A}}\lambda_C}\gamma^2\rho_{\max}^2(2K_3 + 2K_4) + 2K_2\Big] + \frac{\beta}{M}\cdot\frac{16}{\lambda_{\widehat{A}}}\frac{96}{\lambda_{\widehat{A}}\lambda_C}\gamma^2\rho_{\max}^2 10K_5
$$

$$
+ \frac{\alpha^2}{\beta M}\cdot\frac{16}{\lambda_{\widehat{A}}}\Big(1 + \frac{2}{\lambda_C}\Big)\cdot\frac{96}{\lambda_{\widehat{A}}\lambda_C}\gamma^2\rho_{\max}^2 10\Big(\rho_{\max}\frac{1+\gamma}{\min|\lambda(C)|}\Big)^2 K_1 + \frac{\alpha}{M}\cdot\frac{80K_1}{\lambda_{\widehat{A}}}. \tag{45}
$$

By Lemma H.2, we have that

$$
\mathbb{E}\|\tilde{z}^{(m)}\|^2 \leq \Big(\frac{1}{M\beta}\cdot\frac{12}{\lambda_C}\Big)^m\mathbb{E}\|\tilde{z}^{(0)}\|^2
$$

$$
+ 2\cdot\Big[\beta\cdot\frac{24}{\lambda_C}H_{\text{VR}}^2 + \frac{\alpha^2}{\beta^2}\cdot\Big(1 + \frac{2}{\lambda_C}\Big)\cdot\frac{24}{\lambda_C}\Big(\rho_{\max}\frac{1+\gamma}{\min|\lambda(C)|}\Big)^2 G_{\text{VR}}^2 + \frac{1}{M}\frac{24}{\lambda_C}(K_3 + K_4)\Big],
$$

where $K_3$ is specified in eq. (71) of Lemma I.3, and $K_4$ is specified in eq. (75) of Lemma I.4. Here we assume that

$$
E := \frac{1}{M\beta}\cdot\frac{12}{\lambda_C} < 1. \tag{46}
$$

Substituting the above result of Lemma H.2 into eq.(45), we obtain that

$$
\mathbb{E}\|\tilde{\theta}^{(m)} - \theta^*\|^2
$$

$$
\leq D\cdot\mathbb{E}\|\tilde{\theta}^{(m-1)} - \theta^*\|^2
$$

$$+ \frac{16}{\lambda_{\widehat{A}}} \Big\{ \frac{96}{\lambda_{\widehat{A}}\lambda_C} \gamma^2 \rho_{\max}^2 \Big[ \frac{1}{\beta M} + 10\beta + 10\gamma^2 \rho_{\max}^2 \Big( 1 + \frac{2}{\lambda_C} \Big) \cdot \Big( \rho_{\max} \frac{1+\gamma}{\min|\lambda(C)|} \Big)^2 \frac{\alpha^2}{\beta^2} \Big] + \alpha \cdot 5\gamma^2 \rho_{\max}^2 \Big\} \Big\{ E^{m-1} \mathbb{E}\|\tilde{z}^{(0)}\|^2$$

$$+ 2 \cdot \Big[ \beta \cdot \frac{24}{\lambda_C} H_{\mathrm{VR}}^2 + \frac{\alpha^2}{\beta^2} \cdot \Big( 1 + \frac{2}{\lambda_C} \Big) \cdot \frac{24}{\lambda_C} \Big( \rho_{\max} \frac{1+\gamma}{\min|\lambda(C)|} \Big)^2 G_{\mathrm{VR}}^2 + \frac{1}{M} \frac{24}{\lambda_C} (K_3 + K_4) \Big] \Big\}$$

$$+ \frac{1}{M} \cdot \frac{16}{\lambda_{\widehat{A}}} \Big[ \frac{96}{\lambda_{\widehat{A}}\lambda_C} \gamma^2 \rho_{\max}^2 (2K_3 + 2K_4) + 2K_2 \Big] + \frac{\beta}{M} \cdot \frac{16}{\lambda_{\widehat{A}}} \frac{96}{\lambda_{\widehat{A}}\lambda_C} \gamma^2 \rho_{\max}^2 10 K_5$$

$$+ \frac{\alpha^2}{\beta M} \cdot \frac{16}{\lambda_{\widehat{A}}} \Big( 1 + \frac{2}{\lambda_C} \Big) \cdot \frac{96}{\lambda_{\widehat{A}}\lambda_C} \gamma^2 \rho_{\max}^2 10 \Big( \rho_{\max} \frac{1+\gamma}{\min|\lambda(C)|} \Big)^2 K_1 + \frac{\alpha}{M} \cdot \frac{80 K_1}{\lambda_{\widehat{A}}}.$$

701 Furthermore, we assume that $\frac{16}{\lambda_{\widehat{A}}} \big\{ \frac{96}{\lambda_{\widehat{A}}\lambda_C} \gamma^2 \rho_{\max}^2 \big[ \frac{1}{\beta M} + 10\beta + 10\gamma^2 \rho_{\max}^2 \big( 1 + \frac{2}{\lambda_C} \big) \cdot$

702 $\big( \rho_{\max} \frac{1+\gamma}{\min|\lambda(C)|} \big)^2 \frac{\alpha^2}{\beta^2} \big] + \alpha \cdot 5\gamma^2 \rho_{\max}^2 \big\} \leq 1$. Then, telescope the above inequality yields that

$$\mathbb{E}\|\tilde{\theta}^{(m)} - \theta^*\|^2$$

$$\leq D^m \cdot \mathbb{E}\|\tilde{\theta}^{(0)} - \theta^*\|^2 + \frac{D^m - E^m}{D - E} \mathbb{E}\|\tilde{z}^{(0)}\|^2$$

$$+ \frac{32}{\lambda_{\widehat{A}}} \Big\{ \frac{96}{\lambda_{\widehat{A}}\lambda_C} \gamma^2 \rho_{\max}^2 \Big[ \frac{1}{\beta M} + 10\beta + 10\gamma^2 \rho_{\max}^2 \Big( 1 + \frac{2}{\lambda_C} \Big) \cdot \Big( \rho_{\max} \frac{1+\gamma}{\min|\lambda(C)|} \Big)^2 \frac{\alpha^2}{\beta^2} \Big]$$

$$+ \alpha \cdot 5\gamma^2 \rho_{\max}^2 \Big\} \Big\{ \beta \cdot \frac{24}{\lambda_C} H_{\mathrm{VR}}^2 + \frac{\alpha^2}{\beta^2} \cdot \Big( 1 + \frac{2}{\lambda_C} \Big) \cdot \frac{24}{\lambda_C} \Big( \rho_{\max} \frac{1+\gamma}{\min|\lambda(C)|} \Big)^2 G_{\mathrm{VR}}^2 + \frac{1}{M} \frac{24}{\lambda_C} (K_3 + K_4) \Big\}$$

$$+ \frac{1}{M} \cdot \frac{16}{\lambda_{\widehat{A}}} \Big[ \frac{96}{\lambda_{\widehat{A}}\lambda_C} \gamma^2 \rho_{\max}^2 (2K_3 + 2K_4) + 2K_2 \Big] + \frac{\beta}{M} \cdot \frac{16}{\lambda_{\widehat{A}}} \frac{96}{\lambda_{\widehat{A}}\lambda_C} \gamma^2 \rho_{\max}^2 10 K_5$$

$$+ \frac{\alpha^2}{\beta M} \cdot \frac{16}{\lambda_{\widehat{A}}} \Big( 1 + \frac{2}{\lambda_C} \Big) \cdot \frac{96}{\lambda_{\widehat{A}}\lambda_C} \gamma^2 \rho_{\max}^2 10 \Big( \rho_{\max} \frac{1+\gamma}{\min|\lambda(C)|} \Big)^2 K_1 + \frac{\alpha}{M} \cdot \frac{80 K_1}{\lambda_{\widehat{A}}}.$$

703 To further simplify, note that the first two terms are in the order of $D^m$. Also, the third term
704 is a product of two curly brackets, and it is easy to check that this term is dominated by $\mathcal{O}(\beta^2)$
705 under the relation $\beta = \mathcal{O}(\alpha^{2/3})$. To further elaborate this, note that the first bracket of this
706 product is in the order of $\mathcal{O}(\frac{1}{\beta M}) + \mathcal{O}(\beta)$ and the second term of this product is in the order of
707 $\mathcal{O}(\beta)$ (by $E < 1$, we have $\frac{1}{M} \leq \mathcal{O}(\beta)$). Therefore, asymptotically the product is in the order of
708 $(\mathcal{O}(\frac{1}{\beta M}) + \mathcal{O}(\beta)) \times \mathcal{O}(\beta) = \mathcal{O}(\frac{1}{M}) + \mathcal{O}(\beta^2)$. Moreover, under the setting $\beta = \mathcal{O}(\alpha^{2/3})$, $D > E$
709 for sufficiently small $\alpha$ and $\beta$. Therefore, we have the following asymptotic bound

$$\mathbb{E}\|\tilde{\theta}^{(m)} - \theta^*\|^2 = \mathcal{O}(D^m) + \mathcal{O}(\beta^2) + \mathcal{O}(\frac{1}{M}).$$

710 Now we compute its complexity. For sufficiently small $\beta$ and sufficiently large $M$, there always
711 exists constant $I_1, I_2, I_3$ such that

$$\mathbb{E}\|\tilde{\theta}^{(m)} - \theta^*\|^2 \leq D^m I_1 + \beta^2 I_2 + \frac{1}{M} I_3.$$

712 Now we require

713     (i) $\beta^2 I_2 \leq \epsilon/3 \Rightarrow \beta \leq I_2^{1/2} \epsilon^{1/2} = \mathcal{O}(\epsilon^{1/2})$.

714     (ii) $\frac{1}{M} I_3 \leq \epsilon/3 \Rightarrow M \geq \mathcal{O}(\epsilon^{-1})$.

715     (iii) $D^m I_1 \leq \epsilon/3 \Rightarrow m \log D \leq \log \frac{\epsilon}{3 I_1} \Rightarrow m \geq \mathcal{O}(\log \epsilon^{-1})$.

716 Note that in (iii), we have used the condition that $D < 1$ and hence $\log D$ is a negative constant.
717 Since $\alpha = \mathcal{O}(\beta^{2/3})$, the condition that $D \leq 1$ requires that $M \geq \mathcal{O}(\beta^{-3/2})$, which combines with
718 (ii) requires that

$$M \geq \max\{\mathcal{O}(\epsilon^{-1}), \mathcal{O}(\beta^{-3/2})\}.$$

719 Taking into account the constraint on $\beta$ in (i), we just require that $M \geq \mathcal{O}(\epsilon^{-1})$, which leads to the
720 overall complexity result

$$mM \geq \mathcal{O}(\epsilon^{-1} \log \epsilon^{-1}).$$

## G  Proof of Corollary 4.2

**Lemma G.1** (Refined Bounds of $\mathbb{E}\|\tilde{z}\|^2$). *Under the same assumptions as those of Theorem 4.1, choose the learning rate $\alpha, \beta$ and the batch size $M$ such that all requirements of Theorem 4.1 are satisfied. Then, the following refined bound holds.*

$$\mathbb{E}\|\tilde{z}^{(m)}\|^2$$

$$\leq F^m \cdot \mathbb{E}\|\tilde{z}^{(0)}\|^2 + \frac{D^{m-1} - F^{m-1}}{D - F} \cdot \mathbb{E}\|\tilde{\theta}^{(0)} - \theta^*\|^2 + \frac{\frac{D^{m-1} - F^{m-1}}{D - F} - \frac{E^{m-1} - F^{m-1}}{E - F}}{D - E} \mathbb{E}\|\tilde{z}^{(0)}\|^2$$

$$+ \frac{1}{1 - F}\frac{24}{\lambda_C}\Big\{10(1 + \gamma)^2\rho_{\max}^2 \cdot \Big[\Big(1 + \frac{\gamma\rho_{\max}}{\min|\lambda(C)|}\Big)^2\Big(1 + \frac{2}{\lambda_C}\Big) \cdot \Big(\rho_{\max}\frac{1 + \gamma}{\min|\lambda(C)|}\Big)^2\frac{\alpha^2}{\beta^2}$$

$$+ \Big(1 + \frac{1}{\min|\lambda(C)|}\Big)^2\beta\Big] + 120(1 + \gamma)^2\rho_{\max}^2\frac{1}{\lambda_{\widehat{A}}} \cdot \Big[\Big(1 + \frac{\gamma\rho_{\max}}{\min|\lambda(C)|}\Big)^2\Big(1 + \frac{2}{\lambda_C}\Big) \cdot \Big(\rho_{\max}\frac{1 + \gamma}{\min|\lambda(C)|}\Big)^2\frac{\alpha^2}{\beta^2}$$

$$+ \Big(1 + \frac{1}{\min|\lambda(C)|}\Big)^2\beta\Big] \cdot \Big[\frac{1}{\alpha M} + \alpha \cdot 5(1 + \gamma)^2\rho_{\max}^2\Big(1 + \frac{\gamma\rho_{\max}}{\min|\lambda(C)|}\Big)^2\Big]\Big\}$$

$$\times \Big\{\frac{32}{\lambda_{\widehat{A}}}\Big\{\frac{96}{\lambda_{\widehat{A}}\lambda_C}\gamma^2\rho_{\max}^2\Big[\frac{1}{\beta M} + 10\beta + 10\gamma^2\rho_{\max}^2\Big(1 + \frac{2}{\lambda_C}\Big) \cdot \Big(\rho_{\max}\frac{1 + \gamma}{\min|\lambda(C)|}\Big)^2\frac{\alpha^2}{\beta^2}\Big] + \alpha \cdot 5\gamma^2\rho_{\max}^2\Big\}$$

$$\times \Big\{\beta \cdot \frac{24}{\lambda_C}H_{VR}^2 + \frac{\alpha^2}{\beta^2} \cdot \Big(1 + \frac{2}{\lambda_C}\Big) \cdot \frac{24}{\lambda_C}\Big(\rho_{\max}\frac{1 + \gamma}{\min|\lambda(C)|}\Big)^2 G_{VR}^2 + \frac{1}{M}\frac{24}{\lambda_C}(K_3 + K_4)\Big\}$$

$$+ \frac{1}{M} \cdot \frac{16}{\lambda_{\widehat{A}}}\Big[\frac{96}{\lambda_{\widehat{A}}\lambda_C}\gamma^2\rho_{\max}^2(2K_3 + 2K_4) + 2K_2\Big] + \frac{\beta}{M} \cdot \frac{16}{\lambda_{\widehat{A}}}\frac{96}{\lambda_{\widehat{A}}\lambda_C}\gamma^2\rho_{\max}^2 10K_5$$

$$+ \frac{\alpha^2}{\beta M} \cdot \frac{16}{\lambda_{\widehat{A}}}\Big(1 + \frac{2}{\lambda_C}\Big) \cdot \frac{96}{\lambda_{\widehat{A}}\lambda_C}\gamma^2\rho_{\max}^2 10\Big(\rho_{\max}\frac{1 + \gamma}{\min|\lambda(C)|}\Big)^2 K_1 + \frac{\alpha}{M} \cdot \frac{80K_1}{\lambda_{\widehat{A}}}\Big\}$$

$$+ \frac{1}{1 - F}\Big\{\frac{1}{M}\frac{48K_3 + 48K_4}{\lambda_C} + \frac{\beta}{M}\frac{240K_5}{\lambda_C} + \frac{\alpha^2}{\beta^2}\frac{1}{M}\frac{240}{\lambda_C}\Big(\rho_{\max}\frac{1 + \gamma}{\min|\lambda(C)|}\Big)^2 K_1\Big(1 + \frac{2}{\lambda_C}\Big)$$

$$+ \frac{1}{M}(1 + \gamma)^2\rho_{\max}^2\frac{2880}{\lambda_{\widehat{A}}\lambda_C} \cdot \Big[\Big(1 + \frac{\gamma\rho_{\max}}{\min|\lambda(C)|}\Big)^2\Big(1 + \frac{2}{\lambda_C}\Big) \cdot \Big(\rho_{\max}\frac{1 + \gamma}{\min|\lambda(C)|}\Big)^2\frac{\alpha^2}{\beta^2}$$

$$+ \Big(1 + \frac{1}{\min|\lambda(C)|}\Big)^2\beta\Big] \cdot \Big\{2K_2 + \alpha \cdot 5K_1\Big\}\Big\}.$$

*Proof.* Based on the preliminary bound of $\sum_{t=0}^{M-1}\mathbb{E}_{m,0}\|z_t^{(m)}\|^2$ (Lemma H.3), we have

$$\frac{\lambda_C}{16}\beta\sum_{t=0}^{M-1}\mathbb{E}_{m,0}\|z_t^{(m)}\|^2$$

$$\leq \Big[1 + \Big[10\beta^2 + 10\gamma^2\rho_{\max}^2\Big(\alpha^2 + \frac{2\alpha^2}{\lambda_C}\frac{1}{\beta}\Big) \cdot \Big(\rho_{\max}\frac{1 + \gamma}{\min|\lambda(C)|}\Big)^2\Big]M\Big]\mathbb{E}_{m,0}\|\tilde{z}^{(m-1)}\|^2$$

$$+ 10(1 + \gamma)^2\rho_{\max}^2 \cdot \Big[\Big(1 + \frac{\gamma\rho_{\max}}{\min|\lambda(C)|}\Big)^2\Big(\alpha^2 + \frac{2\alpha^2}{\lambda_C}\frac{1}{\beta}\Big) \cdot \Big(\rho_{\max}\frac{1 + \gamma}{\min|\lambda(C)|}\Big)^2$$

$$+ \Big(1 + \frac{1}{\min|\lambda(C)|}\Big)^2\beta^2\Big]\Big(\sum_{t=0}^{M-1}\mathbb{E}_{m,0}\|\theta_t^{(m)} - \theta^*\|^2 + M\mathbb{E}_{m,0}\|\tilde{\theta}^{(m-1)} - \theta^*\|^2\Big)$$

$$+ (2K_3 + 2K_4)\beta + 10K_5\beta^2 + 10\Big(\rho_{\max}\frac{1 + \gamma}{\min|\lambda(C)|}\Big)^2 K_1\Big(\alpha^2 + \frac{2\alpha^2}{\lambda_C}\frac{1}{\beta}\Big), \qquad (47)$$

where $K_1$ is specified in eq. (63) of Lemma I.1, $K_3$ is specified in eq. (71) of Lemma I.3, $K_4$ is specified in eq. (75) of Lemma I.4, and $K_5$ is specified in eq. (76) of Lemma I.5. Note that by Lemma H.1, we have the preliminary bound of $\sum_{t=0}^{M-1}\mathbb{E}_{m,0}\|\theta_t^{(m)} - \theta^*\|^2$:

$$\frac{\lambda_{\widehat{A}}}{12}\alpha\sum_{t=0}^{M-1}\mathbb{E}_{m,0}\|\theta_t^{(m)} - \theta^*\|^2$$

$$\leq \Big[1 + \alpha^2 M \cdot 5(1+\gamma)^2 \rho_{\max}^2 \Big(1 + \frac{\gamma\rho_{\max}}{\min|\lambda(C)|}\Big)^2\Big]\mathbb{E}_{m,0}\|\tilde{\theta}^{(m-1)} - \theta^*\|^2 + \alpha \cdot 2K_2 + \alpha^2 \cdot 5K_1$$

$$+ \alpha \cdot \frac{6}{\lambda_{\widehat{A}}}\gamma^2\rho_{\max}^2 \sum_{t=0}^{M-1} \mathbb{E}_{m,0}\|z_t^{(m)}\|^2 + \alpha^2 M \cdot 5\gamma^2\rho_{\max}^2 \mathbb{E}_{m,0}\|\tilde{z}^{(m-1)}\|^2, \tag{48}$$

where $K_1$ is specified in eq. (63) of Lemma I.1, and $K_2$ is specified in eq. (67) of Lemma I.2. Now we combine eq.(47) and eq.(48) to obtain that

$$\frac{\lambda_C}{16}\beta \sum_{t=0}^{M-1} \mathbb{E}_{m,0}\|z_t^{(m)}\|^2$$

$$\leq \Big[1 + \Big[10\beta^2 + 10\gamma^2\rho_{\max}^2\Big(\alpha^2 + \frac{2\alpha^2}{\lambda_C}\frac{1}{\beta}\Big) \cdot \Big(\rho_{\max}\frac{1+\gamma}{\min|\lambda(C)|}\Big)^2\Big]M\Big]\mathbb{E}_{m,0}\|\tilde{z}^{(m-1)}\|^2$$

$$+ \frac{1}{\alpha} \cdot 120(1+\gamma)^2\rho_{\max}^2 \frac{1}{\lambda_{\widehat{A}}} \cdot \Big[\Big(1 + \frac{\gamma\rho_{\max}}{\min|\lambda(C)|}\Big)^2\Big(\alpha^2 + \frac{2\alpha^2}{\lambda_C}\frac{1}{\beta}\Big) \cdot \Big(\rho_{\max}\frac{1+\gamma}{\min|\lambda(C)|}\Big)^2$$

$$+ \Big(1 + \frac{1}{\min|\lambda(C)|}\Big)^2\beta^2\Big] \cdot \Big\{\Big[1 + \alpha^2 M \cdot 5(1+\gamma)^2\rho_{\max}^2\Big(1 + \frac{\gamma\rho_{\max}}{\min|\lambda(C)|}\Big)^2\Big]\mathbb{E}_{m,0}\|\tilde{\theta}^{(m-1)} - \theta^*\|^2$$

$$+ \alpha \cdot 2K_2 + \alpha^2 \cdot 5K_1 + \alpha \cdot \frac{6}{\lambda_{\widehat{A}}}\gamma^2\rho_{\max}^2 \sum_{t=0}^{M-1} \mathbb{E}_{m,0}\|z_t^{(m)}\|^2 + \alpha^2 M \cdot 5\gamma^2\rho_{\max}^2 \mathbb{E}_{m,0}\|\tilde{z}^{(m-1)}\|^2\Big\}$$

$$+ 10(1+\gamma)^2\rho_{\max}^2 \cdot \Big[\Big(1 + \frac{\gamma\rho_{\max}}{\min|\lambda(C)|}\Big)^2\Big(\alpha^2 + \frac{2\alpha^2}{\lambda_C}\frac{1}{\beta}\Big) \cdot \Big(\rho_{\max}\frac{1+\gamma}{\min|\lambda(C)|}\Big)^2$$

$$+ \Big(1 + \frac{1}{\min|\lambda(C)|}\Big)^2\beta^2\Big] \cdot M\mathbb{E}_{m,0}\|\tilde{\theta}^{(m-1)} - \theta^*\|^2$$

$$+ (2K_3 + 2K_4)\beta + 10K_5\beta^2 + 10\Big(\rho_{\max}\frac{1+\gamma}{\min|\lambda(C)|}\Big)^2 K_1\Big(\alpha^2 + \frac{2\alpha^2}{\lambda_C}\frac{1}{\beta}\Big).$$

Next, we simplify the above inequality. We first expand the curly brackets and simplify, we obtain that

$$\Big\{\frac{\lambda_C}{16}\beta - 120(1+\gamma)^2\rho_{\max}^2 \frac{1}{\lambda_{\widehat{A}}} \cdot \Big[\Big(1 + \frac{\gamma\rho_{\max}}{\min|\lambda(C)|}\Big)^2\Big(\alpha^2 + \frac{2\alpha^2}{\lambda_C}\frac{1}{\beta}\Big) \cdot \Big(\rho_{\max}\frac{1+\gamma}{\min|\lambda(C)|}\Big)^2$$

$$+ \Big(1 + \frac{1}{\min|\lambda(C)|}\Big)^2\beta^2\Big] \cdot \frac{6}{\lambda_{\widehat{A}}}\gamma^2\rho_{\max}^2\Big\} \sum_{t=0}^{M-1} \mathbb{E}_{m,0}\|z_t^{(m)}\|^2$$

$$\leq \Big[1 + \Big[10\beta^2 + 10\gamma^2\rho_{\max}^2\Big(\alpha^2 + \frac{2\alpha^2}{\lambda_C}\frac{1}{\beta}\Big) \cdot \Big(\rho_{\max}\frac{1+\gamma}{\min|\lambda(C)|}\Big)^2\Big]M$$

$$+ 120(1+\gamma)^2\rho_{\max}^2 \frac{1}{\lambda_{\widehat{A}}} \cdot \Big[\Big(1 + \frac{\gamma\rho_{\max}}{\min|\lambda(C)|}\Big)^2\Big(\alpha^2 + \frac{2\alpha^2}{\lambda_C}\frac{1}{\beta}\Big) \cdot \Big(\rho_{\max}\frac{1+\gamma}{\min|\lambda(C)|}\Big)^2$$

$$+ \Big(1 + \frac{1}{\min|\lambda(C)|}\Big)^2\beta^2\Big] \cdot \alpha M \cdot 5\gamma^2\rho_{\max}^2\Big]\mathbb{E}_{m,0}\|\tilde{z}^{(m-1)}\|^2$$

$$+ 120(1+\gamma)^2\rho_{\max}^2 \frac{1}{\lambda_{\widehat{A}}} \cdot \Big[\Big(1 + \frac{\gamma\rho_{\max}}{\min|\lambda(C)|}\Big)^2\Big(\alpha^2 + \frac{2\alpha^2}{\lambda_C}\frac{1}{\beta}\Big) \cdot \Big(\rho_{\max}\frac{1+\gamma}{\min|\lambda(C)|}\Big)^2 + \Big(1 + \frac{1}{\min|\lambda(C)|}\Big)^2\beta^2\Big]$$

$$\times \Big[2K_2 + \alpha \cdot 5K_1\Big]$$

$$+ \Big\{10(1+\gamma)^2\rho_{\max}^2 \cdot \Big[\Big(1 + \frac{\gamma\rho_{\max}}{\min|\lambda(C)|}\Big)^2\Big(\alpha^2 + \frac{2\alpha^2}{\lambda_C}\frac{1}{\beta}\Big) \cdot \Big(\rho_{\max}\frac{1+\gamma}{\min|\lambda(C)|}\Big)^2$$

$$+ \Big(1 + \frac{1}{\min|\lambda(C)|}\Big)^2\beta^2\Big] \cdot M + \frac{1}{\alpha} \cdot 120(1+\gamma)^2\rho_{\max}^2 \frac{1}{\lambda_{\widehat{A}}} \cdot \Big[\Big(1 + \frac{\gamma\rho_{\max}}{\min|\lambda(C)|}\Big)^2\Big(\alpha^2 + \frac{2\alpha^2}{\lambda_C}\frac{1}{\beta}\Big) \cdot \Big(\rho_{\max}\frac{1+\gamma}{\min|\lambda(C)|}\Big)^2$$

$$+ \Big(1 + \frac{1}{\min|\lambda(C)|}\Big)^2\beta^2\Big] \cdot \Big[1 + \alpha^2 M \cdot 5(1+\gamma)^2\rho_{\max}^2\Big(1 + \frac{\gamma\rho_{\max}}{\min|\lambda(C)|}\Big)^2\Big]\Big\}\mathbb{E}_{m,0}\|\tilde{\theta}^{(m-1)} - \theta^*\|^2$$

$$+ (2K_3 + 2K_4)\beta + 10K_5\beta^2 + 10\Big(\rho_{\max}\frac{1+\gamma}{\min|\lambda(C)|}\Big)^2 K_1\Big(\alpha^2 + \frac{2\alpha^2}{\lambda_C}\frac{1}{\beta}\Big)$$

Assume $\frac{\lambda_C}{16}\beta - 120(1+\gamma)^2\rho_{\max}^2\frac{1}{\lambda_{\widehat{A}}}\cdot\Big[\Big(1+\frac{\gamma\rho_{\max}}{\min|\lambda(C)|}\Big)^2\Big(\alpha^2+\frac{2\alpha^2}{\lambda_C}\frac{1}{\beta}\Big)\cdot\Big(\rho_{\max}\frac{1+\gamma}{\min|\lambda(C)|}\Big)^2+\Big(1+\frac{1}{\min|\lambda(C)|}\Big)^2\beta^2\Big]\cdot\frac{6}{\lambda_{\widehat{A}}}\gamma^2\rho_{\max}^2\geq\frac{\lambda_C}{24}\beta$ and define

$$F:=\frac{24}{\lambda_C}\cdot\Big[\frac{1}{\beta M}+10\beta+10\gamma^2\rho_{\max}^2\big(1+\frac{1}{\lambda_C}\big)\cdot\Big(\rho_{\max}\frac{1+\gamma}{\min|\lambda(C)|}\Big)^2\frac{\alpha^2}{\beta^2}$$
$$+\alpha\cdot120(1+\gamma)^2\rho_{\max}^2\frac{1}{\lambda_{\widehat{A}}}\cdot\Big[\Big(1+\frac{\gamma\rho_{\max}}{\min|\lambda(C)|}\Big)^2\big(1+\frac{2}{\lambda_C}\big)\cdot\Big(\rho_{\max}\frac{1+\gamma}{\min|\lambda(C)|}\Big)^2\frac{\alpha^2}{\beta^2}$$
$$++\Big(1+\frac{1}{\min|\lambda(C)|}\Big)^2\beta\Big]\cdot5\gamma^2\rho_{\max}^2\Big].\qquad(49)$$

Also, assume that $F<1$. Applying Jensen's inequality on the left-hand side of the above inequality and dividing $\frac{\lambda_C}{24}\beta M$ on both sides, we obtain that

$$\mathbb{E}_{m,0}\|\tilde{z}^{(m)}\|^2$$
$$\leq F\cdot\mathbb{E}_{m,0}\|\tilde{z}^{(m-1)}\|^2$$
$$+(1+\gamma)^2\rho_{\max}^2\frac{2880}{\lambda_{\widehat{A}}\lambda_C}\cdot\Big[\Big(1+\frac{\gamma\rho_{\max}}{\min|\lambda(C)|}\Big)^2\big(1+\frac{2}{\lambda_C}\big)\cdot\Big(\rho_{\max}\frac{1+\gamma}{\min|\lambda(C)|}\Big)^2\frac{\alpha^2}{\beta^2}+\Big(1+\frac{1}{\min|\lambda(C)|}\Big)^2\beta\Big]$$
$$\times\Big[2K_2+\alpha\cdot5K_1\Big]\frac{1}{M}$$
$$+\frac{24}{\lambda_C}\Big\{10(1+\gamma)^2\rho_{\max}^2\cdot\Big[\Big(1+\frac{\gamma\rho_{\max}}{\min|\lambda(C)|}\Big)^2\big(1+\frac{2}{\lambda_C}\big)\cdot\Big(\rho_{\max}\frac{1+\gamma}{\min|\lambda(C)|}\Big)^2\frac{\alpha^2}{\beta^2}$$
$$+\Big(1+\frac{1}{\min|\lambda(C)|}\Big)^2\beta\Big]+120(1+\gamma)^2\rho_{\max}^2\frac{1}{\lambda_{\widehat{A}}}\cdot\Big[\Big(1+\frac{\gamma\rho_{\max}}{\min|\lambda(C)|}\Big)^2\big(1+\frac{2}{\lambda_C}\big)\cdot\Big(\rho_{\max}\frac{1+\gamma}{\min|\lambda(C)|}\Big)^2\frac{\alpha^2}{\beta^2}$$
$$+\Big(1+\frac{1}{\min|\lambda(C)|}\Big)^2\beta\Big]\cdot\Big[\frac{1}{\alpha M}+\alpha\cdot5(1+\gamma)^2\rho_{\max}^2\Big(1+\frac{\gamma\rho_{\max}}{\min|\lambda(C)|}\Big)^2\Big]\Big\}\mathbb{E}_{m,0}\|\tilde{\theta}^{(m-1)}-\theta^*\|^2$$
$$+\frac{1}{M}\frac{48K_3+48K_4}{\lambda_C}+\frac{\beta}{M}\frac{240K_5}{\lambda_C}+\frac{\alpha^2}{\beta^2}\frac{1}{M}\frac{240}{\lambda_C}\Big(\rho_{\max}\frac{1+\gamma}{\min|\lambda(C)|}\Big)^2K_1\big(1+\frac{2}{\lambda_C}\big),\qquad(50)$$

where we also use the fact that $\beta\leq1$. Recall that we have obtain the final bound of $\mathbb{E}\|\tilde{\theta}^{(m-1)}-\theta^*\|^2$ in Theorem 4.1 as follows:

$$\mathbb{E}\|\tilde{\theta}^{(m)}-\theta^*\|^2$$
$$\leq D^m\cdot\mathbb{E}\|\tilde{\theta}^{(0)}-\theta^*\|^2+\frac{D^m-E^m}{D-E}\mathbb{E}\|\tilde{z}^{(0)}\|^2$$
$$+\frac{32}{\lambda_{\widehat{A}}}\Big\{\frac{96}{\lambda_{\widehat{A}}\lambda_C}\gamma^2\rho_{\max}^2\Big[\frac{1}{\beta M}+10\beta+10\gamma^2\rho_{\max}^2\big(1+\frac{2}{\lambda_C}\big)\cdot\Big(\rho_{\max}\frac{1+\gamma}{\min|\lambda(C)|}\Big)^2\frac{\alpha^2}{\beta^2}\Big]+\alpha\cdot5\gamma^2\rho_{\max}^2\Big\}$$
$$\times\Big\{\beta\cdot\frac{24}{\lambda_C}H_{\mathrm{VR}}^2+\frac{\alpha^2}{\beta^2}\cdot\big(1+\frac{2}{\lambda_C}\big)\cdot\frac{24}{\lambda_C}\Big(\rho_{\max}\frac{1+\gamma}{\min|\lambda(C)|}\Big)^2G_{\mathrm{VR}}^2+\frac{1}{M}\frac{24}{\lambda_C}(K_3+K_4)\Big\}$$
$$+\frac{1}{M}\cdot\frac{16}{\lambda_{\widehat{A}}}\Big[\frac{96}{\lambda_{\widehat{A}}\lambda_C}\gamma^2\rho_{\max}^2(2K_3+2K_4)+2K_2\Big]+\frac{\beta}{M}\cdot\frac{16}{\lambda_{\widehat{A}}}\frac{96}{\lambda_{\widehat{A}}\lambda_C}\gamma^2\rho_{\max}^210K_5$$
$$+\frac{\alpha^2}{\beta M}\cdot\frac{16}{\lambda_{\widehat{A}}}\big(1+\frac{2}{\lambda_C}\big)\cdot\frac{96}{\lambda_{\widehat{A}}\lambda_C}\gamma^2\rho_{\max}^210\Big(\rho_{\max}\frac{1+\gamma}{\min|\lambda(C)|}\Big)^2K_1+\frac{\alpha}{M}\cdot\frac{80K_1}{\lambda_{\widehat{A}}}.\qquad(51)$$

Taking total expectation on both sides of eq.(50) and applying eq.(51), we obtain that

$$\mathbb{E}\|\tilde{z}^{(m)}\|^2$$
$$\leq F\cdot\mathbb{E}\|\tilde{z}^{(m-1)}\|^2$$
$$+(1+\gamma)^2\rho_{\max}^2\frac{2880}{\lambda_{\widehat{A}}\lambda_C}\cdot\Big[\Big(1+\frac{\gamma\rho_{\max}}{\min|\lambda(C)|}\Big)^2\big(1+\frac{2}{\lambda_C}\big)\cdot\Big(\rho_{\max}\frac{1+\gamma}{\min|\lambda(C)|}\Big)^2\frac{\alpha^2}{\beta^2}+\Big(1+\frac{1}{\min|\lambda(C)|}\Big)^2\beta\Big]$$

$$\times \left\{ 2K_2 + \alpha \cdot 5K_1 \right\} \frac{1}{M}$$

$$+ \frac{24}{\lambda_C} \Big\{ 10(1+\gamma)^2 \rho_{\max}^2 \cdot \Big[ \Big( 1 + \frac{\gamma \rho_{\max}}{\min |\lambda(C)|} \Big)^2 \Big( 1 + \frac{2}{\lambda_C} \Big) \cdot \Big( \rho_{\max} \frac{1+\gamma}{\min |\lambda(C)|} \Big)^2 \frac{\alpha^2}{\beta^2}$$

$$+ \Big( 1 + \frac{1}{\min |\lambda(C)|} \Big)^2 \beta \Big] + 120(1+\gamma)^2 \rho_{\max}^2 \frac{1}{\lambda_{\widehat{A}}} \cdot \Big[ \Big( 1 + \frac{\gamma \rho_{\max}}{\min |\lambda(C)|} \Big)^2 \Big( 1 + \frac{2}{\lambda_C} \Big) \cdot \Big( \rho_{\max} \frac{1+\gamma}{\min |\lambda(C)|} \Big)^2 \frac{\alpha^2}{\beta^2}$$

$$+ \Big( 1 + \frac{1}{\min |\lambda(C)|} \Big)^2 \beta \Big] \cdot \Big[ \frac{1}{\alpha M} + \alpha \cdot 5(1+\gamma)^2 \rho_{\max}^2 \Big( 1 + \frac{\gamma \rho_{\max}}{\min |\lambda(C)|} \Big)^2 \Big] \Big\}$$

$$\times \Big\{ D^{m-1} \cdot \mathbb{E} \| \tilde{\theta}^{(0)} - \theta^* \|^2 + \frac{D^{m-1} - E^{m-1}}{D - E} \mathbb{E} \| \tilde{z}^{(0)} \|^2$$

$$+ \frac{32}{\lambda_{\widehat{A}}} \Big\{ \frac{96}{\lambda_{\widehat{A}} \lambda_C} \gamma^2 \rho_{\max}^2 \Big[ \frac{1}{\beta M} + 10\beta + 10\gamma^2 \rho_{\max}^2 \Big( 1 + \frac{2}{\lambda_C} \Big) \cdot \Big( \rho_{\max} \frac{1+\gamma}{\min |\lambda(C)|} \Big)^2 \frac{\alpha^2}{\beta^2} \Big] + \alpha \cdot 5\gamma^2 \rho_{\max}^2 \Big\}$$

$$\times \Big\{ \beta \cdot \frac{24}{\lambda_C} H_{\mathrm{VR}}^2 + \frac{\alpha^2}{\beta^2} \cdot \Big( 1 + \frac{2}{\lambda_C} \Big) \cdot \frac{24}{\lambda_C} \Big( \rho_{\max} \frac{1+\gamma}{\min |\lambda(C)|} \Big)^2 G_{\mathrm{VR}}^2 + \frac{1}{M} \frac{24}{\lambda_C} (K_3 + K_4) \Big\}$$

$$+ \frac{1}{M} \cdot \frac{16}{\lambda_{\widehat{A}}} \Big[ \frac{96}{\lambda_{\widehat{A}} \lambda_C} \gamma^2 \rho_{\max}^2 (2K_3 + 2K_4) + 2K_2 \Big] + \frac{\beta}{M} \cdot \frac{16}{\lambda_{\widehat{A}}} \frac{96}{\lambda_{\widehat{A}} \lambda_C} \gamma^2 \rho_{\max}^2 10 K_5$$

$$+ \frac{\alpha^2}{\beta M} \cdot \frac{16}{\lambda_{\widehat{A}}} \Big( 1 + \frac{2}{\lambda_C} \Big) \cdot \frac{96}{\lambda_{\widehat{A}} \lambda_C} \gamma^2 \rho_{\max}^2 10 \Big( \rho_{\max} \frac{1+\gamma}{\min |\lambda(C)|} \Big)^2 K_1 + \frac{\alpha}{M} \cdot \frac{80 K_1}{\lambda_{\widehat{A}}} \Big\}$$

$$+ \frac{1}{M} \frac{48 K_3 + 48 K_4}{\lambda_C} + \frac{\beta}{M} \frac{240 K_5}{\lambda_C} + \frac{\alpha^2}{\beta^2} \frac{1}{M} \frac{240}{\lambda_C} \Big( \rho_{\max} \frac{1+\gamma}{\min |\lambda(C)|} \Big)^2 K_1 \Big( 1 + \frac{2}{\lambda_C} \Big)$$

$$\leq F \cdot \mathbb{E} \| \tilde{z}^{(m-1)} \|^2 + D^{m-1} \cdot \mathbb{E} \| \tilde{\theta}^{(0)} - \theta^* \|^2 + \frac{D^{m-1} - E^{m-1}}{D - E} \mathbb{E} \| \tilde{z}^{(0)} \|^2$$

$$+ (1+\gamma)^2 \rho_{\max}^2 \frac{2880}{\lambda_{\widehat{A}} \lambda_C} \cdot \Big[ \Big( 1 + \frac{\gamma \rho_{\max}}{\min |\lambda(C)|} \Big)^2 \Big( 1 + \frac{2}{\lambda_C} \Big) \cdot \Big( \rho_{\max} \frac{1+\gamma}{\min |\lambda(C)|} \Big)^2 \frac{\alpha^2}{\beta^2} + \Big( 1 + \frac{1}{\min |\lambda(C)|} \Big)^2 \beta \Big]$$

$$\times \Big\{ 2K_2 + \alpha \cdot 5K_1 \Big\} \frac{1}{M}$$

$$+ \frac{24}{\lambda_C} \Big\{ 10(1+\gamma)^2 \rho_{\max}^2 \cdot \Big[ \Big( 1 + \frac{\gamma \rho_{\max}}{\min |\lambda(C)|} \Big)^2 \Big( 1 + \frac{2}{\lambda_C} \Big) \cdot \Big( \rho_{\max} \frac{1+\gamma}{\min |\lambda(C)|} \Big)^2 \frac{\alpha^2}{\beta^2}$$

$$+ \Big( 1 + \frac{1}{\min |\lambda(C)|} \Big)^2 \beta \Big] + 120(1+\gamma)^2 \rho_{\max}^2 \frac{1}{\lambda_{\widehat{A}}} \cdot \Big[ \Big( 1 + \frac{\gamma \rho_{\max}}{\min |\lambda(C)|} \Big)^2 \Big( 1 + \frac{2}{\lambda_C} \Big) \cdot \Big( \rho_{\max} \frac{1+\gamma}{\min |\lambda(C)|} \Big)^2 \frac{\alpha^2}{\beta^2}$$

$$+ \Big( 1 + \frac{1}{\min |\lambda(C)|} \Big)^2 \beta \Big] \cdot \Big[ \frac{1}{\alpha M} + \alpha \cdot 5(1+\gamma)^2 \rho_{\max}^2 \Big( 1 + \frac{\gamma \rho_{\max}}{\min |\lambda(C)|} \Big)^2 \Big] \Big\}$$

$$\times \Big\{ \frac{32}{\lambda_{\widehat{A}}} \Big\{ \frac{96}{\lambda_{\widehat{A}} \lambda_C} \gamma^2 \rho_{\max}^2 \Big[ \frac{1}{\beta M} + 10\beta + 10\gamma^2 \rho_{\max}^2 \Big( 1 + \frac{2}{\lambda_C} \Big) \cdot \Big( \rho_{\max} \frac{1+\gamma}{\min |\lambda(C)|} \Big)^2 \frac{\alpha^2}{\beta^2} \Big] + \alpha \cdot 5\gamma^2 \rho_{\max}^2 \Big\}$$

$$\times \Big\{ \beta \cdot \frac{24}{\lambda_C} H_{\mathrm{VR}}^2 + \frac{\alpha^2}{\beta^2} \cdot \Big( 1 + \frac{2}{\lambda_C} \Big) \cdot \frac{24}{\lambda_C} \Big( \rho_{\max} \frac{1+\gamma}{\min |\lambda(C)|} \Big)^2 G_{\mathrm{VR}}^2 + \frac{1}{M} \frac{24}{\lambda_C} (K_3 + K_4) \Big\}$$

$$+ \frac{1}{M} \cdot \frac{16}{\lambda_{\widehat{A}}} \Big[ \frac{96}{\lambda_{\widehat{A}} \lambda_C} \gamma^2 \rho_{\max}^2 (2K_3 + 2K_4) + 2K_2 \Big] + \frac{\beta}{M} \cdot \frac{16}{\lambda_{\widehat{A}}} \frac{96}{\lambda_{\widehat{A}} \lambda_C} \gamma^2 \rho_{\max}^2 10 K_5$$

$$+ \frac{\alpha^2}{\beta M} \cdot \frac{16}{\lambda_{\widehat{A}}} \Big( 1 + \frac{2}{\lambda_C} \Big) \cdot \frac{96}{\lambda_{\widehat{A}} \lambda_C} \gamma^2 \rho_{\max}^2 10 \Big( \rho_{\max} \frac{1+\gamma}{\min |\lambda(C)|} \Big)^2 K_1 + \frac{\alpha}{M} \cdot \frac{80 K_1}{\lambda_{\widehat{A}}} \Big\}$$

$$+ \frac{1}{M} \frac{48 K_3 + 48 K_4}{\lambda_C} + \frac{\beta}{M} \frac{240 K_5}{\lambda_C} + \frac{\alpha^2}{\beta^2} \frac{1}{M} \frac{240}{\lambda_C} \Big( \rho_{\max} \frac{1+\gamma}{\min |\lambda(C)|} \Big)^2 K_1 \Big( 1 + \frac{2}{\lambda_C} \Big)$$

where in the second step we assume $\frac{24}{\lambda_C} \Big\{ 10(1+\gamma)^2 \rho_{\max}^2 \cdot \Big[ \Big( 1 + \frac{\gamma \rho_{\max}}{\min |\lambda(C)|} \Big)^2 \Big( 1 + \frac{2}{\lambda_C} \Big) \cdot$ $\Big( \rho_{\max} \frac{1+\gamma}{\min |\lambda(C)|} \Big)^2 \frac{\alpha^2}{\beta^2} + \Big( 1 + \frac{1}{\min |\lambda(C)|} \Big)^2 \beta \Big] + 120(1+\gamma)^2 \rho_{\max}^2 \frac{1}{\lambda_{\widehat{A}}} \cdot \Big[ \Big( 1 + \frac{\gamma \rho_{\max}}{\min |\lambda(C)|} \Big)^2 \Big( 1 + \frac{2}{\lambda_C} \Big) \cdot$ $\Big( \rho_{\max} \frac{1+\gamma}{\min |\lambda(C)|} \Big)^2 \frac{\alpha^2}{\beta^2} + \Big( 1 + \frac{1}{\min |\lambda(C)|} \Big)^2 \beta \Big] \cdot \Big[ \frac{1}{\alpha M} + \alpha \cdot 5(1+\gamma)^2 \rho_{\max}^2 \Big( 1 + \frac{\gamma \rho_{\max}}{\min |\lambda(C)|} \Big)^2 \Big] \Big\} \leq 1.$

743 Lastly, we telescope the above inequality and obtain that

$$\mathbb{E}\|\tilde{z}^{(m)}\|^2$$

$$\leq F^m \cdot \mathbb{E}\|\tilde{z}^{(0)}\|^2 + \frac{D^{m-1} - F^{m-1}}{D - F} \cdot \mathbb{E}\|\tilde{\theta}^{(0)} - \theta^*\|^2 + \frac{\frac{D^{m-1} - F^{m-1}}{D - F} - \frac{E^{m-1} - F^{m-1}}{E - F}}{D - E} \mathbb{E}\|\tilde{z}^{(0)}\|^2$$

$$+ \frac{1}{1 - F}\frac{24}{\lambda_C}\Big\{10(1 + \gamma)^2\rho_{\max}^2 \cdot \Big[\Big(1 + \frac{\gamma\rho_{\max}}{\min|\lambda(C)|}\Big)^2\Big(1 + \frac{2}{\lambda_C}\Big) \cdot \Big(\rho_{\max}\frac{1 + \gamma}{\min|\lambda(C)|}\Big)^2\frac{\alpha^2}{\beta^2}$$

$$+ \Big(1 + \frac{1}{\min|\lambda(C)|}\Big)^2\beta\Big] + 120(1 + \gamma)^2\rho_{\max}^2\frac{1}{\lambda_{\widehat{A}}} \cdot \Big[\Big(1 + \frac{\gamma\rho_{\max}}{\min|\lambda(C)|}\Big)^2\Big(1 + \frac{2}{\lambda_C}\Big) \cdot \Big(\rho_{\max}\frac{1 + \gamma}{\min|\lambda(C)|}\Big)^2\frac{\alpha^2}{\beta^2}$$

$$+ \Big(1 + \frac{1}{\min|\lambda(C)|}\Big)^2\beta\Big] \cdot \Big[\frac{1}{\alpha M} + \alpha \cdot 5(1 + \gamma)^2\rho_{\max}^2\Big(1 + \frac{\gamma\rho_{\max}}{\min|\lambda(C)|}\Big)^2\Big]\Big\}$$

$$\times \Big\{\frac{32}{\lambda_{\widehat{A}}}\Big\{\frac{96}{\lambda_{\widehat{A}}\lambda_C}\gamma^2\rho_{\max}^2\Big[\frac{1}{\beta M} + 10\beta + 10\gamma^2\rho_{\max}^2\Big(1 + \frac{2}{\lambda_C}\Big) \cdot \Big(\rho_{\max}\frac{1 + \gamma}{\min|\lambda(C)|}\Big)^2\frac{\alpha^2}{\beta^2}\Big] + \alpha \cdot 5\gamma^2\rho_{\max}^2\Big\}$$

$$\times \Big\{\beta \cdot \frac{24}{\lambda_C}H_{\mathrm{VR}}^2 + \frac{\alpha^2}{\beta^2} \cdot \Big(1 + \frac{2}{\lambda_C}\Big) \cdot \frac{24}{\lambda_C}\Big(\rho_{\max}\frac{1 + \gamma}{\min|\lambda(C)|}\Big)^2 G_{\mathrm{VR}}^2 + \frac{1}{M}\frac{24}{\lambda_C}(K_3 + K_4)\Big\}$$

$$+ \frac{1}{M} \cdot \frac{16}{\lambda_{\widehat{A}}}\Big[\frac{96}{\lambda_{\widehat{A}}\lambda_C}\gamma^2\rho_{\max}^2(2K_3 + 2K_4) + 2K_2\Big] + \frac{\beta}{M} \cdot \frac{16}{\lambda_{\widehat{A}}}\frac{96}{\lambda_{\widehat{A}}\lambda_C}\gamma^2\rho_{\max}^2 10K_5$$

$$+ \frac{\alpha^2}{\beta M} \cdot \frac{16}{\lambda_{\widehat{A}}}\Big(1 + \frac{2}{\lambda_C}\Big) \cdot \frac{96}{\lambda_{\widehat{A}}\lambda_C}\gamma^2\rho_{\max}^2 10\Big(\rho_{\max}\frac{1 + \gamma}{\min|\lambda(C)|}\Big)^2 K_1 + \frac{\alpha}{M} \cdot \frac{80K_1}{\lambda_{\widehat{A}}}\Big\}$$

$$+ \frac{1}{M}\frac{48K_3 + 48K_4}{\lambda_C} + \frac{\beta}{M}\frac{240K_5}{\lambda_C} + \frac{\alpha^2}{\beta^2}\frac{1}{M}\frac{240}{\lambda_C}\Big(\rho_{\max}\frac{1 + \gamma}{\min|\lambda(C)|}\Big)^2 K_1\Big(1 + \frac{2}{\lambda_C}\Big)$$

$$+ \frac{1}{1 - F}\Big\{\frac{1}{M}(1 + \gamma)^2\rho_{\max}^2\frac{2880}{\lambda_{\widehat{A}}\lambda_C} \cdot \Big[\Big(1 + \frac{\gamma\rho_{\max}}{\min|\lambda(C)|}\Big)^2\Big(1 + \frac{2}{\lambda_C}\Big) \cdot \Big(\rho_{\max}\frac{1 + \gamma}{\min|\lambda(C)|}\Big)^2\frac{\alpha^2}{\beta^2} + \Big(1 + \frac{1}{\min|\lambda(C)|}\Big)^2\beta\Big] \cdot$$

744 To further simplify, note that the first three terms in the right hand side of the above inequality
745 are in the order of $D^m$ ($D > E, D > F$). For the fourth term, it is easy to check that under
746 the relation $\beta = \mathcal{O}(\alpha^{2/3})$, it is in the order of $\mathcal{O}(\beta^3) + \mathcal{O}(\frac{\beta}{M})$. The other terms are dominated
747 by $\frac{1}{M}$. Therefore, the asymptotic error is in the order of $\mathcal{O}(\beta^3) + \mathcal{O}(\frac{1}{M})$. To further elaborate
748 this, note that the fourth term is a product of three curly brackets. The first bracket is in the order
749 of $\mathcal{O}(\beta) + \mathcal{O}(\beta) \times \big(\mathcal{O}(\frac{1}{\alpha M}) + \mathcal{O}(\alpha)\big) = \mathcal{O}(\beta) + \mathcal{O}(\frac{\beta}{\alpha}\frac{1}{M})$, the second one is in the order of
750 $\mathcal{O}(\frac{1}{\beta M}) + \mathcal{O}(\beta)$ and the last one is in the order of $\mathcal{O}(\beta)$. Hence their product is in the order of
751 $\big(\mathcal{O}(\beta) + \mathcal{O}(\frac{\beta}{\alpha}\frac{1}{M})\big) \times \big(\mathcal{O}(\frac{1}{\beta M}) + \mathcal{O}(\beta)\big) \times \mathcal{O}(\beta) = \mathcal{O}(\frac{\beta}{M}) + \mathcal{O}(\frac{\beta}{\alpha}\frac{1}{M^2}) + \mathcal{O}(\beta^3) + \mathcal{O}(\frac{\beta^3}{\alpha}\frac{1}{M}) =$
752 $\mathcal{O}(\frac{\beta}{M}) + \mathcal{O}(\frac{\beta}{\alpha}\frac{1}{M^2}) + \mathcal{O}(\beta^3)$. In summary, we have the following asymptotic result:

$$\mathbb{E}\|\tilde{z}^{(m)}\|^2 \leq \mathcal{O}(D^m) + \mathcal{O}(\beta^3) + \mathcal{O}(\frac{1}{M}).$$

753 By following the same proof logic of Theorem 3.1 and Theorem 4.1, the sample complexity under
754 the optimal setting is $\mathcal{O}(\epsilon^{-1}\log\epsilon^{-1})$.

755 $\qquad\qquad\qquad\qquad\qquad\qquad\qquad\qquad\qquad\qquad\qquad\qquad\qquad\qquad\qquad\qquad\qquad\qquad$ □

# H Key Lemmas for Proving Theorem 4.1

757 **Lemma H.1** (Preliminary Bound for $\sum_{t=0}^{M-1}\|\theta_t^{(m)} - \theta^*\|^2$)**.** *Under the same assumptions as those*
758 *of Theorem 3.1, choose the learning rate $\alpha$ such that*

$$\alpha \leq \min\Big\{\frac{\lambda_{\widehat{A}}}{30}\Big/\Big[(1 + \gamma)^2\rho_{\max}^2\Big(1 + \frac{\gamma\rho_{\max}}{\min|\lambda(C)|}\Big)^2\Big], \frac{3}{5}\frac{1}{\lambda_{\widehat{A}}}\Big\}. \tag{52}$$

759 *Then, the following preliminary bound holds,*

$$\frac{\lambda_{\widehat{A}}}{12}\alpha\sum_{t=0}^{M-1}\mathbb{E}_{m,0}\|\theta_t^{(m)} - \theta^*\|^2$$

$$\leq \Big[1 + \alpha^2 M \cdot 5(1+\gamma)^2 \rho_{\max}^2 \Big(1 + \frac{\gamma \rho_{\max}}{\min|\lambda(C)|}\Big)^2\Big] \mathbb{E}_{m,0}\|\tilde{\theta}^{(m-1)} - \theta^*\|^2 + \alpha \cdot 2K_2 + \alpha^2 \cdot 5K_1$$

$$+ \alpha \cdot \frac{6}{\lambda_{\widehat{A}}} \gamma^2 \rho_{\max}^2 \sum_{t=0}^{M-1} \mathbb{E}_{m,0}\|z_t^{(m)}\|^2 + \alpha^2 M \cdot 5\gamma^2 \rho_{\max}^2 \mathbb{E}_{m,0}\|\tilde{z}^{(m-1)}\|^2,$$

where $K_1$ is specified in eq. (63) of Lemma I.1, and $K_2$ is specified in eq. (67) of Lemma I.2 .

*Proof.* Based on the update rule of VRTDC for Markovian samples, we have that

$$\theta_{t+1}^{(m)} = \Pi_{R_\theta}\Big[\theta_t^{(m)} + \alpha[G_t^{(m)}(\theta_t^{(m)}, z_t^{(m)}) - G_t^{(m)}(\tilde{\theta}^{(m-1)}, \tilde{z}^{(m-1)}) + G^{(m)}(\tilde{\theta}^{(m-1)}, \tilde{z}^{(m-1)})]\Big].$$

The above update rule further implies that

$$\|\theta_{t+1}^{(m)} - \theta^*\|^2 \overset{(i)}{\leq} \|\theta_t^{(m)} - \theta^* + \alpha[G_t^{(m)}(\theta_t^{(m)}, z_t^{(m)}) - G_t^{(m)}(\tilde{\theta}^{(m-1)}, \tilde{z}^{(m-1)}) + G^{(m)}(\tilde{\theta}^{(m-1)}, \tilde{z}^{(m-1)})]\|^2$$

$$= \|\theta_t^{(m)} - \theta^*\|^2 + \alpha^2 \|G_t^{(m)}(\theta_t^{(m)}, z_t^{(m)}) - G_t^{(m)}(\tilde{\theta}^{(m-1)}, \tilde{z}^{(m-1)}) + G^{(m)}(\tilde{\theta}^{(m-1)}, \tilde{z}^{(m-1)})\|^2$$

$$+ 2\alpha\langle\theta_t^{(m)} - \theta^*, G_t^{(m)}(\theta_t^{(m)}, z_t^{(m)}) - G_t^{(m)}(\tilde{\theta}^{(m-1)}, \tilde{z}^{(m-1)}) + G^{(m)}(\tilde{\theta}^{(m-1)}, \tilde{z}^{(m-1)})\rangle, \tag{53}$$

where (i) uses the assumption that $R_\theta \geq \|\theta^*\|$ (i.e., $\theta^*$ is in the ball with radius $R_\theta$) and the fact that $\Pi_{R_\theta}$ is 1-Lipschitz. Then we take $\mathbb{E}_{m,0}$ on both sides. An upper bound for the second term of eq. (53) is given in Lemma I.6. Next, we consider the third term of eq. (53) and obtain that

$$\mathbb{E}_{m,0}\langle\theta_t^{(m)} - \theta^*, G_t^{(m)}(\theta_t^{(m)}, z_t^{(m)}) - G_t^{(m)}(\tilde{\theta}^{(m-1)}, \tilde{z}^{(m-1)}) + G^{(m)}(\tilde{\theta}^{(m-1)}, \tilde{z}^{(m-1)})\rangle$$

$$= \mathbb{E}_{m,0}\langle\theta_t^{(m)} - \theta^*, G^{(m)}(\theta_t^{(m)}, z_t^{(m)})\rangle$$

$$= \mathbb{E}_{m,0}\langle\theta_t^{(m)} - \theta^*, \widehat{A}^{(m)}\theta_t^{(m)} + \widehat{b}^{(m)} + \widehat{B}^{(m)}z_t^{(m)}\rangle$$

$$= \mathbb{E}_{m,0}\langle\theta_t^{(m)} - \theta^*, \widehat{A}^{(m)}\theta_t^{(m)} - \widehat{A}\theta_t^{(m)} + \widehat{A}\theta_t^{(m)} + \widehat{b}^{(m)} - \widehat{b} + \widehat{b}\rangle + \mathbb{E}_{m,0}\langle\theta_t^{(m)} - \theta^*, \widehat{B}^{(m)}z_t^{(m)}\rangle$$

$$= \mathbb{E}_{m,0}\langle\theta_t^{(m)} - \theta^*, (\widehat{A}^{(m)} - \widehat{A})\theta_t^{(m)} + (\widehat{b}^{(m)} - \widehat{b})\rangle + \mathbb{E}_{m,0}\langle\theta_t^{(m)} - \theta^*, \widehat{A}\theta_t^{(m)} - \widehat{A}\theta^* + \widehat{A}\theta^* + \widehat{b}\rangle$$

$$+ \mathbb{E}_{m,0}\langle\theta_t^{(m)} - \theta^*, \widehat{B}^{(m)}z_t^{(m)}\rangle$$

$$= \mathbb{E}_{m,0}\langle\theta_t^{(m)} - \theta^*, (\widehat{A}^{(m)} - \widehat{A})\theta_t^{(m)} + (\widehat{b}^{(m)} - \widehat{b})\rangle + \mathbb{E}_{m,0}\langle\theta_t^{(m)} - \theta^*, \widehat{A}(\theta_t^{(m)} - \theta^*)\rangle$$

$$+ \mathbb{E}_{m,0}\langle\theta_t^{(m)} - \theta^*, \widehat{B}^{(m)}z_t^{(m)}\rangle \tag{54}$$

The first term of eq. (54) is bounded by Lemma I.2. The second term of eq. (54) can be bounded by using the property of negative definite matrix: $\lambda_{\max}(\widehat{A})\|\theta - \theta^*\|^2 \geq (\theta - \theta^*)^T\widehat{A}(\theta - \theta^*) \geq \lambda_{\min}(\widehat{A})\|\theta - \theta^*\|^2$. Recall that $-\lambda_{\widehat{A}} := \lambda_{\max}(\widehat{A} + \widehat{A}^T)$, and we obtain that

$$\mathbb{E}_{m,0}\langle\theta_t^{(m)} - \theta^*, \widehat{A}(\theta_t^{(m)} - \theta^*)\rangle \leq -\frac{\lambda_{\widehat{A}}}{2}\mathbb{E}_{m,0}\|\theta_t^{(m)} - \theta^*\|^2.$$

The third term of eq. (54) is bounded using the polarization identity,

$$\mathbb{E}_{m,0}\langle\theta_t^{(m)} - \theta^*, \widehat{B}^{(m)}z_t^{(m)}\rangle \leq \frac{1}{2}\cdot\frac{\lambda_{\widehat{A}}}{3}\mathbb{E}_{m,0}\|\theta_t^{(m)} - \theta^*\|^2 + \frac{1}{2}\cdot\frac{3}{\lambda_{\widehat{A}}}\gamma^2\rho_{\max}^2\mathbb{E}_{m,0}\|z_t^{(m)}\|^2.$$

Substituting the above bounds into the third term of eq. (53), we obtain that

$$\mathbb{E}_{m,0}\langle\theta_t^{(m)} - \theta^*, G_t^{(m)}(\theta_t^{(m)}, z_t^{(m)}) - G_t^{(m)}(\tilde{\theta}^{(m-1)}, \tilde{z}^{(m-1)}) + G^{(m)}(\tilde{\theta}^{(m-1)}, \tilde{z}^{(m-1)})\rangle$$

$$\leq -\frac{\lambda_{\widehat{A}}}{12}\mathbb{E}_{m,0}\|\theta_t^{(m)} - \theta^*\|^2 + \frac{K_2}{M} + \frac{1}{2}\cdot\frac{3}{\lambda_{\widehat{A}}}\gamma^2\rho_{\max}^2\mathbb{E}_{m,0}\|z_t^{(m)}\|^2. \tag{55}$$

Then, substituting eq. (55) into eq. (53) and re-arranging, we obtain that

$$\mathbb{E}_{m,0}\|\theta_{t+1}^{(m)} - \theta^*\|^2$$

$$\leq \mathbb{E}_{m,0}\|\theta_t^{(m)} - \theta^*\|^2 + \alpha\Big[-\frac{\lambda_{\widehat{A}}}{6}\mathbb{E}_{m,0}\|\theta_t^{(m)} - \theta^*\|^2 + \frac{2K_2}{M} + \frac{3}{\lambda_{\widehat{A}}}\gamma^2\rho_{\max}^2\mathbb{E}_{m,0}\|z_t^{(m)}\|^2\Big]$$

$$+ \alpha^2 \Big[ 5(1+\gamma)^2 \rho_{\max}^2 \Big(1 + \frac{\gamma \rho_{\max}}{\min|\lambda(C)|}\Big)^2 \Big( \mathbb{E}_{m,0}\|\theta_t^{(m)} - \theta^*\|^2 + \mathbb{E}_{m,0}\|\tilde{\theta}^{(m-1)} - \theta^*\|^2 \Big) \Big]$$

$$+ \alpha^2 \Big[ 5\gamma^2 \rho_{\max}^2 \Big( \mathbb{E}_{m,0}\|z_t^{(m)}\|^2 + \mathbb{E}_{m,0}\|\tilde{z}^{(m-1)}\|^2 \Big) + \frac{5K_1}{M} \Big]$$

$$= \mathbb{E}_{m,0}\|\theta_t^{(m)} - \theta^*\|^2 - \Big( \frac{\lambda_{\widehat{A}}}{6}\alpha - \alpha^2 \cdot 5(1+\gamma)^2 \rho_{\max}^2 \Big(1 + \frac{\gamma \rho_{\max}}{\min|\lambda(C)|}\Big)^2 \Big) \mathbb{E}_{m,0}\|\theta_t^{(m)} - \theta^*\|^2$$

$$+ \alpha^2 \cdot 5(1+\gamma)^2 \rho_{\max}^2 \Big(1 + \frac{\gamma \rho_{\max}}{\min|\lambda(C)|}\Big)^2 \mathbb{E}_{m,0}\|\tilde{\theta}^{(m-1)} - \theta^*\|^2 + \frac{\alpha}{M}\cdot 2K_2 + \frac{\alpha^2}{M}\cdot 5K_1$$

$$+ \Big( \alpha \cdot \frac{3}{\lambda_{\widehat{A}}}\gamma^2 \rho_{\max}^2 + \alpha^2 \cdot 5\gamma^2 \rho_{\max}^2 \Big) \mathbb{E}_{m,0}\|z_t^{(m)}\|^2 + \alpha^2 \cdot 5\gamma^2 \rho_{\max}^2 \mathbb{E}_{m,0}\|\tilde{z}^{(m-1)}\|^2.$$

769  Summing the above inequality over $t = 0, \ldots, M-1$, we obtain the following desired bound

$$\Big( \frac{\lambda_{\widehat{A}}}{6}\alpha - \alpha^2 \cdot 5(1+\gamma)^2 \rho_{\max}^2 \Big(1 + \frac{\gamma \rho_{\max}}{\min|\lambda(C)|}\Big)^2 \Big) \sum_{t=0}^{M-1} \mathbb{E}_{m,0}\|\theta_t^{(m)} - \theta^*\|^2$$

$$\leq \Big[ 1 + \alpha^2 M \cdot 5(1+\gamma)^2 \rho_{\max}^2 \Big(1 + \frac{\gamma \rho_{\max}}{\min|\lambda(C)|}\Big)^2 \Big] \mathbb{E}_{m,0}\|\tilde{\theta}^{(m-1)} - \theta^*\|^2 + \alpha \cdot 2K_2 + \alpha^2 \cdot 5K_1$$

$$+ \Big( \alpha \cdot \frac{3}{\lambda_{\widehat{A}}}\gamma^2 \rho_{\max}^2 + \alpha^2 \cdot 5\gamma^2 \rho_{\max}^2 \Big) \sum_{t=0}^{M-1} \mathbb{E}_{m,0}\|z_t^{(m)}\|^2 + \alpha^2 M \cdot 5\gamma^2 \rho_{\max}^2 \mathbb{E}_{m,0}\|\tilde{z}^{(m-1)}\|^2.$$

$$(56)$$

770  Lastly, we simplify the above bound by choosing sufficiently small $\alpha$ such that $\frac{\lambda_{\widehat{A}}}{6}\alpha - \alpha^2 \cdot 5(1+$
771  $\gamma)^2 \rho_{\max}^2 \Big(1 + \frac{\gamma \rho_{\max}}{\min|\lambda(C)|}\Big)^2 \geq \frac{\lambda_{\widehat{A}}}{12}\alpha$, and $\alpha^2 \cdot 5\gamma^2 \rho_{\max}^2 \leq \alpha \cdot \frac{3}{\lambda_{\widehat{A}}}\gamma^2 \rho_{\max}^2$. Note that these requirements
772  can be implied by

$$\alpha \leq \min \Big\{ \frac{\lambda_{\widehat{A}}}{30} \Big/ \Big[ (1+\gamma)^2 \rho_{\max}^2 \Big(1 + \frac{\gamma \rho_{\max}}{\min|\lambda(C)|}\Big)^2 \Big], \frac{3}{5}\frac{1}{\lambda_{\widehat{A}}} \Big\}.$$

773  Applying these simplifications, eq. (56) becomes

$$\frac{\lambda_{\widehat{A}}}{12}\alpha \sum_{t=0}^{M-1} \mathbb{E}_{m,0}\|\theta_t^{(m)} - \theta^*\|^2$$

$$\leq \Big[ 1 + \alpha^2 M \cdot 5(1+\gamma)^2 \rho_{\max}^2 \Big(1 + \frac{\gamma \rho_{\max}}{\min|\lambda(C)|}\Big)^2 \Big] \mathbb{E}_{m,0}\|\tilde{\theta}^{(m-1)} - \theta^*\|^2 + \alpha \cdot 2K_2 + \alpha^2 \cdot 5K_1$$

$$+ \alpha \cdot \frac{6}{\lambda_{\widehat{A}}}\gamma^2 \rho_{\max}^2 \sum_{t=0}^{M-1} \mathbb{E}_{m,0}\|z_t^{(m)}\|^2 + \alpha^2 M \cdot 5\gamma^2 \rho_{\max}^2 \mathbb{E}_{m,0}\|\tilde{z}^{(m-1)}\|^2.$$

774  $\qquad\qquad\qquad\qquad\qquad\qquad\qquad\qquad\qquad\qquad\qquad\qquad\qquad\qquad\qquad\qquad\qquad$ $\square$

775  **Lemma H.2** (Convergence of $\mathbb{E}\|\tilde{z}^{(m)}\|^2$). *Under the same assumptions as those of Theorem 4.1,*
776  *choose the learning rate $\beta$ and the batch size $M$ such that $\beta < 1$ and $M\beta > \frac{12}{\lambda_C}$. Then, the following*
777  *preliminary bound holds.*

$$\mathbb{E}\|\tilde{z}^{(m)}\|^2 \leq \Big( \frac{1}{M\beta} \cdot \frac{12}{\lambda_C} \Big)^m \mathbb{E}\|\tilde{z}^{(0)}\|^2$$

$$+ 2 \cdot \Big[ \beta \cdot \frac{24}{\lambda_C} H_{VR}^2 + \frac{\alpha^2}{\beta^2} \cdot \Big(1 + \frac{2}{\lambda_C}\Big) \cdot \frac{24}{\lambda_C} \Big( \rho_{\max} \frac{1+\gamma}{\min|\lambda(C)|}\Big)^2 G_{VR}^2 + \frac{1}{M}\frac{24}{\lambda_C}(K_3 + K_4) \Big],$$

778  *where $K_3$ is specified in eq. (71) of Lemma I.3, and $K_4$ is specified in eq. (75) of Lemma I.4.*

779  *Proof.* Similar to Lemma D.2, we can obtain the one-step update rule of $z_t^{(m)}$ based on the one-step
780  update rule of $w_t^{(m)}$. Combining this update rule with the assumption that $R_w \geq 2\|C^{-1}\|\|A\|R_\theta$,
781  we obtain that

$$\|z_{t+1}^{(m)}\|^2 \leq \|z_t^{(m)} + \beta\Big[ H_t^{(m)}(\theta_t^{(m)}, z_t^{(m)}) - H_t^{(m)}(\tilde{\theta}^{(m-1)}, \tilde{z}^{(m-1)}) + H^{(m)}(\tilde{\theta}^{(m-1)}, \tilde{z}^{(m-1)}) \Big]$$

$$+ C^{-1}A(\theta_{t+1}^{(m)} - \theta_t^{(m)})\|^2$$
$$= \|z_t^{(m)}\|^2 + 2\beta^2\|H_t^{(m)}(\theta_t^{(m)}, z_t^{(m)}) - H_t^{(m)}(\tilde{\theta}^{(m-1)}, \tilde{z}^{(m-1)}) + H^{(m)}(\tilde{\theta}^{(m-1)}, \tilde{z}^{(m-1)})\|^2$$
$$+ 2\|C^{-1}A(\theta_{t+1}^{(m)} - \theta_t^{(m)})\|^2$$
$$+ 2\beta\langle z_t^{(m)}, H_t^{(m)}(\theta_t^{(m)}, z_t^{(m)}) - H_t^{(m)}(\tilde{\theta}^{(m-1)}, \tilde{z}^{(m-1)}) + H^{(m)}(\tilde{\theta}^{(m-1)}, \tilde{z}^{(m-1)})\rangle$$
$$+ 2\langle z_t^{(m)}, C^{-1}A(\theta_{t+1}^{(m)} - \theta_t^{(m)}).\rangle \tag{57}$$

782 For the last term of eq. (57), we bound it as

$$2\langle z_t^{(m)}, C^{-1}A(\theta_{t+1}^{(m)} - \theta_t^{(m)})\rangle \le \frac{\lambda_C}{2}\beta\|z_t^{(m)}\|^2 + \frac{2}{\lambda_C}\frac{1}{\beta}\|C^{-1}A(\theta_{t+1} - \theta_t)\|^2. \tag{58}$$

783 Then, we apply Lemma J.4 to bound the last term of eq. (58). Also, we apply Lemma J.5 to bound
784 the second term of eq. (57). Then, we obtain that

$$\|z_{t+1}^{(m)}\|^2 \le \|z_t^{(m)}\|^2 + 2\beta^2 H_{\text{VR}}^2 + \left(\alpha^2 + \frac{2\alpha^2}{\lambda_C}\frac{1}{\beta}\right)\cdot 2\left(\rho_{\max}\frac{1+\gamma}{\min|\lambda(C)|}\right)^2 G_{\text{VR}}^2 + \frac{\lambda_C}{2}\beta\|z_t^{(m)}\|^2$$
$$+ 2\beta\langle z_t^{(m)}, H_t^{(m)}(\theta_t^{(m)}, z_t^{(m)}) - H_t^{(m)}(\tilde{\theta}^{(m-1)}, \tilde{z}^{(m-1)}) + H^{(m)}(\tilde{\theta}^{(m-1)}, \tilde{z}^{(m-1)})\rangle \tag{59}$$

785 Next, we further bound the last term of eq. (59).

$$\mathbb{E}_{m,0}\langle z_t^{(m)}, H_t^{(m)}(\theta_t^{(m)}, z_t^{(m)}) - H_t^{(m)}(\tilde{\theta}^{(m-1)}, \tilde{z}^{(m-1)}) + H^{(m)}(\tilde{\theta}^{(m-1)}, \tilde{z}^{(m-1)})\rangle$$
$$= \mathbb{E}_{m,0}\langle z_t^{(m)}, H^{(m)}(\theta_t^{(m)}, z_t^{(m)})\rangle$$
$$= \mathbb{E}_{m,0}\langle z_t^{(m)}, \bar{A}^{(m)}\theta_t^{(m)} + \bar{b}^{(m)} + C^{(m)}z_t^{(m)}\rangle$$
$$= \mathbb{E}_{m,0}\langle z_t^{(m)}, \bar{A}^{(m)}\theta_t^{(m)} + \bar{b}^{(m)}\rangle + \mathbb{E}_{m,0}\langle z_t^{(m)}, (C^{(m)} - C)z_t^{(m)}\rangle + \mathbb{E}_{m,0}\langle z_t^{(m)}, Cz_t^{(m)}\rangle. \tag{60}$$

786 Then, we apply Lemma I.3 to bound the first term of eq. (60), apply Lemma I.4 to bound the second
787 term of eq. (60) and apply the negative definiteness of $C$ to bound the last term of eq. (60). We obtain
788 that

$$\mathbb{E}_{m,0}\langle z_t^{(m)}, H_t^{(m)}(\theta_t^{(m)}, z_t^{(m)}) - H_t^{(m)}(\tilde{\theta}^{(m-1)}, \tilde{z}^{(m-1)}) + H^{(m)}(\tilde{\theta}^{(m-1)}, \tilde{z}^{(m-1)})\rangle$$
$$\le \left(\frac{\lambda_C}{8}\mathbb{E}_{m,0}\|z_t^{(m)}\|^2 + \frac{K_3}{M}\right) + \left(\frac{\lambda_C}{12}\mathbb{E}_{m,0}\|z_t^{(m)}\|^2 + \frac{K_4}{M}\right) - \frac{\lambda_C}{2}\mathbb{E}_{m,0}\|z_t^{(m)}\|^2$$
$$= -\frac{7\lambda_C}{24}\mathbb{E}_{m,0}\|z_t^{(m)}\|^2 + \frac{K_3 + K_4}{M}. \tag{61}$$

789 Substituting the above inequality into eq. (59) yields that

$$\mathbb{E}_{m,0}\|z_{t+1}^{(m)}\|^2 \le \mathbb{E}_{m,0}\|z_t^{(m)}\|^2 + 2\beta^2 H_{\text{VR}}^2 + \left(\alpha^2 + \frac{2\alpha^2}{\lambda_C}\frac{1}{\beta}\right)\cdot 2\left(\rho_{\max}\frac{1+\gamma}{\min|\lambda(C)|}\right)^2 G_{\text{VR}}^2$$
$$- \frac{\lambda_C}{12}\beta\mathbb{E}_{m,0}\|z_t^{(m)}\|^2 + \frac{2K_3 + 2K_4}{M}\beta.$$

790 Telescoping the above inequality over one batch, we further obtain that

$$\mathbb{E}_{m,0}\|z_M^{(m)}\|^2 \le \mathbb{E}_{m,0}\|z_0^{(m)}\|^2 + 2\beta^2 M H_{\text{VR}}^2 + \left(\alpha^2 + \frac{2\alpha^2}{\lambda_C}\frac{1}{\beta}\right)M\cdot 2\left(\rho_{\max}\frac{1+\gamma}{\min|\lambda(C)|}\right)^2 G_{\text{VR}}^2$$
$$- \frac{\lambda_C}{12}\beta\sum_{t=0}^{M-1}\mathbb{E}_{m,0}\|z_t^{(m)}\|^2 + (2K_3 + 2K_4)\beta.$$

791 Next, we move the term $\sum_{t=0}^{M-1}\mathbb{E}_{m,0}\|z_t^{(m)}\|^2$ in the above inequality to the left-hand side and apply
792 Jensen's inequality, we obtain that

$$\frac{\lambda_C}{12}\beta M\mathbb{E}_{m,0}\|\tilde{z}^{(m)}\|^2 \le \|\tilde{z}^{(m-1)}\|^2 + 2\beta^2 M H_{\text{VR}}^2$$
$$+ \left(\alpha^2 + \frac{2\alpha^2}{\lambda_C}\frac{1}{\beta}\right)M\cdot 2\left(\rho_{\max}\frac{1+\gamma}{\min|\lambda(C)|}\right)^2 G_{\text{VR}}^2 + (2K_3 + 2K_4)\beta.$$

Lastly, we divide $\frac{\lambda_C}{12}\beta M$ on both sides of the above inequality and obtain that

$$\mathbb{E}\|\tilde{z}^{(m)}\|^2 \le \frac{1}{M\beta} \cdot \frac{12}{\lambda_C}\mathbb{E}\|\tilde{z}^{(m-1)}\|^2 + \beta \cdot \frac{24}{\lambda_C}H_{\mathrm{VR}}^2 + \Big(\frac{\alpha^2}{\beta} + \frac{2}{\lambda_C}\frac{\alpha^2}{\beta^2}\Big) \cdot \frac{24}{\lambda_C}\Big(\rho_{\max}\frac{1+\gamma}{\min|\lambda(C)|}\Big)^2 G_{\mathrm{VR}}^2$$
$$+ \frac{1}{M}\frac{24}{\lambda_C}(K_3 + K_4),$$

which, after telescoping, leads to

$$\mathbb{E}\|\tilde{z}^{(m)}\|^2 \le \Big(\frac{1}{M\beta} \cdot \frac{12}{\lambda_C}\Big)^m \mathbb{E}\|\tilde{z}^{(0)}\|^2$$
$$+ \frac{1}{1 - \frac{1}{M\beta}\cdot\frac{12}{\lambda_C}}\Big[\beta \cdot \frac{24}{\lambda_C}H_{\mathrm{VR}}^2 + \Big(\frac{\alpha^2}{\beta} + \frac{2}{\lambda_C}\frac{\alpha^2}{\beta^2}\Big)\cdot\frac{24}{\lambda_C}\Big(\rho_{\max}\frac{1+\gamma}{\min|\lambda(C)|}\Big)^2 G_{\mathrm{VR}}^2 + \frac{1}{M}\frac{24}{\lambda_C}(K_3 + K_4)\Big]$$

To further simplify the above inequality, we assume $\beta < 1$ and $M\beta > \frac{24}{\lambda_C}$. Then, we have

$$\mathbb{E}\|\tilde{z}^{(m)}\|^2 \le \Big(\frac{1}{M\beta} \cdot \frac{12}{\lambda_C}\Big)^m \mathbb{E}\|\tilde{z}^{(0)}\|^2$$
$$+ 2 \cdot \Big[\beta \cdot \frac{24}{\lambda_C}H_{\mathrm{VR}}^2 + \frac{\alpha^2}{\beta^2}\cdot\Big(1 + \frac{2}{\lambda_C}\Big)\cdot\frac{24}{\lambda_C}\Big(\rho_{\max}\frac{1+\gamma}{\min|\lambda(C)|}\Big)^2 G_{\mathrm{VR}}^2 + \frac{1}{M}\frac{24}{\lambda_C}(K_3 + K_4)\Big].$$

$\square$

**Lemma H.3** (Preliminary Bound for $\sum_{t=0}^{M-1}\|z_t^{(m)}\|^2$). *Under the same assumptions as those of Theorem 4.1, choose the learning rate $\alpha$ and $\beta$ such that*

$$\frac{\lambda_C}{12}\beta - 10\beta^2 - 10\gamma^2\rho_{\max}^2\Big(\alpha^2 + \frac{2\alpha^2}{\lambda_C}\frac{1}{\beta}\Big)\cdot\Big(\rho_{\max}\frac{1+\gamma}{\min|\lambda(C)|}\Big)^2 \ge \frac{\lambda_C}{16}\beta.$$

*Then, the following preliminary bound holds.*

$$\frac{\lambda_C}{16}\beta\sum_{t=0}^{M-1}\mathbb{E}_{m,0}\|z_t^{(m)}\|^2$$
$$\le\Big[1 + \Big[10\beta^2 + 10\gamma^2\rho_{\max}^2\Big(\alpha^2 + \frac{2\alpha^2}{\lambda_C}\frac{1}{\beta}\Big)\cdot\Big(\rho_{\max}\frac{1+\gamma}{\min|\lambda(C)|}\Big)^2\Big]M\Big]\mathbb{E}_{m,0}\|\tilde{z}^{(m-1)}\|^2$$
$$+ 10(1+\gamma)^2\rho_{\max}^2\cdot\Big[\Big(1 + \frac{\gamma\rho_{\max}}{\min|\lambda(C)|}\Big)^2\Big(\alpha^2 + \frac{2\alpha^2}{\lambda_C}\frac{1}{\beta}\Big)\cdot\Big(\rho_{\max}\frac{1+\gamma}{\min|\lambda(C)|}\Big)^2$$
$$+ \Big(1 + \frac{1}{\min|\lambda(C)|}\Big)^2\beta^2\Big]\Big(\sum_{t=0}^{M-1}\mathbb{E}_{m,0}\|\theta_t^{(m)} - \theta^*\|^2 + M\mathbb{E}_{m,0}\|\tilde{\theta}^{(m-1)} - \theta^*\|^2\Big)$$
$$+ (2K_3 + 2K_4)\beta + 10K_5\beta^2 + 10\Big(\rho_{\max}\frac{1+\gamma}{\min|\lambda(C)|}\Big)^2 K_1\Big(\alpha^2 + \frac{2\alpha^2}{\lambda_C}\frac{1}{\beta}\Big).$$

*Proof.* Similar to Lemma H.2, we firstly consider the one-step update of $z_t^{(m)}$:

$$\|z_{t+1}^{(m)}\|^2 \le \|z_t^{(m)}\|^2 + 2\beta^2\|H_t^{(m)}(\theta_t^{(m)}, z_t^{(m)}) - H_t^{(m)}(\tilde{\theta}^{(m-1)}, \tilde{z}^{(m-1)}) + H^{(m)}(\tilde{\theta}^{(m-1)}, \tilde{z}^{(m-1)})\|^2$$
$$+ 2\|C^{-1}A(\theta_{t+1}^{(m)} - \theta_t^{(m)})\|^2 + 2\langle z_t^{(m)}, C^{-1}A(\theta_{t+1}^{(m)} - \theta_t^{(m)})\rangle$$
$$+ 2\beta\langle z_t^{(m)}, H_t^{(m)}(\theta_t^{(m)}, z_t^{(m)}) - H_t^{(m)}(\tilde{\theta}^{(m-1)}, \tilde{z}^{(m-1)}) + H^{(m)}(\tilde{\theta}^{(m-1)}, \tilde{z}^{(m-1)})\rangle$$
$$\le \|z_t^{(m)}\|^2 + 2\beta^2\|H_t^{(m)}(\theta_t^{(m)}, z_t^{(m)}) - H_t^{(m)}(\tilde{\theta}^{(m-1)}, \tilde{z}^{(m-1)}) + H^{(m)}(\tilde{\theta}^{(m-1)}, \tilde{z}^{(m-1)})\|^2$$
$$+ 2\|C^{-1}A(\theta_{t+1}^{(m)} - \theta_t^{(m)})\|^2 + \frac{\lambda_C}{2}\beta\|z_t^{(m)}\|^2 + \frac{2}{\lambda_C}\frac{1}{\beta}\|C^{-1}A(\theta_{t+1} - \theta_t)\|^2.$$
$$+ 2\beta\langle z_t^{(m)}, H_t^{(m)}(\theta_t^{(m)}, z_t^{(m)}) - H_t^{(m)}(\tilde{\theta}^{(m-1)}, \tilde{z}^{(m-1)}) + H^{(m)}(\tilde{\theta}^{(m-1)}, \tilde{z}^{(m-1)})\rangle$$
$$\le \|z_t^{(m)}\|^2 + \frac{\lambda_C}{2}\beta\|z_t^{(m)}\|^2 + 2\beta^2\|H_t^{(m)}(\theta_t^{(m)}, z_t^{(m)}) - H_t^{(m)}(\tilde{\theta}^{(m-1)}, \tilde{z}^{(m-1)}) + H^{(m)}(\tilde{\theta}^{(m-1)}, \tilde{z}^{(m-1)})\|^2$$

$$+ \left(\alpha^2 + \frac{2\alpha^2}{\lambda_C}\frac{1}{\beta}\right) \cdot 2\left(\rho_{\max}\frac{1+\gamma}{\min|\lambda(C)|}\right)^2 \|G_t^{(m)}(\theta_t^{(m)}, z_t^{(m)}) - G_t^{(m)}(\tilde{\theta}^{(m-1)}, \tilde{z}^{(m-1)}) + G^{(m)}(\tilde{\theta}^{(m-1)}, \tilde{z}^{(m-1)})\|$$
$$+ 2\beta\langle z_t^{(m)}, H_t^{(m)}(\theta_t^{(m)}, z_t^{(m)}) - H_t^{(m)}(\tilde{\theta}^{(m-1)}, \tilde{z}^{(m-1)}) + H^{(m)}(\tilde{\theta}^{(m-1)}, \tilde{z}^{(m-1)})\rangle$$
$$(62)$$

801  where the second inequality applies the polarization identity to $\langle z_t^{(m)}, C^{-1}A(\theta_{t+1}^{(m)} - \theta_t^{(m)})\rangle$ . For
802  the last term of eq. (62), it is bounded by eq. (61) as

$$\mathbb{E}_{m,0}\langle z_t^{(m)}, H_t^{(m)}(\theta_t^{(m)}, z_t^{(m)}) - H_t^{(m)}(\tilde{\theta}^{(m-1)}, \tilde{z}^{(m-1)}) + H^{(m)}(\tilde{\theta}^{(m-1)}, \tilde{z}^{(m-1)})\rangle$$
$$\leq -\frac{7\lambda_C}{24}\mathbb{E}_{m,0}\|z_t^{(m)}\|^2 + \frac{K_3 + K_4}{M}.$$

803  To further bound eq. (62), we apply Lemma I.6 to bound its fourth term and apply Lemma I.7 to
804  bound its third term. Then, we obtain that

$$\mathbb{E}_{m,0}\|z_{t+1}^{(m)}\|^2$$
$$\leq \mathbb{E}_{m,0}\|z_t^{(m)}\|^2 - \frac{\lambda_C}{12}\beta\mathbb{E}_{m,0}\|z_t^{(m)}\|^2 + \frac{2K_3 + 2K_4}{M}\beta$$
$$+ 2\beta^2\Big[5\Big(\mathbb{E}_{m,0}\|z_t^{(m)}\|^2 + \mathbb{E}_{m,0}\|\tilde{z}^{(m-1)}\|^2\Big) + \frac{5K_5}{M}$$
$$+ 5(1+\gamma)^2\rho_{\max}^2\Big(1 + \frac{1}{\min|\lambda(C)|}\Big)^2\Big(\mathbb{E}_{m,0}\|\theta_t^{(m)} - \theta^*\|^2 + \mathbb{E}_{m,0}\|\tilde{\theta}^{(m-1)} - \theta^*\|^2\Big)\Big]$$
$$+ \Big(\alpha^2 + \frac{2\alpha^2}{\lambda_C}\frac{1}{\beta}\Big) \cdot 2\Big(\rho_{\max}\frac{1+\gamma}{\min|\lambda(C)|}\Big)^2\Big[5\gamma^2\rho_{\max}^2\Big(\mathbb{E}_{m,0}\|z_t^{(m)}\|^2 + \mathbb{E}_{m,0}\|\tilde{z}^{(m-1)}\|^2\Big) + \frac{5K_1}{M}$$
$$+ 5(1+\gamma)^2\rho_{\max}^2\Big(1 + \frac{\gamma\rho_{\max}}{\min|\lambda(C)|}\Big)^2\Big(\mathbb{E}_{m,0}\|\theta_t^{(m)} - \theta^*\|^2 + \mathbb{E}_{m,0}\|\tilde{\theta}^{(m-1)} - \theta^*\|^2\Big)\Big].$$

805  Next, we re-arrange the above inequality and obtain that

$$\mathbb{E}_{m,0}\|z_{t+1}^{(m)}\|^2$$
$$\leq \mathbb{E}_{m,0}\|z_t^{(m)}\|^2 - \Big[\frac{\lambda_C}{12}\beta - 10\beta^2 - 10\gamma^2\rho_{\max}^2\Big(\alpha^2 + \frac{2\alpha^2}{\lambda_C}\frac{1}{\beta}\Big) \cdot \Big(\rho_{\max}\frac{1+\gamma}{\min|\lambda(C)|}\Big)^2\Big]\mathbb{E}_{m,0}\|z_t^{(m)}\|^2$$
$$+ \Big[10\beta^2 + 10\gamma^2\rho_{\max}^2\Big(\alpha^2 + \frac{2\alpha^2}{\lambda_C}\frac{1}{\beta}\Big) \cdot \Big(\rho_{\max}\frac{1+\gamma}{\min|\lambda(C)|}\Big)^2\Big]\mathbb{E}_{m,0}\|\tilde{z}^{(m-1)}\|^2$$
$$+ 10(1+\gamma)^2\rho_{\max}^2 \cdot \Big[\Big(1 + \frac{\gamma\rho_{\max}}{\min|\lambda(C)|}\Big)^2\Big(\alpha^2 + \frac{2\alpha^2}{\lambda_C}\frac{1}{\beta}\Big) \cdot \Big(\rho_{\max}\frac{1+\gamma}{\min|\lambda(C)|}\Big)^2$$
$$+ \Big(1 + \frac{1}{\min|\lambda(C)|}\Big)^2\beta^2\Big]\Big(\mathbb{E}_{m,0}\|\theta_t^{(m)} - \theta^*\|^2 + \mathbb{E}_{m,0}\|\tilde{\theta}^{(m-1)} - \theta^*\|^2\Big)$$
$$+ \frac{1}{M} \cdot \Big[(2K_3 + 2K_4)\beta + 10K_5\beta^2 + 10\Big(\rho_{\max}\frac{1+\gamma}{\min|\lambda(C)|}\Big)^2 K_1\Big(\alpha^2 + \frac{2\alpha^2}{\lambda_C}\frac{1}{\beta}\Big)\Big].$$

806  Telescoping the above inequality over one batch, we obtain that

$$\Big[\frac{\lambda_C}{12}\beta - 10\beta^2 - 10\gamma^2\rho_{\max}^2\Big(\alpha^2 + \frac{2\alpha^2}{\lambda_C}\frac{1}{\beta}\Big) \cdot \Big(\rho_{\max}\frac{1+\gamma}{\min|\lambda(C)|}\Big)^2\Big]\sum_{t=0}^{M-1}\mathbb{E}_{m,0}\|z_t^{(m)}\|^2$$
$$\leq \Big[1 + \Big[10\beta^2 + 10\gamma^2\rho_{\max}^2\Big(\alpha^2 + \frac{2\alpha^2}{\lambda_C}\frac{1}{\beta}\Big) \cdot \Big(\rho_{\max}\frac{1+\gamma}{\min|\lambda(C)|}\Big)^2\Big]M\Big]\mathbb{E}_{m,0}\|\tilde{z}^{(m-1)}\|^2$$
$$+ 10(1+\gamma)^2\rho_{\max}^2 \cdot \Big[\Big(1 + \frac{\gamma\rho_{\max}}{\min|\lambda(C)|}\Big)^2\Big(\alpha^2 + \frac{2\alpha^2}{\lambda_C}\frac{1}{\beta}\Big) \cdot \Big(\rho_{\max}\frac{1+\gamma}{\min|\lambda(C)|}\Big)^2$$
$$+ \Big(1 + \frac{1}{\min|\lambda(C)|}\Big)^2\beta^2\Big]\Big(\sum_{t=0}^{M-1}\mathbb{E}_{m,0}\|\theta_t^{(m)} - \theta^*\|^2 + M\mathbb{E}_{m,0}\|\tilde{\theta}^{(m-1)} - \theta^*\|^2\Big)$$
$$+ (2K_3 + 2K_4)\beta + 10K_5\beta^2 + 10\Big(\rho_{\max}\frac{1+\gamma}{\min|\lambda(C)|}\Big)^2 K_1\Big(\alpha^2 + \frac{2\alpha^2}{\lambda_C}\frac{1}{\beta}\Big)$$

807 To simplify the above inequality, we assume $\frac{\lambda_C}{12}\beta - 10\beta^2 - 10\gamma^2\rho_{\max}^2(\alpha^2 +$
808 $\frac{2\alpha^2}{\lambda_C}\frac{1}{\beta})(\rho_{\max}\frac{1+\gamma}{\min|\lambda(C)|})^2 \geq \frac{\lambda_C}{16}\beta$ and obtain the desired result. $\qquad\square$

## I   Other Supporting Lemmas for Proving Theorem 4.1

810 **Lemma I.1.** *Under the same assumption as those of Theorem 4.1, the following inequality holds.*

$$\mathbb{E}_{m,0}\|\widehat{A}^{(m)}\theta^* + \widehat{b}^{(m)}\|^2 \leq \frac{K_1}{M},$$

811 *where $K_1$ is defined as*

$$K_1 := \Big[(1+\gamma)R_\theta + r_{\max}\Big]^2 \rho_{\max}^2 \Big(1 + \frac{\gamma\rho_{\max}}{\min|\lambda(C)|}\Big)^2 \cdot \Big(1 + \kappa\frac{2\rho}{1-\rho}\Big). \tag{63}$$

812 *Proof.* Recall that $\widehat{A}^{(m)} = \sum_{t=0}^{M-1}\widehat{A}_t^{(m)}$ and $\widehat{b}^{(m)} = \sum_{t=0}^{M-1}\widehat{b}_t^{(m)}$. We expand the square as follows:

$$\|\widehat{A}^{(m)}\theta^* + \widehat{b}^{(m)}\|^2 \leq \|\sum_{t=0}^{M-1}\widehat{A}_t^{(m)}\theta^* + \sum_{t=0}^{M-1}\widehat{b}_t^{(m)}\|^2$$

$$= \frac{1}{M^2}\Big[\sum_{i=j}\|\widehat{A}_i^{(m)}\theta^* + \widehat{b}_i^{(m)}\|^2 + \sum_{i\neq j}\langle\widehat{A}_i^{(m)}\theta^* + \widehat{b}_i^{(m)}, \widehat{A}_j^{(m)}\theta^* + \widehat{b}_j^{(m)}\rangle\Big]$$

$$\leq \frac{1}{M}\cdot\Big[(1+\gamma)R_\theta + r_{\max}\Big]^2\rho_{\max}^2\Big(1 + \frac{\gamma\rho_{\max}}{\min|\lambda(C)|}\Big)^2 + \frac{1}{M^2}\sum_{i\neq j}\langle\widehat{A}_i^{(m)}\theta^* + \widehat{b}_i^{(m)}, \widehat{A}_j^{(m)}\theta^* + \widehat{b}_j^{(m)}\rangle,$$

813 where in the last step we apply Lemma J.2 to bound the first term. Now we consider the conditional
814 expectation of the second inner product term. Without loss of generality, we assume $i < j$, then

$$\mathbb{E}_{m,0}\langle\widehat{A}_i^{(m)}\theta^* + \widehat{b}_i^{(m)}, \widehat{A}_j^{(m)}\theta^* + \widehat{b}_j^{(m)}\rangle$$

$$= \mathbb{E}_{m,0}\langle\widehat{A}_i^{(m)}\theta^* + \widehat{b}_i^{(m)}, \mathbb{E}_{m,i}\Big[\widehat{A}_j^{(m)}\theta^* + \widehat{b}_j^{(m)}\Big]\rangle$$

$$\leq \Big[(1+\gamma)R_\theta + r_{\max}\Big]\rho_{\max}\Big(1 + \frac{\gamma\rho_{\max}}{\min|\lambda(C)|}\Big)\cdot\mathbb{E}_{m,0}\|\mathbb{E}_{m,i}\Big[\Big(\widehat{A}_j^{(m)}\theta^* - \widehat{A}\theta^*\Big) + \widehat{A}\theta^* + \Big(\widehat{b}_j^{(m)} - \widehat{b}\Big) + \widehat{b}\Big]\|$$

$$\leq \Big[(1+\gamma)R_\theta + r_{\max}\Big]\rho_{\max}\Big(1 + \frac{\gamma\rho_{\max}}{\min|\lambda(C)|}\Big)\cdot\Big[\mathbb{E}_{m,0}\Big(\|\mathbb{E}_{m,i}\widehat{A}_j^{(m)} - \widehat{A}\|R_\theta\Big) + \mathbb{E}_{m,0}\Big(\|\mathbb{E}_{m,i}\widehat{b}_j^{(m)} - \widehat{b}\|\Big)\Big] \tag{64}$$

815 where in the last inequality we apply the equation $\widehat{A}\theta^* = (A - BC^{-1}A)(-A^{-1}b) = -b + BC^{-1}b =$
816 $-\widehat{b}$. Moreover, by Assumption 2.3, we have

$$\|\mathbb{E}_{m,i}\widehat{A}_j^{(m)} - \widehat{A}\| = \|\int\widehat{A}(s)\,\mathrm{d}\mathbb{P}(\cdot|s_{j-i}^{(m)}) - \int\widehat{A}(s)\,\mathrm{d}\mu_{\pi_b}\|$$

$$\leq \|\widehat{A}(s)\|\cdot\mathrm{dist}(\mathbb{P}(\cdot|s_{j-i}^{(m)}), \mu_{\pi_b})$$

$$\leq (1+\gamma)\rho_{\max}\Big(1 + \frac{\gamma\rho_{\max}}{\min|\lambda(C)|}\Big)\kappa\rho^{j-i}. \tag{65}$$

817 Similarly,

$$\|\mathbb{E}_{m,i}\widehat{b}_j^{(m)} - \widehat{b}\| \leq \rho_{\max}r_{\max}\Big(1 + \frac{\gamma\rho_{\max}}{\min|\lambda(C)|}\Big)\kappa\rho^{j-i}. \tag{66}$$

818 Substituting eq. (65) and eq. (66) into eq. (64) yields that

$$\mathbb{E}_{m,0}\sum_{i\neq j}\langle\widehat{A}_i^{(m)}\theta^* + \widehat{b}_i^{(m)}, \widehat{A}_j^{(m)}\theta^* + \widehat{b}_j^{(m)}\rangle$$

$$\leq \Big[(1+\gamma)R_\theta + r_{\max}\Big]^2\rho_{\max}^2\Big(1 + \frac{\gamma\rho_{\max}}{\min|\lambda(C)|}\Big)^2\cdot\kappa\frac{2M\rho}{1-\rho},$$

819 which leads to the desired bound

$$\mathbb{E}_{m,0}\|\widehat{A}^{(m)}\theta^* + \widehat{b}^{(m)}\|^2 \leq \frac{1}{M}\cdot\Big[(1+\gamma)R_\theta + r_{\max}\Big]^2\rho_{\max}^2\Big(1 + \frac{\gamma\rho_{\max}}{\min|\lambda(C)|}\Big)^2\cdot\Big(1 + \kappa\frac{2\rho}{1-\rho}\Big).$$

820 $\qquad\square$

821 **Lemma I.2.** *Under the same assumptions as those of Theorem 4.1, the following inequality holds:*

$$\mathbb{E}_{m,0}\langle \theta_t^{(m)} - \theta^*, (\widehat{A}^{(m)} - \widehat{A})\theta_t^{(m)} + (\widehat{b}^{(m)} - \widehat{b})\rangle \leq \frac{\lambda_{\widehat{A}}}{4}\mathbb{E}_{m,0}\|\theta_t^{(m)} - \theta^*\|^2 + \frac{K_2}{M},$$

822 *where $K_2$ is defined as*

$$K_2 := \frac{2}{\lambda_{\widehat{A}}}\Big[R_\theta^2(1+\gamma)^2 + r_{\max}^2\Big]\cdot 4\rho_{\max}^2\Big(1 + \frac{\gamma\rho_{\max}}{\min|\lambda(C)|}\Big)^2\Big[1 + \kappa\frac{\rho}{1-\rho}\Big]. \qquad (67)$$

823 *Proof.* By the polarization identity and the Jensen's inequality, we obtain that

$$\mathbb{E}_{m,0}\langle \theta_t^{(m)} - \theta^*, (\widehat{A}^{(m)} - \widehat{A})\theta_t^{(m)} + (\widehat{b}^{(m)} - \widehat{b})\rangle$$

$$\leq \frac{1}{2}\cdot\frac{\lambda_{\widehat{A}}}{2}\mathbb{E}_{m,0}\|\theta_t^{(m)} - \theta^*\|^2 + \frac{1}{2}\cdot\frac{2}{\lambda_{\widehat{A}}}\mathbb{E}_{m,0}\|(\widehat{A}^{(m)} - \widehat{A})\theta_t^{(m)} + (\widehat{b}^{(m)} - \widehat{b})\|^2$$

$$\leq \frac{\lambda_{\widehat{A}}}{4}\mathbb{E}_{m,0}\|\theta_t^{(m)} - \theta^*\|^2 + \frac{2}{\lambda_{\widehat{A}}}R_\theta^2\mathbb{E}_{m,0}\|\widehat{A}^{(m)} - \widehat{A}\|^2 + \frac{2}{\lambda_{\widehat{A}}}\mathbb{E}_{m,0}\|\widehat{b}^{(m)} - \widehat{b}\|^2.$$

824 Next, we further bound $\mathbb{E}_{m,0}\|\widehat{A}^{(m)} - \widehat{A}\|^2$ and $\mathbb{E}_{m,0}\|\widehat{b}^{(m)} - \widehat{b}\|^2$, respectively.

$$\mathbb{E}_{m,0}\|\widehat{A}^{(m)} - \widehat{A}\|^2 \leq \mathbb{E}_{m,0}\|\widehat{A}^{(m)} - \widehat{A}\|_F^2$$

$$\leq \frac{1}{M^2}\mathbb{E}_{m,0}\Big[\sum_{i=j}\|\widehat{A}_i^{(m)} - \widehat{A}\|_F^2 + \sum_{i\neq j}\langle\widehat{A}_i^{(m)} - \widehat{A}, \widehat{A}_j^{(m)} - \widehat{A}\rangle\Big]$$

$$\leq \frac{1}{M^2}\mathbb{E}_{m,0}\Big[4(1+\gamma)^2\rho_{\max}^2\Big(1 + \frac{\gamma\rho_{\max}}{\min|\lambda(C)|}\Big)^2 M + \sum_{i\neq j}\langle\widehat{A}_i^{(m)} - \widehat{A}, \widehat{A}_j^{(m)} - \widehat{A}\rangle\Big].$$

825 For the last inner product term, without loss of generality, assume $i < j$ and we obtain that

$$\mathbb{E}_{m,0}\langle\widehat{A}_i^{(m)} - \widehat{A}, \widehat{A}_j^{(m)} - \widehat{A}\rangle$$

$$\leq \mathbb{E}_{m,0}\langle\widehat{A}_i^{(m)} - \widehat{A}, \mathbb{E}_{m,i}\widehat{A}_j^{(m)} - \widehat{A}\rangle$$

$$\leq 2(1+\gamma)\rho_{\max}\Big(1 + \frac{\gamma\rho_{\max}}{\min|\lambda(C)|}\Big)\mathbb{E}_{m,0}\|\mathbb{E}_{m,i}\widehat{A}_j^{(m)} - \widehat{A}\|_F$$

$$\leq 2(1+\gamma)^2\rho_{\max}^2\Big(1 + \frac{\gamma\rho_{\max}}{\min|\lambda(C)|}\Big)^2\kappa\rho^{j-i}. \qquad (68)$$

826 Summing the above inequality over all $i \neq j$ yields that

$$\mathbb{E}_{m,0}\sum_{i\neq j}\langle\widehat{A}_i^{(m)} - \widehat{A}, \widehat{A}_j^{(m)} - \widehat{A}\rangle \leq 4(1+\gamma)^2\rho_{\max}^2\Big(1 + \frac{\gamma\rho_{\max}}{\min|\lambda(C)|}\Big)^2\kappa\frac{M\rho}{1-\rho},$$

827 which implies that

$$\mathbb{E}_{m,0}\|\widehat{A}^{(m)} - \widehat{A}\|^2 \leq \frac{1}{M}\cdot 4(1+\gamma)^2\rho_{\max}^2\Big(1 + \frac{\gamma\rho_{\max}}{\min|\lambda(C)|}\Big)^2\Big[1 + \kappa\frac{\rho}{1-\rho}\Big]. \qquad (69)$$

828 Following the same approach, we obtain that

$$\mathbb{E}_{m,0}\|\widehat{b}^{(m)} - \widehat{b}\|^2 \leq \frac{1}{M}\cdot 4\rho_{\max}^2 r_{\max}^2\Big(1 + \frac{\gamma\rho_{\max}}{\min|\lambda(C)|}\Big)^2\Big[1 + \kappa\frac{\rho}{1-\rho}\Big]. \qquad (70)$$

829 Combining eqs. (68), (69) and (70), we obtain that

$$\mathbb{E}_{m,0}\langle\theta_t^{(m)} - \theta^*, (\widehat{A}^{(m)} - \widehat{A})\theta_t^{(m)} + (\widehat{b}^{(m)} - \widehat{b})\rangle$$

$$\leq \frac{\lambda_{\widehat{A}}}{4}\mathbb{E}_{m,0}\|\theta_t^{(m)} - \theta^*\|^2 + \frac{1}{M}\cdot\frac{2}{\lambda_{\widehat{A}}}\Big[R_\theta^2(1+\gamma)^2 + r_{\max}^2\Big]\cdot 4\rho_{\max}^2\Big(1 + \frac{\gamma\rho_{\max}}{\min|\lambda(C)|}\Big)^2\Big[1 + \kappa\frac{\rho}{1-\rho}\Big].$$

830 $\square$

831 **Lemma I.3.** *Under the same assumption as those of Theorem 4.1, the following inequality holds:*

$$\mathbb{E}_{m,0}\langle z_t^{(m)}, \bar{A}^{(m)}\theta_t^{(m)} + \bar{b}^{(m)}\rangle \leq \frac{\lambda_C}{8}\mathbb{E}_{m,0}\|z_t^{(m)}\|^2 + \frac{K_3}{M}$$

832 *where $K_3$ is defined as*

$$K_3 := \left(\frac{32}{\lambda_C}\left[R_\theta^2(1+\gamma)^2 + r_{\max}^2\right]\cdot\rho_{\max}^2 + \frac{16}{\lambda_C}\frac{\rho_{\max}(1+\gamma)R_\theta + \rho_{\max}r_{\max}}{\min|\lambda(C)|}\right)\left[1 + \kappa\frac{\rho}{1-\rho}\right]. \tag{71}$$

833 *Proof.* Note that the following equations hold.

$$\mathbb{E}_{m,0}\langle z_t^{(m)}, \bar{A}^{(m)}\theta_t^{(m)} + \bar{b}^{(m)}\rangle = \mathbb{E}_{m,0}\langle z_t^{(m)}, (A^{(m)} - C^{(m)}C^{-1}A)\theta_t^{(m)} + b^{(m)} - C^{(m)}C^{-1}b\rangle$$
$$= \mathbb{E}_{m,0}\langle z_t^{(m)}, (A^{(m)} - A)\theta_t^{(m)} + b^{(m)} - b\rangle$$
$$- \mathbb{E}_{m,0}\langle z_t^{(m)}, (C^{(m)} - C)C^{-1}A\theta_t^{(m)} + (C^{(m)} - C)C^{-1}b\rangle. \tag{72}$$

834 For the first term of eq. (72), we bound it using the polarization identity and Jensen's inequality as
835 follows.

$$\mathbb{E}_{m,0}\langle z_t^{(m)}, (A^{(m)} - A)\theta_t^{(m)} + b^{(m)} - b\rangle$$
$$\leq \frac{1}{2}\cdot\frac{\lambda_C}{8}\mathbb{E}_{m,0}\|z_t^{(m)}\|^2 + \frac{1}{2}\cdot\frac{8}{\lambda_C}\mathbb{E}_{m,0}\|(A^{(m)} - A)\theta_t^{(m)} + b^{(m)} - b\|^2$$
$$\leq \frac{\lambda_C}{16}\mathbb{E}_{m,0}\|z_t^{(m)}\|^2 + \frac{8}{\lambda_C}\left(R_\theta^2\mathbb{E}_{m,0}\|A^{(m)} - A\|_F^2 + \mathbb{E}_{m,0}\|b^{(m)} - b\|^2\right).$$

836 Then, following a similar proof logic as that of Lemma I.2, we obtain that

$$\mathbb{E}_{m,0}\|A^{(m)} - A\|_F^2 \leq \frac{1}{M}\cdot 4(1+\gamma)^2\rho_{\max}^2\left[1 + \kappa\frac{\rho}{1-\rho}\right],$$

837 and

$$\mathbb{E}_{m,0}\|b^{(m)} - b\|^2 \leq \frac{1}{M}\cdot 4\rho_{\max}^2 r_{\max}^2\left[1 + \kappa\frac{\rho}{1-\rho}\right].$$

838 Combining the above bounds, we obtain the following bound for the first term of eq. (72)

$$\mathbb{E}_{m,0}\langle z_t^{(m)}, (A^{(m)} - A)\theta_t^{(m)} + b^{(m)} - b\rangle$$
$$\leq \frac{\lambda_C}{16}\mathbb{E}_{m,0}\|z_t^{(m)}\|^2 + \frac{1}{M}\cdot\frac{32}{\lambda_C}\left[R_\theta^2(1+\gamma)^2 + r_{\max}^2\right]\cdot\rho_{\max}^2\left[1 + \kappa\frac{\rho}{1-\rho}\right]. \tag{73}$$

839 For the second term of eq. (72), we have

$$- \mathbb{E}_{m,0}\langle z_t^{(m)}, (C^{(m)} - C)C^{-1}A\theta_t^{(m)} + (C^{(m)} - C)C^{-1}b\rangle$$
$$\leq \frac{1}{2}\cdot\frac{\lambda_C}{8}\mathbb{E}_{m,0}\|z_t^{(m)}\|^2 + \frac{1}{2}\cdot\frac{8}{\lambda_C}\mathbb{E}_{m,0}\|C^{(m)} - C\|_F^2\cdot\frac{\rho_{\max}(1+\gamma)R_\theta + \rho_{\max}r_{\max}}{\min|\lambda(C)|}.$$

840 Moreover, note that

$$\mathbb{E}_{m,0}\|C^{(m)} - C\|_F^2 \leq \frac{1}{M}\cdot 4\left[1 + \kappa\frac{\rho}{1-\rho}\right],$$

841 and therefore we have

$$- \mathbb{E}_{m,0}\langle z_t^{(m)}, (C^{(m)} - C)C^{-1}A\theta_t^{(m)} + (C^{(m)} - C)C^{-1}b\rangle$$
$$\leq \frac{\lambda_C}{16}\mathbb{E}_{m,0}\|z_t^{(m)}\|^2 + \frac{1}{M}\cdot\frac{16}{\lambda_C}\frac{\rho_{\max}(1+\gamma)R_\theta + \rho_{\max}r_{\max}}{\min|\lambda(C)|}\left[1 + \kappa\frac{\rho}{1-\rho}\right] \tag{74}$$

842 Combining eqs. (72), (73) and (74), we obtain that

$$\mathbb{E}_{m,0}\langle z_t^{(m)}, \bar{A}^{(m)}\theta_t^{(m)} + \bar{b}^{(m)}\rangle$$
$$\leq \frac{\lambda_C}{8}\mathbb{E}_{m,0}\|z_t^{(m)}\|^2 + \frac{1}{M}\cdot\left(\frac{32}{\lambda_C}\left[R_\theta^2(1+\gamma)^2 + r_{\max}^2\right]\cdot\rho_{\max}^2 + \frac{16}{\lambda_C}\frac{\rho_{\max}(1+\gamma)R_\theta + \rho_{\max}r_{\max}}{\min|\lambda(C)|}\right)\left[1 + \kappa\frac{\rho}{1-\rho}\right].$$

843 $\qquad\qquad\qquad\qquad\qquad\qquad\qquad\qquad\qquad\qquad\qquad\qquad\qquad\qquad\qquad\qquad\qquad\qquad\qquad\qquad\qquad\square$

**Lemma I.4.** *Under the same assumptions as those of Theorem 4.1, the following inequality holds:*

$$\mathbb{E}_{m,0}\langle z_t^{(m)}, (C^{(m)} - C)z_t^{(m)}\rangle \leq \frac{\lambda_C}{12}\mathbb{E}_{m,0}\|z_t^{(m)}\|^2 + \frac{K_4}{M},$$

*where $K_4$ is defined as*

$$K_4 := \frac{12}{\lambda_C}R_w^2\Big[1 + \kappa\frac{\rho}{1-\rho}\Big]. \tag{75}$$

*Proof.* The proof is very similar to that of Lemma I.2 and Lemma I.3, and we only outline the proof below.

$$\mathbb{E}_{m,0}\langle z_t^{(m)}, (C^{(m)} - C)z_t^{(m)}\rangle \leq \frac{1}{2}\frac{\lambda_C}{6}\mathbb{E}_{m,0}\|z_t^{(m)}\|^2 + \frac{1}{2}\frac{6}{\lambda_C}R_w^2\mathbb{E}_{m,0}\|C^{(m)} - C\|_F^2$$

$$\leq \frac{\lambda_C}{12}\mathbb{E}_{m,0}\|z_t^{(m)}\|^2 + \frac{1}{M}\cdot\frac{12}{\lambda_C}R_w^2\Big[1 + \kappa\frac{\rho}{1-\rho}\Big].$$

$\square$

**Lemma I.5.** *Under the same assumptions as those of Theorem 4.1, the following inequality holds:*

$$\mathbb{E}_{m,0}\|\bar{A}^{(m)}\theta^* + \bar{b}^{(m)}\|^2 \leq \frac{K_5}{M}$$

*where $K_5$ is defined as*

$$K_5 := \Big[(1+\gamma)R_\theta + r_{\max}\Big]^2\rho_{\max}^2\Big(1 + \frac{1}{\min|\lambda(C)|}\Big)^2\cdot\Big(1 + \kappa\frac{2\rho}{1-\rho}\Big). \tag{76}$$

*Proof.* The proof is similar to that of Lemma I.1 and we outline the proof below. We obtain that

$$\mathbb{E}_{m,0}\|\bar{A}^{(m)}\theta^* + \bar{b}^{(m)}\|^2$$

$$\leq \frac{1}{M^2}\mathbb{E}_{m,0}\Big[\sum_{i=j}\|\bar{A}_i^{(m)}\theta^* + \bar{b}_i^{(m)}\|^2 + \sum_{i\neq j}\langle\bar{A}_i^{(m)}\theta^* + \bar{b}_i^{(m)}, \bar{A}_j^{(m)}\theta^* + \bar{b}_j^{(m)}\rangle\Big]$$

$$\leq \frac{1}{M^2}\Big[M\cdot\Big[(1+\gamma)R_\theta + r_{\max}\Big]^2\rho_{\max}^2\Big(1 + \frac{1}{\min|\lambda(C)|}\Big)^2 + \sum_{i\neq j}\mathbb{E}_{m,0}\langle\bar{A}_i^{(m)}\theta^* + \bar{b}_i^{(m)}, \bar{A}_j^{(m)}\theta^* + \bar{b}_j^{(m)}\rangle\Big]. \tag{77}$$

Cconsider the last term of eq. (77). Without loss of generality, assume that $i < j$. Then

$$\mathbb{E}_{m,0}\langle\bar{A}_i^{(m)}\theta^* + \bar{b}_i^{(m)}, \bar{A}_j^{(m)}\theta^* + \bar{b}_j^{(m)}\rangle$$

$$=\mathbb{E}_{m,0}\langle\bar{A}_i^{(m)}\theta^* + \bar{b}_i^{(m)}, \mathbb{E}_{m,i}\bar{A}_j^{(m)}\theta^* + \mathbb{E}_{m,i}\bar{b}_j^{(m)}\rangle$$

$$\leq\mathbb{E}_{m,0}\|\bar{A}_i^{(m)}\theta^* + \bar{b}_i^{(m)}\|\cdot\mathbb{E}_{m,0}\Big[\|(\mathbb{E}_{m,i}\bar{A}_j^{(m)} - \bar{A})\theta^*\| + \|\mathbb{E}_{m,i}\bar{b}_j^{(m)} - \bar{b}\|\Big]$$

$$\leq\Big[(1+\gamma)R_\theta + r_{\max}\Big]^2\rho_{\max}^2\Big(1 + \frac{1}{\min|\lambda(C)|}\Big)^2\cdot\kappa\rho^{j-i}$$

Summing the above inequality over all $i\neq j$ and substituting it into eq. (77), we obtain that

$$\mathbb{E}_{m,0}\|\bar{A}^{(m)}\theta^* + \bar{b}^{(m)}\|^2$$

$$\leq\frac{1}{M}\cdot\Big[(1+\gamma)R_\theta + r_{\max}\Big]^2\rho_{\max}^2\Big(1 + \frac{1}{\min|\lambda(C)|}\Big)^2\cdot\Big(1 + \kappa\frac{2\rho}{1-\rho}\Big).$$

$\square$

**Lemma I.6** (One-Step Update of $\theta_t^{(m)}$)**.** *Under the same assumption as those of Theorem 4.1, the square norm of one-step update of $\theta_t^{(m)}$ using Algorithm 2 is bounded as*

$$\mathbb{E}_{m,0}\|G_t^{(m)}(\theta_t^{(m)}, z_t^{(m)}) - G_t^{(m)}(\tilde{\theta}^{(m-1)}, \tilde{z}^{(m-1)}) + G^{(m)}(\tilde{\theta}^{(m-1)}, \tilde{z}^{(m-1)})\|^2$$

$$\leq 5(1+\gamma)^2\rho_{\max}^2\Big(1 + \frac{\gamma\rho_{\max}}{\min|\lambda(C)|}\Big)^2\Big(\mathbb{E}_{m,0}\|\theta_t^{(m)} - \theta^*\|^2 + \mathbb{E}_{m,0}\|\tilde{\theta}^{(m-1)} - \theta^*\|^2\Big)$$

$$+ 5\gamma^2\rho_{\max}^2\Big(\mathbb{E}_{m,0}\|z_t^{(m)}\|^2 + \mathbb{E}_{m,0}\|\tilde{z}^{(m-1)}\|^2\Big) + \frac{5K_1}{M}$$

*where $K_1$ is specified in (63) of Lemma I.1.*

*Proof.* By the definitions of $G_t^{(m)}(\cdot)$ and $G^{(m)}(\cdot)$ and the one-step update of $\theta_t^{(m)}$, we obtain that

$$\|G_t^{(m)}(\theta_t^{(m)}, z_t^{(m)}) - G_t^{(m)}(\tilde{\theta}^{(m-1)}, \tilde{z}^{(m-1)}) + G^{(m)}(\tilde{\theta}^{(m-1)}, \tilde{z}^{(m-1)})\|^2$$

$$=\|\widehat{A}_t^{(m)}\theta_t^{(m)} + \widehat{b}_t^{(m)} + B_t^{(m)} z_t^{(m)} - \widehat{A}_t^{(m)}\tilde{\theta}^{(m-1)} - \widehat{b}_t^{(m)} - B_t^{(m)}\tilde{z}^{(m-1)} + \widehat{A}^{(m)}\tilde{\theta}^{(m-1)} + \widehat{b}^{(m)} + B^{(m)}\tilde{z}^{(m-1)}\|^2$$

$$=\|\left(\widehat{A}_t^{(m)}\theta_t^{(m)} - \widehat{A}_t^{(m)}\theta^*\right) + \left(\widehat{A}_t^{(m)}\theta^* - \widehat{A}_t^{(m)}\tilde{\theta}^{(m-1)} + \widehat{A}^{(m)}\tilde{\theta}^{(m-1)} - \widehat{A}^{(m)}\theta^*\right) + \left(\widehat{A}^{(m)}\theta^* + \widehat{b}^{(m)}\right)$$

$$+ B_t^{(m)} z_t^{(m)} - B_t^{(m)}\tilde{z}^{(m-1)} + B^{(m)}\tilde{z}^{(m-1)}\|^2$$

$$\leq 5\|\widehat{A}_t^{(m)}\|^2\|\theta_t^{(m)} - \theta^*\|^2 + 5\|\left(\widehat{A}_t^{(m)}\tilde{\theta}^{(m-1)} - \widehat{A}_t^{(m)}\theta^*\right) - \left(\widehat{A}^{(m)}\tilde{\theta}^{(m-1)} - \widehat{A}^{(m)}\theta^*\right)\|^2 + 5\|\widehat{A}^{(m)}\theta^* + \widehat{b}^{(m)}\|^2$$

$$+ 5\|B_t^{(m)}\|^2\|z_t^{(m)}\|^2 + 5\|B_t^{(m)}\tilde{z}^{(m-1)} - B^{(m)}\tilde{z}^{(m-1)}\|^2 \tag{78}$$

where the last step applies Jensen's inequality. Next, we bound the third term of eq. (78) $\|\widehat{A}^{(m)}\theta^* + \widehat{b}^{(m)}\|^2$ using Lemma I.1, and note that the second term of eq. (78) can be bounded as

$$\mathbb{E}_{m,t-1}\|\left(\widehat{A}_t^{(m)}\tilde{\theta}^{(m-1)} - \widehat{A}_t^{(m)}\theta^*\right) - \left(\widehat{A}^{(m)}\tilde{\theta}^{(m-1)} - \widehat{A}^{(m)}\theta^*\right)\|^2$$

$$=\mathrm{Var}_{m,t-1}\left(\widehat{A}_t^{(m)}\tilde{\theta}^{(m-1)} - \widehat{A}_t^{(m)}\theta^*\right)$$

$$\leq\mathbb{E}_{m,t-1}\left(\widehat{A}_t^{(m)}\tilde{\theta}^{(m-1)} - \widehat{A}_t^{(m)}\theta^*\right)^2$$

$$\leq\mathbb{E}_{m,t-1}\|\widehat{A}_t^{(m)}\|^2\|\tilde{\theta}^{(m-1)} - \theta^*\|^2, \tag{79}$$

and similarly, the last term of eq. (78) can be bounded as

$$\mathbb{E}_{m,t-1}\|B_t^{(m)}\tilde{z}^{(m-1)} - B^{(m)}\tilde{z}^{(m-1)}\|^2 \leq \mathbb{E}_{m,t-1}\|B_t^{(m)}\|^2\|\tilde{z}^{(m-1)}\|^2 \tag{80}$$

Combining Lemma I.1, (80), (79), and (78), we get the desired bound

$$\mathbb{E}_{m,0}\|G_t^{(m)}(\theta_t^{(m)}, z_t^{(m)}) - G_t^{(m)}(\tilde{\theta}^{(m-1)}, \tilde{z}^{(m-1)}) + G^{(m)}(\tilde{\theta}^{(m-1)}, \tilde{z}^{(m-1)})\|^2$$

$$\leq 5(1+\gamma)^2\rho_{\max}^2\left(1 + \frac{\gamma\rho_{\max}}{\min|\lambda(C)|}\right)^2\left(\mathbb{E}_{m,0}\|\theta_t^{(m)} - \theta^*\|^2 + \mathbb{E}_{m,0}\|\tilde{\theta}^{(m-1)} - \theta^*\|^2\right)$$

$$+ 5\gamma^2\rho_{\max}^2\left(\mathbb{E}_{m,0}\|z_t^{(m)}\|^2 + \mathbb{E}_{m,0}\|\tilde{z}^{(m-1)}\|^2\right) + \frac{1}{M}\cdot 5K_1.$$

$\square$

**Lemma I.7** (One-Step Update of $z_t^{(m)}$). *Under the same assumption as those of Theorem 4.1, the square norm of one-step update of $z_t^{(m)}$ using Algorithm 2 is bounded as*

$$\mathbb{E}_{m,0}\|H_t^{(m)}(\theta_t^{(m)}, z_t^{(m)}) - H_t^{(m)}(\tilde{\theta}^{(m-1)}, \tilde{z}^{(m-1)}) + H^{(m)}(\tilde{\theta}^{(m-1)}, \tilde{z}^{(m-1)})\|^2$$

$$\leq 5(1+\gamma)^2\rho_{\max}^2\left(1 + \frac{1}{\min|\lambda(C)|}\right)^2\left(\mathbb{E}_{m,0}\|\theta_t^{(m)} - \theta^*\|^2 + \mathbb{E}_{m,0}\|\tilde{\theta}^{(m-1)} - \theta^*\|^2\right)$$

$$+ 5\left(\mathbb{E}_{m,0}\|z_t^{(m)}\|^2 + \mathbb{E}_{m,0}\|\tilde{z}^{(m-1)}\|^2\right) + \frac{5K_5}{M}.$$

*where $K_5$ is specified in eq. (76) of Lemma I.5.*

*Proof.* The proof is very similar to that of Lemma I.6 and we outline the proof below. We obtain that

$$\|H_t^{(m)}(\theta_t^{(m)}, z_t^{(m)}) - H_t^{(m)}(\tilde{\theta}^{(m-1)}, \tilde{z}^{(m-1)}) + H^{(m)}(\tilde{\theta}^{(m-1)}, \tilde{z}^{(m-1)})\|^2$$

$$\leq 5\|\bar{A}_t^{(m)}\|^2\|\theta_t^{(m)} - \theta^*\|^2 + 5\|\left(\bar{A}_t^{(m)}\tilde{\theta}^{(m-1)} - \bar{A}_t^{(m)}\theta^*\right) - \left(\bar{A}^{(m)}\tilde{\theta}^{(m-1)} - \bar{A}^{(m)}\theta^*\right)\|^2 + 5\|\bar{A}^{(m)}\theta^* + \bar{b}^{(m)}\|^2$$

$$+ 5\|C_t^{(m)}\|^2\|z_t^{(m)}\|^2 + 5\|C_t^{(m)}\tilde{z}^{(m-1)} - C^{(m)}\tilde{z}^{(m-1)}\|^2.$$

Taking $\mathbb{E}_{m,0}$ on both sides of the above inequality yields that

$$\mathbb{E}_{m,0}\|H_t^{(m)}(\theta_t^{(m)}, z_t^{(m)}) - H_t^{(m)}(\tilde{\theta}^{(m-1)}, \tilde{z}^{(m-1)}) + H^{(m)}(\tilde{\theta}^{(m-1)}, \tilde{z}^{(m-1)})\|^2$$

$$\leq 5(1+\gamma)^2\rho_{\max}^2\left(1 + \frac{1}{\min|\lambda(C)|}\right)^2\left(\mathbb{E}_{m,0}\|\theta_t^{(m)} - \theta^*\|^2 + \mathbb{E}_{m,0}\|\tilde{\theta}^{(m-1)} - \theta^*\|^2\right)$$

$$+ 5\left(\mathbb{E}_{m,0}\|z_t^{(m)}\|^2 + \mathbb{E}_{m,0}\|\tilde{z}^{(m-1)}\|^2\right) + \frac{5K_5}{M}.$$

$\square$

 # J  Other Lemmas on Constant-Level Bounds

869 **Lemma J.1.** *The following constant-level bounds hold.*

870 $\bullet$ $\|\widehat{A}_t^{(m)}\| \leq (1+\gamma)\rho_{\max}\big(1 + \frac{\gamma\rho_{\max}}{\min|\lambda(C)|}\big)$

871 $\bullet$ $\|\widehat{b}^{(m)}\| \leq \rho_{\max}r_{\max}\big(1 + \frac{\gamma\rho_{\max}}{\min|\lambda(C)|}\big)$

872 $\bullet$ $\|\bar{A}_t^{(m)}\| \leq (1+\gamma)\rho_{\max}\big(1 + \frac{1}{\min|\lambda(C)|}\big)$

873 $\bullet$ $\|\bar{b}_t^{(m)}\| \leq \rho_{\max}r_{\max}\big(1 + \frac{1}{\min|\lambda(C)|}\big)$

874 *Proof.* We prove the first inequality as an example. For the rest of them, we omit the proof details.
875 Recall that $\widehat{A}_t^{(m)} := A_t^{(m)} - B_t^{(m)}C^{-1}A$, from which we obtain that

$$
\begin{aligned}
\|\widehat{A}_t^{(m)}\| &= \|A_t^{(m)} - B_t^{(m)}C^{-1}A\| \\
&\leq \|A_t^{(m)}\| + \|B_t^{(m)}\|\|C^{-1}\|\|A\| \\
&\leq (1+\gamma)\rho_{\max} + \gamma\rho_{\max}\frac{1}{\min|\lambda(C)|}(1+\gamma)\rho_{\max} \\
&= (1+\gamma)\rho_{\max}\big(1 + \frac{\gamma\rho_{\max}}{\min|\lambda(C)|}\big),
\end{aligned}
$$

876 where the third step applies Assumption 2.2 to the definitions in eq. (1), more precisely,
877 $\|A_t^{(m)}\| = \|\rho(s_t^{(m)}, a_t^{(m)})\phi(s_t^{(m)})(\gamma\phi(s_{t+1}^{(m)}) - \phi(s_t^{(m)})^\top\| \leq (1+\gamma)\rho_{\max}$ and $\|B_t^{(m)}\| = \| -$
878 $\gamma\rho(s_t^{(m)}, a_t^{(m)})\phi(s_{t+1}^{(m)})\phi(s_t^{(m)})^\top\|$. Similarly, we can obtain the following results:

$$
\begin{aligned}
\|\widehat{b}_t^{(m)}\| &= \|b_t^{(m)} - B_t^{(m)}C^{-1}b\| \\
&\leq \|b_t^{(m)}\| + \|B_t^{(m)}\|\|C^{-1}\|\|b\| \\
&\leq \rho_{\max}r_{\max} + \gamma\rho_{\max}\frac{1}{\min|\lambda(C)|}\rho_{\max}r_{\max} \\
&= \rho_{\max}r_{\max}\big(1 + \frac{\gamma\rho_{\max}}{\min|\lambda(C)|}\big);
\end{aligned}
$$

879

$$
\begin{aligned}
\|\bar{A}_t^{(m)}\| &= \|A_t^{(m)} - C_t^{(m)}C^{-1}A\| \\
&\leq (1+\gamma)\rho_{\max} + \frac{1}{\min|\lambda(C)|}(1+\gamma)\rho_{\max} \\
&= (1+\gamma)\rho_{\max}\big(1 + \frac{1}{\min|\lambda(C)|}\big);
\end{aligned}
$$

880

$$
\begin{aligned}
\|\bar{b}_t^{(m)}\| &= b_t^{(m)} - C_t^{(m)}C^{-1}b \\
&\leq \rho_{\max}r_{\max} + \frac{1}{\min|\lambda(C)|}\rho_{\max}r_{\max} \\
&= \rho_{\max}r_{\max}\big(1 + \frac{1}{\min|\lambda(C)|}\big).
\end{aligned}
$$

881 $\square$

882 **Lemma J.2.** *The following constant-level bound holds.*

$$
\|\widehat{A}_t^{(m)}\theta^* + \widehat{b}_t^{(m)}\| \leq \big[(1+\gamma)R_\theta + r_{\max}\big]\rho_{\max}\big(1 + \frac{\gamma\rho_{\max}}{\min|\lambda(C)|}\big).
$$

883 *Proof.* Combining Lemma J.1 and the assumption that $\|\theta^*\| \leq R_\theta$, we have

$$
\|\widehat{A}_t^{(m)}\theta^* + \widehat{b}_t^{(m)}\| \leq \|\widehat{A}_t^{(m)}\|R_\theta + \|\widehat{b}_t^{(m)}\|
$$

$$\leq (1+\gamma)\rho_{\max}\big(1+\frac{\gamma\rho_{\max}}{\min|\lambda(C)|}\big)R_\theta + \rho_{\max}r_{\max}\big(1+\frac{\gamma\rho_{\max}}{\min|\lambda(C)|}\big)$$

$$= \big[(1+\gamma)R_\theta + r_{\max}\big]\rho_{\max}\big(1+\frac{\gamma\rho_{\max}}{\min|\lambda(C)|}\big).$$

884 □

885 **Lemma J.3.** *The following constant-level bound holds.*

$$\|\bar{A}^{(m)}\theta^* + \bar{b}^{(m)}\| \leq \big[(1+\gamma)R_\theta + r_{\max}\big]\rho_{\max}\big(1+\frac{1}{\min|\lambda(C)|}\big)$$

886 *Proof.* Combining Lemma J.1 and $\|\theta^*\| \leq R_\theta$, we have

$$\|\bar{A}^{(m)}\theta^* + \bar{b}^{(m)}\| \leq \|\bar{A}_t^{(m)}\|R_\theta + \|\bar{b}_t^{(m)}\|$$

$$\leq (1+\gamma)\rho_{\max}\big(1+\frac{1}{\min|\lambda(C)|}\big)R_\theta + \rho_{\max}r_{\max}\big(1+\frac{1}{\min|\lambda(C)|}\big)$$

$$= \big[(1+\gamma)R_\theta + r_{\max}\big]\rho_{\max}\big(1+\frac{1}{\min|\lambda(C)|}\big).$$

887 □

888 **Lemma J.4** (One-Step Update of $\theta_t^{(m)}$). *The upper bound of one-step update of $\theta_t^{(m)}$ given in both*
889 *Algorithm 1 and Algorithm 2 is given by*

$$\|G_t^{(m)}(\theta_t^{(m)}, z_t^{(m)}) - G_t^{(m)}(\tilde{\theta}^{(m-1)}, \tilde{z}^{(m-1)}) + G^{(m)}(\tilde{\theta}^{(m-1)}, \tilde{z}^{(m-1)})\| \leq G_{VR},$$

890 *where $G_{VR}$ is defined as*

$$G_{VR} := 3\big[(1+\gamma)R_\theta + r_{\max}\big]\rho_{\max}\big(1+\frac{\gamma\rho_{\max}}{\min|\lambda(C)|}\big). \tag{81}$$

*Proof.* Recall that $G_t^{(m)}(\theta, z) := \widehat{A}_t^{(m)}\theta + \widehat{b}_t^{(m)} + C_t^{(m)}w$. Using Lemma J.2 and Lemma J.3, we obtain that

$$\|G_t^{(m)}(\theta, z)\| \leq \big[(1+\gamma)R_\theta + r_{\max}\big]\rho_{\max}\big(1+\frac{\gamma\rho_{\max}}{\min|\lambda(C)|}\big).$$

By the definition of $G^{(m)}(\tilde{\theta}^{(m-1)}, \tilde{z}^{(m-1)})$ and Jensen's inequality, we obtain that

$$\|G^{(m)}(\tilde{\theta}^{(m-1)}, \tilde{z}^{(m-1)})\| \leq \big[(1+\gamma)R_\theta + r_{\max}\big]\rho_{\max}\big(1+\frac{\gamma\rho_{\max}}{\min|\lambda(C)|}\big).$$

891 Combining the above upper bounds for $\|G_t^{(m)}(\theta, z)\|$ and $\|G^{(m)}(\tilde{\theta}^{(m-1)}, \tilde{z}^{(m-1)})\|$, we further
892 obtain that

$$\|G_t^{(m)}(\theta_t^{(m)}, z_t^{(m)}) - G_t^{(m)}(\tilde{\theta}^{(m-1)}, \tilde{z}^{(m-1)}) + G^{(m)}(\tilde{\theta}^{(m-1)}, \tilde{z}^{(m-1)})\|$$

$$\leq \|G_t^{(m)}(\theta_t^{(m)}, z_t^{(m)})\| + \|G_t^{(m)}(\tilde{\theta}^{(m-1)}, \tilde{z}^{(m-1)})\| + \|G^{(m)}(\tilde{\theta}^{(m-1)}, \tilde{z}^{(m-1)})\|$$

$$\leq 3\big[(1+\gamma)R_\theta + r_{\max}\big]\rho_{\max}\big(1+\frac{\gamma\rho_{\max}}{\min|\lambda(C)|}\big).$$

893 □

894 **Lemma J.5** (One-Step Update of $w_t^{(m)}$). *The upper bound of one-step update of $w_t^{(m)}$ given in both*
895 *Algorithm 1 and Algorithm 2 is given by*

$$\|H_t^{(m)}(\theta_t^{(m)}, z_t^{(m)}) - H_t^{(m)}(\tilde{\theta}^{(m-1)}, \tilde{z}^{(m-1)}) + H^{(m)}(\tilde{\theta}^{(m-1)}, \tilde{z}^{(m-1)})\| \leq H_{VR}$$

896 *where $H_{VR}$ is defined as*

$$H_{VR} := 3\big[(1+\gamma)R_\theta + r_{\max}\big]\rho_{\max}\big(1+\frac{1}{\min|\lambda(C)|}\big). \tag{82}$$

897 *Proof.* The proof is very similar to that of Lemma J.4 and we omit the proof.

$$\|H_t^{(m)}(\theta_t^{(m)}, z_t^{(m)}) - H_t^{(m)}(\tilde{\theta}^{(m-1)}, \tilde{z}^{(m-1)}) + H^{(m)}(\tilde{\theta}^{(m-1)}, \tilde{z}^{(m-1)})\|$$
$$\leq \|H_t^{(m)}(\theta_t^{(m)}, z_t^{(m)})\| + \|H_t^{(m)}(\tilde{\theta}^{(m-1)}, \tilde{z}^{(m-1)})\| + \|H^{(m)}(\tilde{\theta}^{(m-1)}, \tilde{z}^{(m-1)})\|$$
$$\leq 3\big[(1+\gamma)R_\theta + r_{\max}\big]\rho_{\max}\Big(1 + \frac{1}{\min|\lambda(C)|}\Big).$$

898 $\qquad\qquad\qquad\qquad\qquad\qquad\qquad\qquad\qquad\qquad\qquad\qquad\qquad\qquad\qquad\qquad$ □

899