[Reviews · NeurIPS 2020]

Review 1

Summary and Contributions: This paper develops a variance reduction scheme for the two time-scale TDC algorithm in the off-policy setting and establish its finite-time bounds over both iid and Markovian setting. In the iid sample settings, it is shown that the proposed algorithm achieves better sample complexity than the SOTA result. And in the Markovian sample setting, the algorithm is proved to be on par with other variance reduction algorithms and nearly matches the lower bound. Experimental results demonstrate that the proposed variance-reduced TDC achieves a smaller asymptotic convergence error than both the conventional TDC and the variance-reduced TD.

Strengths: This paper develops the first result on variance reduction for two time-scale TDC learning and derives the non-asymptotic bounds under both iid samples and Markovian samples. A new analysis technique is developed based on recursive refinement strategy.

Weaknesses: Extending TDC to variance reduction version is relatively straightforward given the previous progress in this area. The proposed algorithm is simply applying SVRG to the two time-scale variables in TDC. Therefore, the contribution & novelty in the algorithm development side is relatively low, although its theoretical analysis could be involving.

Correctness: Yes

Clarity: Yes

Relation to Prior Work: Yes

Reproducibility: Yes

Additional Feedback: The problem of non-i.i.d. sample issue could possibly be alleviated by the simple & popular approach of experience replay, which collects trajectories into replay buffer and then randomly sample transition tuples from it. This would break the Markovian correlation over samples and make the data samples virtually i.i.d. Unless the problem is strictly restricted to the online learning setting, experience replay could be a relatively cheap fix in practice. In this regard, analyzing the proposed algorithm in Markovian setting is not as significant as it seems to be. In Theorems 3.1 & 4.1, m is chosen to be m=O(log epsilon^{-1}), which leads to the omission of the terms that are relevant to the problem’s condition number. It would be more enlightening to keep the constant terms that are related to the condition number in the final sample complexity bounds (for both iid setting and the Markovian setting). It will be helpful to summarize the bounds of the proposed algorithm along with other existing algorithms’ bounds under the two settings (iid and Markovian) in a table. ******** After rebuttal ******** The author response has addressed my questions and I have raised my score.


Review 2

Summary and Contributions: This paper proposes a variance reduction scheme for the two time-scale TDC algorithm in the off-policy setting and analyze its non-asymptotic convergence rate over both i.i.d. and Markovian samples. Specifically, under i.i.d. sampling, the proposed algorithm reaches a lower sample complexity, which improves a factor by $O(\epsilon^{-2/5})$. Under Markovian sampling, the proposed algorithm reaches the best known sample complexity.

Strengths: This paper proposes two variance-reduced TDC algorithms in the off-policy setting for i.i.d. samples and Markovian samples, respectively. Moreover, it analyzes the sample complexity of the proposed algorithms under some mild assumptions. Some experimental results on Garnet problem and Frozen Lake game demonstrate the efficiency of the proposed algorithms. In particular, some theoretical analysis techniques are novelty, which are of interest to reinforcement learning and optimization communities.

Weaknesses: This paper should detail the equations (1)-(2). Let readers understand some notations such as $A_t$, $B_t$, $b_t$ and $C_t$.

Correctness: The methods in the paper are correct.

Clarity: This paper is basically well written.

Relation to Prior Work: In related work part, the paper at detail discusses the differences between the proposed methods of the previous methods.

Reproducibility: Yes

Additional Feedback: Some comments are given as follows: C1: In Theorem 3.1 & 4.1, it is required that the stepsizes satisfy alpha = beta^2/3, is there an intuitive explanation for this particular choice of stepsize? C2: After Theorem 4.2, the authors commented that they could also develop refined error bounds following the recursive refinement strategy, but decided not to do so as a preliminary bound suffices to prove the desired tight result. Can the authors very briefly elaborate on the order of the refined error bounds? It might be better to provide tight error bounds even though they do not lead to improved complexity. C3: The experiments show that VRTDC significantly out performs other competitors like TDC and VRTD. Moreover, it is observed that the training under VRTDC has much smaller variance than other training algorithms. I think this is a much desired property in RL training. Also, I suggest the authors to evaluate and compare the variance of the stochastic updates of these algorithms to further explore whether VRTDC induces a smaller optimization variance. %%%%%%%%%%%%%%%%%%%%%% Thanks for your responses. I still maintain earlier positive review and recommend acceptance.


Review 3

Summary and Contributions: This paper proposes two variance-reduced two time-scale TD algorithms, and show that convergence rates of these two algorithms to a neighborhood of the optimal solution and epsilon approximate optimal solution in both i.i.d. and Markovian settings. Multiple numerical results show that variance reduced two time-scale TD algorithms outperforms than the classical ones.

Strengths: The main strength of this work is the theoretical analysis of the two variance reduced two time-scale TD algorithms in both i.i.d. and Markovian settings. The problem is classic and until very recently there are some advances in quantifying the convergence rates of the TD algorithms to optimal solutions. This work sharpens the convergence rate of the variance reduced two time-scale TD algorithm to 1/epsilon, which is the state-of-the-art result (to my knowledge). Especially in the Markovian setting, the analysis is non-trivial.

Weaknesses: One issue is that the two time-scale TD algorithm is not well motivated. I suggest that it would be better to emphasize the importance of the two time-scale TD algorithm in sec. 2.2. Second, please show the reasons why performing the variance reduction by the two time-scale algorithm in the Markovian case is challenging in sec. 4.2 (as what the authors did in sec. 3.2). And what the difficulties of analyzing the corresponding convergence rate are.

Correctness: To my understanding, the claims make sense and the numerical results are sufficient and valid.

Clarity: Yes.

Relation to Prior Work: Acceptable. But it would be better to show a table or chat to explicitly compare this work and the existing ones in terms of problem settings, convergence rates, etc.

Reproducibility: Yes

Additional Feedback: ====================== After rebuttal =============================== Yes. If more detailed comparison would be provided, then the main contributions of this work would be more clear. I can understand that authors had elaborated on these issues in the introduction part, but a simple and clear motivating example would be more helpful.


Review 4

Summary and Contributions: This paper introduces a new off-policy method that includes a prior variance reduction technique. Results are given regarding non-asymptotic convergence rate, sample complexity, and empirical comparison to the ordinary TD methods.

Strengths: The new variance reduction TDC method for iid samples has a new low sample complexity within its class. Moreover, the paper was easy to follow.

Weaknesses: I find the utility of the provided method rather limited. The new method for Markovian samples does not give a sample complexity lower than the previous best one. This was attempted to be countered through experimental results. However, I am not convinced that the experimental results show a clear advantage. In fact, I don’t see a case where the new methods significantly outperformed the existing methods. The variance reduction idea from CTD was applied on top of TDC, so novelty-wise the method development did not add any significant contribution.

Correctness: Did not check the proofs in the appendix.

Clarity: The paper was well written. However, the intuition behind the CTD technique and how it resulted in variance reduction should have been added. Without that, it remains mysterious where these new methods come from.

Relation to Prior Work: Yes.

Reproducibility: Yes

Additional Feedback: In the discussion of off-policy learning, GTD is discussed in such a way as if that were the only way of achieving convergent off-policy TD methods. The discussion of the other path that leads to ETD is ignored. How does the proposed approach relate to ETD and is it extensible to that case as well? It will be useful to know as high variance is particularly problematic for ETD methods. **** After rebuttal **** Overall, I am bumping up my score a bit. Regarding the empirical results, you are right that VRTDs appear to outperform TDs, which I already noticed. But the errorbars appear to be overlapping till the end; hence my comment about the lack of statistical significance. Now the added figures are more convincing and helpful, which does alleviate my concern a bit. It seems with further attempts, better cases for VRTDC might be found. The fact that the bound for the Markovian setting couldn’t be improved over what already exists could be emphasized as an admission of a reducible gap (in addition to mentioning that it is near optimal) to encourage future researchers to improve further.

[Author Response · NeurIPS 2020]

We thank the reviewers for providing valuable comments. Below are point-to-point responses to the important questions.

**Reviewer 1:** Q1: Non-i.i.d. issue can be alleviated by experience replay. Markovian setting is not that significant.

A: We respectfully disagree with the reviewer. We agree that experience replay can be used in the offline Markovian
setting, in which the optimization problem involves finite samples and essentially falls into the i.i.d. setup. In comparison,
we study VRTDC in the online Markovian setting, which covers many real-world RL applications that have online
nature, e.g., traffic control, online portfolio optimization, etc. We also believe that studying policy evaluation algorithms
in the online Markovian setting has become an important fundamental topic for the RL theory community.

Q2: Keep problem's condition number in the complexity result.

A: Thanks for the suggestion. We will include all constants explicitly in the complexity result in the revision.

Q3: Summarize the bounds along with other existing algorithms' bounds in a table.

A: Thanks. We will add a table to compare our bounds with those of VRTD and TDC in i.i.d. and Markovian setting.

**Reviewer 2:** Q4: Intuition behind step-size choice $\alpha = O(\beta^{2/3})$?

A: For i.i.d. case, the inequality above Line 482 has the error term $c_1\beta + c_2(\alpha^2/\beta^2)$ for constants $c_1, c_2$. Minimizing it
yields the desired learning rate $\alpha = O(\beta^{2/3})$. For the Markovian case, we obtain the same error term in Line 703.

Q5: Better to provide tight error bounds even though they do not lead an improved complexity.

A: Thanks for the suggestion and we totally agree. We will derive refined bounds for the Markovian setting and update
them in the revision (it involves heavy computation that takes some time).

Q6: Empirically evaluate and compare the update variance.

A: In the following figure, we plot the estimated variance of the stochastic updates of $\theta$ (left) and $w$ (right) for different
algorithms. It can be seen that VRTDC significantly reduces the variance of TDC in both time-scales. Also, the variance
of $\theta$ updates of VRTDC is slightly smaller than that of VRTD. We will include these results in the revision.

21
**Reviewer 3:** Q7: Emphasize importance of two time-scale, variance reduction and technical difficulties.

A: Thanks for the suggestions. We will emphasize and elaborate on these issues in the revision.

**Reviewer 4:** Q10: No strict sample complexity improvement in the Markovian case.

A: We agree with the reviewer, and our complexity result almost match the lower bound of linear two time-scale SA in
Kaledin'20 (up to a logarithm factor).

Q11: The experimental results don't show a clear advantage.

A: In Figure 1 (left) and 2 (left), VRTDC achieves the highest solution accuracy, while TD and TDC cannot achieve a
high accuracy. This is the desired effectiveness of variance reduction (i.e., can find high-accuracy solutions). From the
above figure, it can be seen that VRTDC can significantly reduce the optimization variance of TDC in both time-scales.

Q12: Variance reduction idea is from CTD so there is no novel contribution.

A: VRTDC is the first variance reduction method for two time-scale Markovian TD learning. Our analysis requires to
deal with the coupled $\theta$ and $w$, which leads to the new development of recursively refined error bounds to decouple
these parameters and obtain the tight bounds.

Q13: Intuition behind the CTD and how it resulted in variance reduction should be added.

A: Thanks for the suggestion. We will discuss and elaborate on CTD with more details in the revision.

Q14: Discuss ETD. Can variance reduction be applied to ETD?

A: Thanks for pointing out the very interesting ETD method. Yes, one can still apply variance reduction to the ETD
update. However, the main difference is that ETD is a one time-scale algorithm, and its update involves an emphasis
factor $F_t$ (i.e., the discounted interests of the states in history). The analysis will need to bound the variance and
Markovian bias of ETD update in the presence of $F_t$. We expect that one needs to develop certain recursive bounds to
address this issue. We will discuss ETD and cite related references in the revision.

[Meta-Review · NeurIPS 2020]

The results in this work are novel and non-trivial. The reviewers generally supported the work. The key limitation to address is to be more clear about the theoretical and empirical outcomes in the paper. A point highlighted by a reviewer is that the rate is not improved in the Markov setting, with the variance reduction approach, and the experiments are not that compelling in showing that the rate is notably improved. It would be more useful to clearly state that you do not get a rate improvement in the Markov setting, and provide a discussion about this (somewhat negative, but nonetheless realistic) result. Additionally, it would be useful to more clearly explain the improved rate of VRTDC over VRTD. It is a big deal to show that a TDC variant converges faster than a TD variant. Explaining this result, and how it is possible, would more clearly motivate the approach. Here, it would also be useful to contrast VRTDC with the rate from TDC, not just compared to VRTD.